



# SURFER v3.0 : a fast model with ice sheet tipping points and carbon cycle feedbacks for short and long-term climate scenarios

Victor Couplet[1], Marina Martínez Montero[1], and Michel Crucifix[1]

[1]Earth and Life Institute, UCLouvain, Louvain-la-Neuve, Belgium

**Correspondence:** Victor Couplet (victor.couplet@uclouvain.be)

**Abstract.** Simple climate models that are computationally inexpensive, transparent, and easy to modify are useful for assessing climate policies in the presence of uncertainties. This motivated the creation of SURFER v2.0, a model designed to estimate the impact of $CO_2$ emissions and solar radiation modification on global mean temperatures, sea-level rise, and ocean pH. However, SURFER v2.0 is unsuitable for simulations beyond a few thousand years because it lacks some carbon cycle processes. This

is problematic for assessing the long-term evolution of ice sheets and the associated sea level rise. Here, we present SURFER v3.0, an extension to SURFER v2.0 that allows for accurate simulation of the climate, carbon cycle, and sea level rise on time scales ranging from decades to millions of years. We incorporated in the model a dynamic cycling of alkalinity in the ocean, a carbonate sediments reservoir, and weathering fluxes. With these additions, we show that SURFER v3.0 reproduces results from a large class of models, ranging from centennial CMIP6 projections to 1 Myr runs performed with the cGENIE model of

intermediate complexity. We show that compared to SURFER v2.0, including long-term carbon-cycle processes in SURFER v3.0 leads to a stabilisation of the Greenland ice sheet for the middle of the road emission scenarios, and a significant reduction in sea level rise contribution from Antarctica for high emissions scenarios.

## 1 Introduction

Human activities have significantly altered Earth's climate by releasing greenhouse gases, primarily carbon dioxide ($CO_2$) and

methane ($CH_4$), into the atmosphere at unprecedented rates (IPCC, 2021). The consequences of these emissions are already palpable, with 2023 marking the warmest year on record. Moreover, the frequency and intensity of extreme weather events, such as heatwaves, wildfires, droughts, and hurricanes have increased and are projected to increase further under global warming. While these impacts are immediate and observable, others unfold over longer time frames. For instance, the melting polar ice caps will contribute to sea level rise for centuries, and even millennia, after emissions of greenhouse gases are reduced or

stopped (Clark et al., 2016; Breedam et al., 2020). There is also growing concern over climate tipping points, where potentially abrupt and irreversible changes could occur and lead to cascades of unforeseen consequences in the long term trajectories of the Earth system (Lenton et al., 2008, 2019; Armstrong McKay et al., 2022; Steffen et al., 2018).

Making informed decisions about climate change thus necessitates a comprehensive examination across multiple temporal scales, to balance the immediate needs of current populations with the imperative of safeguarding the planet's habitability

for future generations. To this end, climate models are indispensable for scientists and policymakers. They come in different



sizes and complexities. On the one side of the complexity spectrum, state-of-the-art Earth system models include detailed representation of physical and biogeochemical processes. However, due to their size and complexity, they are hard to analyse and computationally expensive to run, with most simulations ending in 2100. On the other side of the spectrum, conceptual box models trade complexity and spatial resolution for speed and simplicity. They provide valuable insights into fundamental

climate processes and feedbacks, facilitating transparent and intuitive assessments of long-term climate trends. Due to their low computational cost, they can be run many times, with different choices of parameters and for different forcing scenarios. This allows an extensive exploration of mitigation and adaptation strategies, such as to take into account possible errors caused by simplifications and other lacks of knowledge.

This is the spirit in which SURFER was designed (Martínez Montero et al., 2022). SURFER v2.0 is a model based on nine

ordinary differential equations designed to estimate global mean temperature increase, sea level rise and ocean acidification in response to $CO_2$ emissions and aerosols injections. It is fast, easy to understand and modify, making it appropriate for use in policy assessments. For example, in Montero et al. (2023), it helped assess the long-term sustainability of short term climate policies based on a novel commitment metric. Yet, SURFER v2.0 lacks some processes in its carbon cycle implementation. It can only simulate quantities reliably for up to two or three millennia, and only accounts for carbon dioxide emissions in the

carbon cycle. Here, we introduce an extended version of the model with new processes, SURFER v3.0. Among other things, we have added: a representation of atmospheric methane, a distinction between land-use and fossil emissions, an additional oceanic layer, a dependence of the solubility and dissociation constants on temperature and pressure, a dynamic representation of alkalinity, a sediments box, and weathering processes.

The present paper is structured as follow. We explain in detail the new additions of SURFER v3.0 in section 2. In section 3,

we compare the results of SURFER v3.0 to other models on time scales ranging from centuries to a million year. We first show that SURFER v3.0 can reproduce the historical record. Then, we show that SURFER v3.0 is in the range of IPCC class models for different quantities modelled up to 2100 and 2300 AD. We next compare the $CO_2$ draw-down computed by SURFER over 10000 years to models that participated in the LTMIP project. Lastly, we compare SURFER v3.0 outputs to 1 Myr runs performed with the cGENIE climate model of intermediate complexity. In section 4, we show that including new processes

in the carbon cycle reduces the committed sea level rise (SLR) estimations on millennial time scales compared to SURFER v2.0. In section 5 we discuss the advantages and limitations of our model. Finally, in section 6, we conclude and provide some perspectives on future research.

## 2  Model specification

The nine ordinary differential equations of SURFER v2.0 describe the exchanges of carbon between four different reservoirs

(atmosphere, upper ocean layer, deeper ocean layer and land), the evolution of temperature anomalies of two different reservoirs (upper ocean layer and deeper ocean layer), the volumes of Greenland and Antarctic ice sheets, and the sea level rise related to glaciers (Martínez Montero et al., 2022). In version v3.0, we have added :

– an ocean layer of intermediate depth,





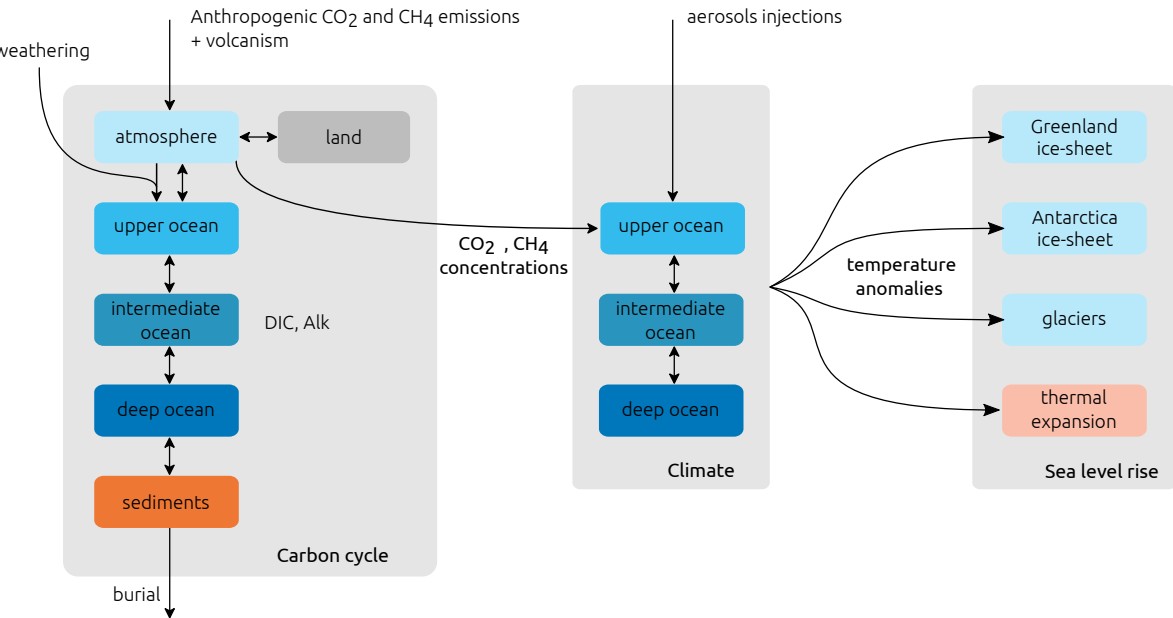

**Figure 1.** Conceptual diagram of SURFER v3.0.

- – a sediment box with $CaCO_3$ accumulation and burial fluxes,

- – dynamic alkalinity cycling between the 3 ocean layers,

- – an explicit description of the soft-tissue and carbonate pumps in the ocean,

- – volcanic outgassing and weathering fluxes,

- – an equation for the methane evolution in the atmosphere,

- – a temperature and depth (pressure) dependence of the solubility and dissociation constants,

- – a second equation for the land reservoir that allows for a better distinction between land-use and fossil greenhouse gas
  emissions.

SURFER v3.0 now consists of 17 coupled ordinary differential equations, while keeping its original architecture. It contains
three main components, for, respectively, the carbon cycle, climate, and the sea-level. The model is forced by land-use and
fossil emissions of $CO_2$ and $CH_4$, and by aerosols injections. State variables defining the model components, carbon and
energy fluxes between those components, and forcings are schematically summarised in Figure 1.





In this section, we explain in detail the model implementation. Sub-sections 2.1, 2.2, and 2.3 focus respectively on the implementation of the the carbon, climate and sea-level rise components. In sub-section 2.4 we show how the model was calibrated and how the initial conditions are chosen. Sub-section 2.5 discusses the algorithmic implementation and speed.

## 2.1 Carbon cycle component

The equations for the carbon cycle component of SURFER v3.0 are given by

$$\frac{dM_A}{dt} = V + E_{\text{fossil}}^{CO_2} + E_{\text{land-use}}^{CO_2} - F_{A \to U} - F_{A \to L} + F_{CH_4,\text{ox}} - E_{\text{natural}}^{CH_4} - F_{\text{weathering}} ,  \tag{1}$$

$$\frac{dM_A^{CH_4}}{dt} = E_{\text{fossil}}^{CH_4} + E_{\text{land-use}}^{CH_4} + E_{\text{natural}}^{CH_4} - F_{CH_4,\text{ox}} ,  \tag{2}$$

$$\frac{dM_L}{dt} = F_{A \to L} - E_{\text{land-use}}^{CO_2} - E_{\text{land-use}}^{CH_4} ,  \tag{3}$$

$$\frac{dM_U}{dt} = F_{A \to U} - F_{U \to I} + F_{\text{river}} ,  \tag{4}$$

$$\frac{dM_I}{dt} = F_{U \to I} - F_{I \to D} ,  \tag{5}$$

$$\frac{dM_D}{dt} = F_{I \to D} - F_{\text{acc}} ,  \tag{6}$$

$$\frac{d\tilde{Q}_U}{dt} = -\tilde{F}_{U \to I} + \tilde{F}_{\text{river}} ,  \tag{7}$$

$$\frac{d\tilde{Q}_I}{dt} = \tilde{F}_{U \to I} - \tilde{F}_{I \to D} ,  \tag{8}$$

$$\frac{d\tilde{Q}_D}{dt} = \tilde{F}_{I \to D} - \tilde{F}_{\text{acc}} ,  \tag{9}$$

$$\frac{dM_S}{dt} = F_{\text{acc}} - F_{\text{burial}} .  \tag{10}$$

They describe the fluxes of carbon between six reservoirs, the atmosphere (A), the land (L), the upper (U), intermediate (I), and deep (D) ocean layers, and the deep sea sediments (S). $M_A$ and $M_A^{CH_4}$ are the masses of carbon in the atmosphere in $CO_2$ and $CH_4$ forms respectively. $M_L$ is the mass of carbon on land (in vegetation and soils) and $M_S$ is the mass of carbon in the erodible $CaCO_3$ deep sea sediments. $M_U$, $M_I$ and $M_D$ are the masses of dissolved inorganic carbon in the ocean layers, while

$\tilde{Q}_U$, $\tilde{Q}_I$, $\tilde{Q}_D$ are total alkalinities. All these quantities are expressed in PgC, and the fluxes in $\text{PgC yr}^{-1}$ (equation 14 explains what this means for alkalinity).

Variable names are chosen to be self-explanatory, and tildes indicate quantities and fluxes related to alkalinity. Volcanism ($V$) and anthropogenic $CO_2$ emissions ($E_{\text{fossil}}^{CO_2}$, $E_{\text{land-use}}^{CO_2}$) increase the $CO_2$ content of the atmosphere. Methane anthropogenic emissions ($E_{\text{fossil}}^{CH_4}$, $E_{\text{land-use}}^{CH_4}$) as well as natural emissions ($E_{\text{natural}}^{CH_4}$) increase the $CH_4$ content in the atmosphere. Methane is

rapidly oxidized into $CO_2$ ($F_{CH_4,\text{ox}}$). Atmosphere and land reservoirs exchange $CO_2$ through photosynthesis and respiration (combined in $F_{A \to L}$). Weathering ($F_{\text{weathering}}$) removes carbon dioxide from the atmosphere through chemical reactions with rocks and minerals. The products of these reactions are then transported to the ocean by the rivers ($F_{\text{river}}$, $\tilde{F}_{\text{river}}$). Carbon is exchanged between the different layers of the ocean by mixing and the different carbon pumps ($F_{U \to I}$, $F_{I \to D}$, $\tilde{F}_{U \to I}$, $\tilde{F}_{I \to D}$).





A fraction of the carbon accumulates as sediments on the ocean floor ($F_\text{acc}$, $\tilde{F}_\text{acc}$) where it can be permanently buried ($F_\text{burial}$)

and removed from the system. Ultimately, plate tectonics movements transport this carbon in the mantle of the Earth (not
explicitly modelled in SURFER) where it will be transformed back to $CO_2$ and eventually emitted in the atmosphere through
volcanism, closing the cycle.

We detail the implementation of the above fluxes in sections 2.1.4 to 2.1.9. Before that, we motivate the addition of a
new oceanic layer (section 2.1.1) and discuss the treatment of alkalinity (section 2.1.2) and of the solubility and dissociation
constants (section 2.1.3).

### 2.1.1 Additional ocean layer

SURFER v2.0 used two ocean layers: an upper layer (U) of 150 m depth and a deep layer (D) of 3000 m depth. In SURFER
v3.0 we use three ocean layers: an upper layer (U) of 150 m depth, an intermediate layer (I) of 500 m depth, and a deep layer
(D) of 3150 m depth. Overall, the total ocean depth in SURFER v3.0 is greater than in SURFER v2.0 and closer to the global
mean estimate (∼3700 m). The new intermediate layer allows a smoother transition between the upper and deeper layers which
have different properties (temperature, salinity, pH, ...), see Figure 3. It also allows a faster carbon transport out of the upper
layer, because the exchange with the new intermediate layer is faster than with the old deep layer. This modification partly
fixes two problems we had in SURFER v2.0: $CO_2$ uptake by the ocean was too slow, and the surface pH decreased too fast
(surface ocean acidified too quickly). Indeed, since the dissolved $CO_2$ now leaves the upper layer quicker, the surface ocean
can absorb $CO_2$ from the atmosphere at faster rates. Moreover, with reduced "stagnation" of dissolved $CO_2$ in the upper layer,
surface acidity increases slower (pH decreases slower). Differences in atmosphere-to-ocean carbon flux and surface ocean pH
between SURFER v2.0 and SURFER v3.0 are shown in Figure 7.

### 2.1.2 Total alkalinity

Dissolved inorganic carbon (DIC) is defined as the sum of the carbonate species: carbonate ions $CO_3^{2-}$, bicarbonate ions $HCO_3^-$
and $H_2CO_3^*$ which represents a mix of aqueous carbon dioxide, $CO_2$(aqueous) and carbonic acid $H_2CO_3$. For concentrations,
we have

$$DIC = [H_2CO_3^*] + [HCO_3^-] + [CO_3^{2-}].$$ (11)

Alkalinity, on the other hand, is defined as the excess of bases over acids in water :

$$Alk = [HCO_3^-] + 2[CO_3^{2-}] + [OH^-] - [H^+] + [B(OH)_4^-] + \text{ minor bases}.$$ (12)

Intuitively, it measures the ability of a water mass to resist changes in pH when an acid is added. This happens for example when
excess $CO_2$ in the atmosphere dissolves in seawater (see eqs R1-R3). Alkalinity thus plays a crucial role in buffering ocean
acidification, which is important for many marine organisms who depend on stable pH levels for their survival and growth.
Being related to the concentrations of dissolved inorganic carbon species ($CO_3^{2-}$ and $HCO_3^-$), alkalinity also has a role in
regulating the oceanic uptake of atmospheric $CO_2$. In SURFER v2.0, alkalinity is considered constant and is furthermore





approximated by carbonate alkalinity $\text{Alk} \approx \text{Alk}_c = \left[\text{HCO}_3^-\right] + 2\left[\text{CO}_3^{2-}\right]$. In SURFER v3.0, we include $\text{CaCO}_3$ sediment dissolution, and this requires having variable alkalinity (equations 7-9). We estimate alkalinity based on the carbonate, borate and water self-ionisation alkalinity, which includes the contributions of the hydroxide ions $\left[\text{OH}^-\right]$, free protons $\left[\text{H}^+\right]$ and borate ions $\left[\text{B(OH)}_4^-\right]$:

$$\text{Alk} \approx \text{Alk}_{CBW} = \left[\text{HCO}_3^-\right] + 2\left[\text{CO}_3^{2-}\right] + \left[\text{OH}^-\right] - \left[\text{H}^+\right] + \left[\text{B(OH)}_4^-\right]. \tag{13}$$

This treatment produces more accurate computations of the concentration of chemical species than SURFER v2.0, specifically $\left[\text{H}^+\right]$ and thus pH, but comes at a greater computational cost. See appendix C for more details.

We use units of PgC for the variables representing alkalinity $\tilde{Q}_\text{U}$, $\tilde{Q}_\text{I}$, and $\tilde{Q}_\text{D}$ even though our working approximation now include terms that do no contain carbon. We do this for purely practical reasons, as all other variables of the carbon cycle component are also in PgC. The alkalinity concentration $\text{Alk}_i$ for the layer $i \in \{\text{U}, \text{I}, \text{D}\}$ in $\text{µmol kg}^{-1}$, a more common unit, is simply given by

$$\text{Alk}_i = \frac{\tilde{Q}_i}{W_i \bar{m}_\text{C}} \times 10^{18}, \tag{14}$$

where $W_i$ is the weight in kg of ocean layer $i$, and $\bar{m}_\text{C}$ is the molar mass of carbon in $\text{mol kg}^{-1}$. This is the same equation as for the conversion from dissolved organic carbon mass in PgC to concentration in $\text{µmol kg}^{-1}$

$$\text{DIC}_i = \frac{M_i}{W_i \bar{m}_\text{C}} \times 10^{18}. \tag{15}$$

The weight of layer $i$ is given by

$$W_i = \frac{h_i \bar{m}_\text{w} m_\text{O}}{h_\text{U} + h_\text{I} + h_\text{D}}, \tag{16}$$

with $\bar{m}_\text{w}$ the molar mass of water and $m_\text{O}$ the number of moles in the ocean.

### 2.1.3 Solubility and dissociation constants

When atmospheric $\text{CO}_2$ enters the ocean, it undergoes a series of chemical reactions

$$\text{CO}_{2(\text{gas})} + \text{H}_2\text{O} \quad \rightleftharpoons \quad \text{H}_2\text{CO}_3^*, \tag{R1}$$

$$\text{H}_2\text{CO}_3^* \quad \rightleftharpoons \quad \text{HCO}_3^- + \text{H}^+, \tag{R2}$$

$$\text{HCO}_3^- \quad \rightleftharpoons \quad \text{CO}_3^{2-} + 2\text{H}^+. \tag{R3}$$

These reactions are fast and we assume that they are in equilibrium. The relationships between the equilibrium concentrations of the different chemical species are determined by the dissociation constants :

$$K_1 = \frac{[\text{H}^+][\text{HCO}_3^-]}{[\text{H}_2\text{CO}_3^*]}, \tag{17}$$

$$K_2 = \frac{[\text{H}^+][\text{CO}_3^{2-}]}{[\text{HCO}_3^-]}, \tag{18}$$



Similarly, we have for the equilibrium concentrations of $OH^-$ and $B(OH)_4^-$

$$K_w = [H^+][OH^-],\tag{19}$$

$$K_b = \frac{[H^+][B(OH)_4^-]}{[H_3BO_3]},\tag{20}$$

where we additionally assume that the total equilibrium boron concentration is proportional to salinity (Uppström, 1974) :

$$TB = [B(OH)_4^-] + [H_3BO_3] = c_b \cdot S,\tag{21}$$

with $c_b = 11.88\,\mu mol\,kg^{-1}psu^{-1}$. The solubility of $CO_2$ in seawater, $K_0$, relates the concentration of $H_2CO_3^*$ and the partial pressure of dissolved $CO_2$, $p_{CO_2}$ :

$$K_0 = \frac{[H_2CO_3^*]}{p_{CO_2}}.\tag{22}$$

In SURFER v2.0, $K_0$, $K_1$ and $K_2$ were constant. In SURFER v3.0, $K_0$, $K_1$, $K_2$, $K_b$, and $K_w$ depend on temperature and salinity based on Weiss (1974); Mehrbach et al. (1973); Dickson and Millero (1987); Millero (1995); Dickson (1990). Salinity is assumed to be invariant on time, so we effectively only have a dependence on temperature . We also introduce a pressure (depth) dependence for $K_1$, $K_2$, $K_w$, and $K_b$, based on Millero (1995). This allows a better characterisation of pH in the deep ocean layer. Details on the parametrisations are in appendix B.

### 2.1.4 $F_{A \to U}$

We derive an expression for the atmosphere-to-ocean flux $F_{A \to U}$ in a slightly different way than in SURFER v2.0. The sea-air $CO_2$ exchange is proportional to the difference in $CO_2$ partial pressure between the atmosphere and the surface ocean layer. We have

$$F_{A \to U} = \bar{m}_C A_O \rho k K_0 (p_{CO_2}^A - p_{CO_2}^U),\tag{23}$$

where $A_O$ is the ocean surface area (in $m^2$), $\rho$ is the density of sea-water (in $kg\,m^3$), $\bar{m}_C$ is the molar mass of carbon (in $kg\,mol^{-1}$), $k$ is the gas transfer velocity (in $m\,yr^{-1}$) and $K_0$ is the solubility constant of $CO_2$ (in $mol\,kg^{-1}\,atm^{-1}$). This same expression is used by Lenton (2000) and Zeebe (2012), for carbon cycles models of similar complexity, with a multiplication by the molar mass of carbon $\bar{m}_C$ added here to obtain a flux in $kg\,yr^{-1}$ instead of $mol\,yr^{-1}$. We can express $p_{CO_2}^A$ and $p_{CO_2}^U$ in terms of model variables $M_A$ and $M_U$

$$F_{A \to U} = \bar{m}_C A_O \rho k K_0 (p_{CO_2}^A - p_{CO_2}^U),\tag{24}$$

$$= \bar{m}_C A_O \rho k K_0 \left( \frac{M_A}{m_A \bar{m}_C} \cdot 1\,atm - \frac{M_U'}{W_U \bar{m}_C K_0} \right),\tag{25}$$

$$= \frac{A_O \rho k}{m_A} K_0 \left( M_A \cdot 1\,atm - \frac{m_A}{W_U K_0} \frac{M_U'}{M_U} M_U \right),\tag{26}$$

$$= \bar{k}_{A \to U} K_0 \left( M_A \cdot 1\,atm - \frac{m_A}{W_U K_0} B_U(M_U, \tilde{Q}_U, T_U) M_U \right).\tag{27}$$





Here, $M'_U$ represents the mass of $H_2CO_3^*$ in the upper ocean layer. The factor $1$ atm is introduced for unit consistency and if $M_A$ and $M_U$ are expressed in PgC, then the flux will be in PgC yr$^{-1}$. We have defined $\bar{k}_{A\to U} = \frac{A_O \rho k}{m_A}$. We would recover the equation of SURFERv2.0 with $k_{A\to U} = \bar{k}_{A\to U} K_0$, but $K_0$ now depends on temperature. There are two advantages of proceeding as we did here compared to SURFER v2.0. First the coefficient $\bar{k}_{A\to U}$ is a function of well-identifed physical quantities. Second, we have not used the equilibrium condition for $F_{A\to U}(t_{PI})$ for the derivation of our expression, meaning that we can use it to constrain other quantities. Indeed, once $\bar{k}_{A\to U}$ and $M_A$ are set, the equilibrium condition $F_{A\to U}(t_{PI}) = -(F_{CaCO_3,0} + F_{CaSiO_3,0})$ will determine $M'_U$ (see section 2.4.2).

The term $B_U$ is a factor tracking the ocean's buffer capacity. In SURFER v2.0, it was a function of $M_U$ only. Now, the buffer factor also depends on the variable alkalinity ($\tilde{Q}_U$) and on temperature through the dissociation constants:

$$B_U\left(M_U, \tilde{Q}_U, T_U\right) \equiv \frac{M'_U}{M_U} = \frac{[H_2CO_3^*]_U}{DIC_U} = \left(1 + \frac{K_1}{[H^+]_U} + \frac{K_1 K_2}{[H^+]_U^2}\right)^{-1}. \tag{28}$$

To obtain this equation, we have used equations 17 and 18 to write DIC (eq 11) in terms of $[H_2CO_3^*]_U$ and $[H^+]$. In SURFER v2.0, we could compute analytically $[H^+]$ and $B_U$ as a function of $M_U$ and $\tilde{Q}_U$. We cannot do that anymore because we use a more complete approximation for alkalinity. We still compute $[H^+]$ as a function of $M_U$ and $\tilde{Q}_U$ (and the dissociation constants), but we have to numerically solve a degree 5 equation in $[H^+]$. Appendix C explains how we proceed.

### 2.1.5 $F_{U\to I}$, $F_{I\to D}$, $\tilde{F}_{U\to I}$, and $\tilde{F}_{I\to D}$

We now focus on carbon transport within the ocean. The processes and fluxes included in SURFER v3.0 are schematised in Figure 2. Carbon enters the upper ocean layer (U) through $CO_2$ exchanges with the atmosphere and through the riverine influx of weathering products. $CO_2$ intake from the atmosphere increases DIC but doesn't change alkalinity (see equations R1-R3). This is why we have a $F_{A\to U}$ DIC flux, but no $\tilde{F}_{A\to U}$ alkalinity flux. The riverine influx of weathering products increases DIC and alkalinity in the same amount (1:1 ratio); see section 2.1.7. We divide the exchange of carbon between the ocean layers into three parts which we explain below: the carbonate pump, the soft-tissue pump, and residual processes.

In the surface layer, calcifying organisms take up bicarbonate ions to form their calcium carbonate ($CaCO_3$) shells (forward reaction R4).

$$Ca^{2+} + 2HCO_3^- \rightleftharpoons CO_2 + H_2O + CaCO_3. \tag{R4}$$

These organisms eventually die and sink to the bottom ocean. On the way down, some of the $CaCO_3$ is dissolved (reverse reaction R4), resulting in a downward transport of DIC and alkalinity at a 1:2 ratio. This is the carbonate pump. We represent the export of $CaCO_3$ at a depth of 150 m (so from layer U to I) by $P^{CaCO_3}$. A fraction $\phi_I^{CaCO_3}$ of this export simultaneously dissolves in the intermediate layer (I) and a fraction $\phi_D^{CaCO_3}$ simultaneously dissolves in the deep layer (D), which leaves a fraction $1 - \phi_I^{CaCO_3} - \phi_D^{CaCO_3}$ reaching the sediments . Of that $CaCO_3$ that reaches the bottom of the ocean, some dissolves and some is permanently buried (details on this in section 2.1.6).

In the surface layer, organisms also take up carbon through photosynthesis (primary production, forward reaction R5).

$$106CO_2 + 16NO_3^- + HPO_4^{2-} + 122H_2O + 18H^+ \rightleftharpoons (CH_2O)_{106}(NH_3)_{16}(H_3PO_4) + 138O_2. \tag{R5}$$





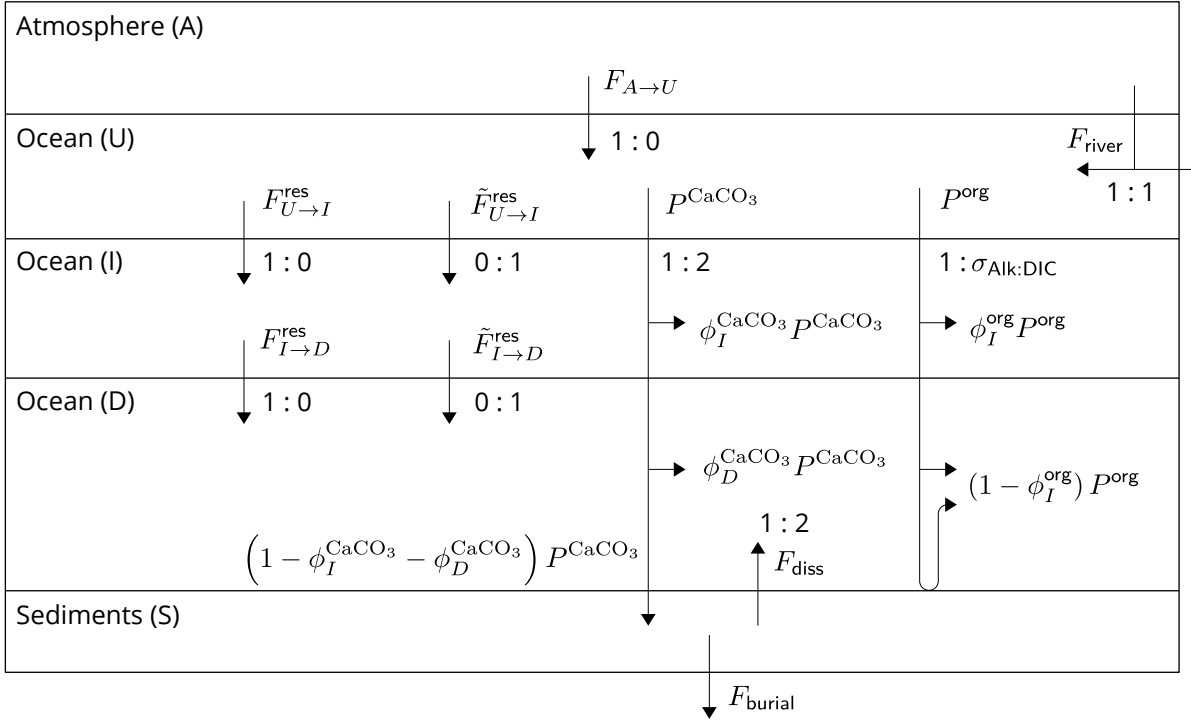

**Figure 2.** Schematic diagram of the DIC and Alk fluxes in SURFER v3.0. The ratios are ratios of DIC to alkalinity changes, i.e a $a : b$ ratio indicates that for a DIC change of $a$ moles, there is an associated alkalinity change of $b$ moles.

The $CO_2$ is transformed into organic carbon that will eventually sink to the bottom ocean. On the way down, some of the organic carbon is remineralised (reverse reaction R5), resulting in a downward transport of DIC. This is the soft-tissue pump. It also acts on alkalinity, mainly through the uptake and release of $H^+$ needed for the transformation of inorganic nitrate ($NO_3^-$) into organic nitrogen. Primary production of organic matter increases alkalinity, while remineralisation decreases alkalinity.

We represent the ratio of alkalinity to DIC change for primary production and remineralisation with the parameter $\sigma_{\text{Alk:DIC}}$. We represent the export of organic carbon at a depth of $150\,\text{m}$ (so from layer U to I) by $P^{\text{org}}$. We consider that all this organic carbon is simultaneously remineralised in the water column, with a fraction $\phi_{\text{I}}^{\text{org}}$ remineralised in the intermediate layer (I), and a fraction $\phi_{\text{D}}^{\text{org}} = 1 - \phi_{\text{I}}^{\text{org}}$ remineralised in the deep layer (D). In reality, some organic carbon accumulates on the sea-floor sediments where a large part is remineralised and only a small amount is permanently buried (Sarmiento and Gruber, 2006).

Here, by setting $\phi_{\text{D}}^{\text{org}} = 1 - \phi_{\text{I}}^{\text{org}}$, we neglect this burial and effectively consider that all organic carbon that falls on sediments is remineralised.

Apart from the carbonate and soft-tissue pumps, forming the biological pumps, other processes are responsible for the transport of carbon between the ocean layers. Ocean circulation and mixing will propagate variations in DIC caused by spatial and temporal variations of air-sea gas exchanges. This is termed the gas exchange pump by Sarmiento and Gruber (2006).



For example, a component of the gas exchange pump is the solubility pump (Volk and Hoffert, 1985): cold waters have a higher solubility and are thus enriched in $CO_2$, they are also denser and will sink in the high latitudes, resulting in a downward transport of DIC. Besides, upwelling is responsible for an upward transport of DIC and alkalinity, necessary to counteract the carbonate and soft-tissue pumps. We gather all the processes related to ocean circulation and mixing in the residual terms $F_{U \to I}^{res}$ and $F_{I \to D}^{res}$, $\tilde{F}_{U \to I}^{res}$ and $\tilde{F}_{I \to D}^{res}$. We consider the residual fluxes of DIC and alkalinity to be independent because some processes

such as the solubility pump only act on DIC.

Based on the above considerations, the masses of dissolved inorganic carbon and the mass of carbon in the sediments evolve according to the following equations:

$$\frac{dM_U}{dt} = F_{A \to U} - P^{CaCO_3} - P^{org} - F_{U \to I}^{res} + F_{river}, \tag{29}$$

$$\frac{dM_I}{dt} = \phi_I^{CaCO_3} P^{CaCO_3} + \phi_I^{org} P^{org} + F_{U \to I}^{res} - F_{I \to D}^{res}, \tag{30}$$

$$\frac{dM_D}{dt} = \phi_D^{CaCO_3} P^{CaCO_3} + \left(1 - \phi_I^{org}\right) P^{org} + F_{I \to D}^{res} + F_{diss}, \tag{31}$$

$$\frac{dM_S}{dt} = (1 - \phi_I^{CaCO_3} - \phi_D^{CaCO_3}) P^{CaCO_3} - F_{diss} - F_{burial}. \tag{32}$$

We recover equations 1-10 by setting

$$F_{U \to I} = F_{U \to I}^{CaCO_3} + F_{U \to I}^{org} + F_{U \to I}^{res}, \tag{33}$$

$$F_{I \to D} = F_{I \to D}^{CaCO_3} + F_{I \to D}^{org} + F_{I \to D}^{res}, \tag{34}$$

$$F_{acc} = (1 - \phi_I^{CaCO_3} - \phi_D^{CaCO_3}) P^{CaCO_3} - F_{diss}, \tag{35}$$

with the fluxes associated to the carbonate pump defined as

$$F_{U \to I}^{CaCO_3} = P^{CaCO_3}, \tag{36}$$

$$F_{I \to D}^{CaCO_3} = (1 - \phi_I^{CaCO_3}) P^{CaCO_3}, \tag{37}$$

and the fluxes associated to the soft-tissue pump defined as

$$F_{U \to I}^{org} = P^{org}, \tag{38}$$

$$F_{I \to D}^{org} = (1 - \phi_I^{org}) P^{org}. \tag{39}$$

Consequently, for the carbonate alkalinity fluxes, we have

$$\tilde{F}_{U \to I} = 2 \times F_{U \to I}^{CaCO_3} + \sigma_{Alk:DIC} \times F_{U \to I}^{org} + \tilde{F}_{U \to I}^{res}, \tag{40}$$

$$\tilde{F}_{I \to D} = 2 \times F_{I \to D}^{CaCO_3} + \sigma_{Alk:DIC} \times F_{I \to D}^{org} + \tilde{F}_{I \to D}^{res}, \tag{41}$$

$$\tilde{F}_{acc} = 2 \times F_{acc}. \tag{42}$$

In the experiments presented here (sections 3 and 4), we keep $P^{CaCO_3}$, $\phi_I^{CaCO_3}$, $\phi_D^{CaCO_3}$, $P^{org}$, and $\phi_I^{org}$ constant. This is an idealisation. Production and export of $CaCO_3$ and organic matter are biological processes and, in reality, depend on the





temperature, pH, salinity, nutrient concentration and other properties of the ocean. For example, changes in primary production (and thus export of organic matter) may have contributed to the $CO_2$ changes during the glacial-interglacial cycles (Kohfeld
and Ridgwell (2009)). In the future, changes in the biological pumps are also possible and might lead to additional feedbacks in the carbon cycle (Henson et al., 2022; Planchat et al., 2023).

For the residual exchange terms, we adopt a linear formalism with exchanges coefficients, similarly as in SURFER v2.0:

$$F_{U \to I}^{res} = k_{U \to I} M_U - k_{I \to U} M_I, \tag{43}$$

$$F_{I \to D}^{res} = k_{I \to D} M_I - k_{D \to I} M_D, \tag{44}$$

$$\tilde{F}_{U \to I}^{res} = \tilde{k}_{U \to I} \tilde{Q}_U - \tilde{k}_{I \to U} \tilde{Q}_I, \tag{45}$$

$$\tilde{F}_{I \to D}^{res} = \tilde{k}_{I \to D} \tilde{Q}_I - \tilde{k}_{D \to I} \tilde{Q}_D. \tag{46}$$

### 2.1.6 $F_{acc}$, $\tilde{F}_{acc}$, and $F_{burial}$

The $CaCO_3$ raining on the ocean floor either accumulates and gets buried in sediments, or dissolves, depending on the saturation state of the ocean waters with respect to $CO_3^{2-}$. Typically, the upper ocean is supersaturated in $CO_3^{2-}$ while the deeper
ocean is under-saturated, mainly due to the pressure dependence of $CaCO_3$ solubility. This means that most of the accumulation in sediments will happen in a region above where most of the dissolution happens. A transition zone separates the accumulation and dissolution regions. The top boundary of the transition zone is the lysocline, defined as the depth where the calcium carbonate content of sediments starts to decrease sharply, or in other words, the depth below which the rate of dissolution of $CaCO_3$ starts to increase significantly. The bottom boundary of the transition zone is called the carbonate (or calcite)
compensation depth (CCD) and is the depth at which the rate of $CaCO_3$ dissolution is equal to the rate of supply through the $CaCO_3$ rain. Below this depth, no $CaCO_3$ is preserved in the sediments. The depth of the transition zone varies between places and ocean basins but is generally between 3000 and 5000 m deep. We may thus safely consider in our model that both dissolution and accumulation happen in the deep ocean layer (D), and that is why we group both processes in a single term, $F_{acc}$ which can be positive or negative, negative values indicating net dissolution. We neglect $CaCO_3$ on continental shelves.
Locally, the dissolution rate depends on the concentration of $CaCO_3$ in the sediments and the saturation state of pore water around the sediments with respect to carbonate (Sarmiento and Gruber, 2006). We thus assume that the globally integrated dissolution flux is a function of two variables only: the deep ocean mean concentration of $CO_3^{2-}$, and the total mass of erodible $CaCO_3$ sediments. We use the following parametrisation

$$F_{diss} = \begin{cases} (1 - \phi_I^{CaCO_3} - \phi_D^{CaCO_3})P^{CaCO_3} & \text{if } M_S = 0 \text{ and } D > (1 - \phi_I^{CaCO_3} - \phi_D^{CaCO_3})P^{CaCO_3}, \\ D & \text{otherwise}, \end{cases} \tag{47}$$

with

$$D = F_{diss,0} + \alpha_{diss} \left( [CO_3^{2-}]_D - [CO_3^{2-}]_D(t_{PI}) \right) + \beta_{diss} (M_S - M_S(t_{PI}))$$
$$+ \gamma_{diss} \left( [CO_3^{2-}]_D - [CO_3^{2-}]_D(t_{PI}) \right) (M_S - M_S(t_{PI})). \tag{48}$$

The first case in equation 47 indicates that if the erodible sediments reservoir is empty, the dissolution flux can't exceed the $CaCO_3$ rain flux, regardless of the saturation state of deep waters. The accumulation fluxes of DIC and alkalinity, $F_{acc}$ and $\tilde{F}_{acc}$, are then given by equations 35 and 42. A similar parametrisation is proposed and tested in Archer et al. (1998), but they use the $CO_3^{2-}$ concentration at a particular point in the deep pacific and the mass of $CaCO_3$ in the bioturbated layer of the sediments as their two variables, instead of the mean deep ocean $CO_3^{2-}$ and the mass of erodible $CaCO_3$. They show that the parametrised version of their model is comparable to the non-parametrised one, except for a dissolution spike in the first 1000-2000 years following the fossil fuel emissions and invasion in the ocean. We choose the coefficient of our parametrisation based on theirs. Details can be found in section 2.4.1.

In the sediments, the bioturbated layer is the layer where sediments are mixed by biological activity (bioturbation). Dissolution only occurs in the top few centimetres near the sediment-ocean interface but can effectively reach deeper because of bioturbation. If the dissolution rate is greater than the rain rate of $CaCO_3$ on the sea floor, the sediments will lose $CaCO_3$ until the bioturbated layer becomes saturated with non-erodible material. At this point, dissolution stops and deeper carbonates are isolated from the carbon cycle. This explains why the effective reservoir of sediment carbonates to be accounted for here ($M_S$), has a limited size, which is estimated to be around 1600 PgC as of today (Archer et al., 1998). The erodible depth is defined as the lower boundary of the erodible sediments inventory (Archer et al., 1998). By definition, $CaCO_3$ sediments that move below the erodible depth will never be dissolved and we say that they are buried. Locally, the burial rate depends on the rain rate of non-erodible material and the concentration of $CaCO_3$ in sediments at the erodible depth. We assume that the former is constant and that the total burial flux only depends on the total mass of erodible $CaCO_3$. We use the following linear parametrisation

$$F_{burial} = \alpha_{burial} M_S \, . \tag{49}$$

### 2.1.7 $V$, $F_{weathering}$, $F_{river\ influx}$, $\tilde{F}_{river\ influx}$

Continental weathering of carbonate and silicate rocks can absorb $CO_2$ out of the atmosphere through the following reactions (Berner et al., 1983; Walker et al., 1981) :

$$CO_2 + H_2O + CaCO_3 \quad \rightleftharpoons \quad Ca^{2+} + 2HCO_3^- \, , \tag{R6}$$
$$2CO_2 + H_2O + CaSiO_3 \quad \rightleftharpoons \quad Ca^{2+} + 2HCO_3^- + SiO_2 \, . \tag{R7}$$

Let us consider $F_{CaCO_3}$ and $F_{CaSiO_3}$ the fluxes of $Ca^{2+}$ produced by these two processes. For carbonate weathering (equation R6), the production of one mole of $Ca^{2+}$ consumes one mole of carbon ($CO_2$) from the atmosphere and produces two moles of DIC and alkalinity that are transported by the rivers to the ocean. For the silicate weathering (equation R7), the production of one mole of $Ca^{2+}$ consumes two moles of carbon ($CO_2$) from the atmosphere and produces two moles of DIC and alkalinity



that are transported by the rivers to the ocean. Hence, we set

$$F_{\text{weathering}} = F_{\text{CaCO}_3} + 2F_{\text{CaSiO}_3} \tag{50}$$

$$F_{\text{river influx}} = 2F_{\text{CaCO}_3} + 2F_{\text{CaSiO}_3} \tag{51}$$

$$F_{\text{river influx}}^{alk} = 2F_{\text{CaCO}_3} + 2F_{\text{CaSiO}_3} \tag{52}$$

Like with alkalinity, we use $\text{PgC yr}^{-1}$ for $F_{\text{CaCO}_3}$ and $F_{\text{CaSiO}_3}$, even though they are defined as fluxes of $\text{Ca}^{2+}$. To go from $\text{Tmolyr}^{-1}$ to $\text{PgC yr}^{-1}$, one just has to divide by the molar mass of carbon $12 \cdot 10^{-3} \text{ mol kg}^{-1}$.

For every two moles of DIC produced by carbonate weathering, one mole of carbon is taken from the atmosphere reservoir and one mole of carbon comes from carbonate rocks on land, which are not explicitly described as a reservoir in our model. This
extra mole is thus treated as an external source of carbon, like volcanism. The carbon entering the ocean from weathering fluxes will eventually precipitate back as $\text{CaCO}_3$ in the sediments, which releases $\text{CO}_2$. Thus, the net effect of carbonate weathering is to transfer $\text{CaCO}_3$ from rocks on land to sediments in the ocean, but with no long-term net effect on the atmospheric $\text{CO}_2$. On the other hand, since silicate weathering consumes one more mole of $\text{CO}_2$ from the atmosphere for the same $\text{Ca}^{2+}$ flux, its net effect is to remove carbon from the atmosphere. At equilibrium, this net removal is compensated by volcanic outgassing.

The $F_{\text{CaCO}_3}$ and $F_{\text{CaSiO}_3}$ fluxes aren't constant and depend on many factors. For example, temperature affects the rates of reactions R6 and R7, water run-off on land influences how much under-saturated water will come in contact with rocks for weathering, and vegetation may affect the acidity of soils and thus the rates of dissolution. Colbourn et al. (2013) provide parametrisations of all these processes for the model of terrestrial rock weathering RockGEM, which was included in the GENIE Earth system modelling framework. They showed that the temperature feedback was the dominant one and we choose
to consider this one only. Following their parametrisation, we set

$$F_{\text{CaCO}_3} = F_{\text{CaCO}_3,0}(1 + k_{\text{Ca}}\delta T_{\text{U}}), \tag{53}$$

$$F_{\text{CaSiO}_3} = F_{\text{CaSiO}_3,0}e^{k_{\text{T}}\delta T_{\text{U}}}, \tag{54}$$

where $k_{\text{Ca}}$ and $k_{\text{T}}$ are constants and $\delta T_{\text{U}}$ is the temperature anomaly of the upper ocean (and atmosphere) modelled by SURFER.

**2.1.8 Methane**

The evolution of the methane concentration in the atmosphere is mainly controlled by 3 processes: natural and anthropogenic emissions increase the $\text{CH}_4$ concentration whereas oxidation into $\text{CO}_2$ decreases it.

Anthropogenic emissions can be divided in two sources, fossil emissions ($E_{\text{fossil}}^{\text{CH}_4}$) and land-use emissions ($E_{\text{land-use}}^{\text{CH}_4}$). Fossil emissions come from the industry sector and the use and exploitation of fossil fuels. Land-use emissions result from agriculture
(rice production, cattle, ...), agricultural waste burning, and burning of biomass such as forests, grasslands and peat. Natural emissions ($E_{\text{natural}}^{\text{CH}_4}$) mainly come the from anaerobic decomposition of organic matter in wetlands, but also from freshwater systems, termites, and geological sources such as volcanoes, permafrost and methane hydrates. For a detailed treatment of





the different methane emissions see Saunois et al. (2020). The rate of natural emissions may depend on temperature, and if they increase upon global warming, this could lead to positive feedbacks and eventually tipping points (Nisbet et al., 2023; 350  Fewster et al., 2022; Archer et al., 2009). For simplicity, we assume constant natural emissions. To ensure the conservation of carbon, land-use $CH_4$ emissions are taken from the land reservoir, while natural $CH_4$ emissions are taken directly from the $CO_2$ atmospheric reservoir of carbon. The reason for this difference is explained in the next section.

Oxidation of methane ($F_{CH_4,ox}$, equation R8) is the main sink of methane out of the atmosphere and releases $CO_2$:

$$CH_4 + 2O_2 \rightarrow CO_2 + 2H_2O \,. \tag{R8}$$

We describe this a simple decay process :

$$F_{CH_4,ox} = -\frac{M_A^{CH_4}}{\tau_{CH_4}} \,, \tag{55}$$

where $\tau_{CH_4}$ is the atmospheric $CH_4$ lifetime. In principle, it may vary depending on temperature and on the availability of the hydroxyl radical OH, which is necessary for the intermediate steps of reaction R8, and which itself depends on the concentrations of $CH_4$, $N_2O$, CO and other trace gases. However, for simplicity, we choose to keep $\tau_{CH_4}$ constant and set its 360  value to 9.5 yr. We add the product of oxidation to $M_A$.

### 2.1.9  Land reservoir and land-use emissions

In SURFER v3.0, we distinguish greenhouse gas emissions caused by fossil fuel use from those caused by land use. While fossil $CO_2$ and $CH_4$ emissions are directly added to the atmosphere, land-use $CO_2$ and $CH_4$ emissions ($E_{land-use}^{CO_2}$ and $E_{land-use}^{CH_4}$) must be taken from the land reservoir (eq 3). In SURFER v2.0, based on the outputs of the ZECMIP experiments (Jones et al., 365  2019; MacDougall et al., 2020), we parametrised the carbon flux from the land to the atmosphere in the following way

$$F_{A \rightarrow L} = k_{A \rightarrow L} \left( \beta_L M_{A(t_{PI})} \left( 1 - \frac{M_A(t_{PI})}{M_A} \right) - (M_L - M_L(t_{PI})) \right) . \tag{56}$$

This is equivalent to saying that $M_L$ relaxes to an equilibrium mass $M_L^{eq}(M_A)$ equal to

$$M_L^{eq}(M_A) = \beta_L M_A(t_{PI}) \left( 1 - \frac{M_A(t_{PI})}{M_A} \right) + M_L(t_{PI}), \tag{57}$$

with a timescale $1/k_{A \rightarrow L}$.

Now suppose that we have a certain amount of land-use emissions; we transfer some carbon out of the land reservoir into the atmosphere reservoir, and let the model evolve without changing anything else. Then the $F_{A \rightarrow L}$ flux, as it is, will increase and land will absorb the carbon lost until initial equilibrium is reached again. Physically, this means that the forest that was replaced by grassland or crop fields has grown back to its original size. In the real world, this may happen, but if the land is managed (e.g. for agriculture), the forest does not regrow. To take this into account, we subtract the cumulative $CO_2$ land-use 375  emissions from the equilibrium mass to which $M_L$ relaxes

$$M_L^{eq}(M_A,t) = \beta_L M_{A(t_{PI})} \left( 1 - \frac{M_A(t_{PI})}{M_A} \right) + M_L(t_{PI}) - \int_{t_0}^{t} E_{land-use}^{CO_2}(t) \,. \tag{58}$$





This is the same procedure as in the model from Lenton (2000), except that they subtract only a fraction of the cumulative land-use emissions, allowing for some forest regrowth. Here we subtract all land-use emissions because, in principle, the negative emissions coming from forest regrowth should already be accounted for in the net reported land-use emissions (IPCC, 2021,
p.688). We can rewrite the new $F_{A \to L}$ flux as

$$F_{A \to L} = k_{A \to L} \left( \beta_L M_{A(t_{PI})} \left( 1 - \frac{M_A(t_{PI})}{M_A} \right) - (M_L - M_{L^*}) \right), \tag{59}$$

where $M_L^*$ is a new model variable, the evolution of which is described by

$$\frac{dM_L^*}{dt} = -E_{\text{land-use}}^{CO_2}(t), \tag{60}$$

with $M_L^*(t_{PI}) = M_L(t_{PI})$.

We haven't included methane land-use emissions in equations 58-60. This means that the carbon mass in $CH_4$ form taken from the land reservoir is reabsorbed by land once methane is oxidized back to $CO_2$. In other words, methane land-use emissions do not cause a net addition of $CO_2$ in the atmosphere through oxidation. This makes sense if you consider methane emissions coming from the anaerobic decomposition of organic matter in rice cultures or cattle stomachs. Indeed, the decomposed organic matter was formed not too long before by absorbing $CO_2$ from the atmosphere through photosynthesis, so when
the methane oxidises into $CO_2$, it closes the loop and there is no net effect on atmospheric $CO_2$.

The same reasoning applies to most natural emissions of methane which arise from wetlands: they do not cause a net increase of atmospheric $CO_2$ concentrations. In principle, these natural methane emissions should be subtracted from the land carbon reservoir, as anthropogenic land-use emissions, and there should be a non-zero atmosphere-to-land equilibrium flux that compensates the $CO_2$ created from oxidation. However, this is impossible with our parametrisation of the atmosphere-
to-land flux, which is zero at preindustrial times by construction. To avoid introducing yet another parameterisation or a more detailed representation of the carbon on land and to maintain carbon conservation, we merely subtract the natural emissions directly from the $CO_2$ mass of carbon in the atmosphere. This approach is justified by our assumption that the natural methane production-oxidation cycle is in equilibrium.

Natural $CH_4$ emissions coming from geological processes such as volcanism or natural gas leaks would have a net impact
on $CO_2$ levels but they are negligible compared to emissions coming from wetlands (Saunois et al., 2020). Natural $CH_4$ emissions coming from permafrost would also have a lasting effect on $CO_2$ concentrations because the organic matter from which they originate was formed thousands of years before. Accounting for emissions from permafrost would require yet another parametrisation, and we rather neglected them in SURFER v3.0.

### 2.2   Climate component

The equations for the climate component are essentially the same as in SURFER v2.0, with the addition of an oceanic box and the radiative forcing due to methane. The atmosphere is considered to be in thermal equilibrium with the surface ocean layer



(U). The evolutions of temperature anomalies for the three oceanic layers are dictated by

$$c_{\mathrm{vol}} h_{\mathrm{U}} \frac{d\delta T_{\mathrm{U}}}{\mathrm{d}t} = F\left(M_{\mathrm{A}}, M_{\mathrm{A}}^{\mathrm{CH_4}}, I\right) - \beta \delta T_{\mathrm{U}} - \gamma_{\mathrm{U}\to\mathrm{I}}\left(\delta T_{\mathrm{U}} - \delta T_{\mathrm{I}}\right), \tag{61}$$

$$c_{\mathrm{vol}} h_{\mathrm{I}} \frac{d\delta T_{\mathrm{I}}}{\mathrm{d}t} = \gamma_{\mathrm{U}\to\mathrm{I}}\left(\delta T_{\mathrm{U}} - \delta T_{\mathrm{I}}\right) - \gamma_{\mathrm{I}\to\mathrm{D}}\left(\delta T_{\mathrm{I}} - \delta T_{\mathrm{D}}\right), \tag{62}$$

$$c_{\mathrm{vol}} h_{\mathrm{D}} \frac{d\delta T_{\mathrm{D}}}{\mathrm{d}t} = \gamma_{\mathrm{I}\to\mathrm{D}}\left(\delta T_{\mathrm{I}} - \delta T_{\mathrm{D}}\right), \tag{63}$$

where $F\left(M_{\mathrm{A}}, M_{\mathrm{A}}^{\mathrm{CH_4}}, I\right)$ is the anthropogenic radiative forcing. Its expression is given by

$$F\left(M_{\mathrm{A}}, M_{\mathrm{A}}^{\mathrm{CH_4}}, I\right) = F_{2\times} \log_2\left(\frac{M_{\mathrm{A}}}{M_{\mathrm{A}}\left(t_{\mathrm{PI}}\right)}\right) + \alpha_{\mathrm{CH_4}} \sqrt{M_{\mathrm{A}}^{\mathrm{CH_4}} - M_{\mathrm{A}}^{\mathrm{CH_4}}(t_{\mathrm{PI}})} - \alpha_{\mathrm{SO_2}} \exp\left(-\left(\beta_{\mathrm{SO_2}}/I\right)^{\gamma_{\mathrm{SO_2}}}\right). \tag{64}$$

The first two terms describe the contribution of $CO_2$ and methane to an increased greenhouse effect. The third term corresponds to solar radiation modification in the form of $SO_2$ injections (Martínez Montero et al., 2022).

## 2.3 Sea level rise sub-model

Sea level rise is estimated as the sum of four contributions: thermal expansion, and melt from the mountain glaciers, the Greenland ice sheet, and the Antarctic ice sheet:

$$S_{\mathrm{tot}} = S_{\mathrm{th}} + S_{\mathrm{gl}} + S_{\mathrm{GIS}} + S_{\mathrm{AIS}}. \tag{65}$$

Compared to SURFER v2.0, we only change the parametrisation of thermal expansion, where we need to add a term to take into account the new intermediate layer.

$$S_{\mathrm{th}} = \alpha_{\mathrm{U}} h_{\mathrm{U}} \delta T_{\mathrm{U}} + \alpha_{\mathrm{I}} h_{\mathrm{I}} \delta T_{\mathrm{I}} + \alpha_{\mathrm{D}} h_{\mathrm{D}} \delta T_{\mathrm{D}}. \tag{66}$$

Here $\alpha_i$ is the thermal expansion coefficient corresponding to layer $i \in \{\mathrm{U},\mathrm{I},\mathrm{D}\}$. The other contributions are the same as in Martínez Montero et al. (2022). We recall them here, and more details are provided in the original publication.

The evolution of the sea-level rise contribution from glaciers is given by the equation

$$\frac{\mathrm{d}S_{\mathrm{gl}}}{\mathrm{d}t} = \frac{1}{\tau_{\mathrm{gl}}}\left(S_{\mathrm{gleq}}\left(\delta T_{\mathrm{U}}\right) - S_{\mathrm{gl}}\right), \tag{67}$$

with

$$S_{\mathrm{gleq}}\left(\delta T_{\mathrm{U}}\right) = S_{\mathrm{gl\,pot}} \tanh\left(\frac{\delta T_{\mathrm{U}}}{\zeta}\right). \tag{68}$$

Here $\tau_{\mathrm{gl}}$ is a relaxation timescale, $S_{\mathrm{gl\,pot}}$ is the potential sea level rise due to mountain glaciers and $\zeta$ is a sensitivity coefficient.

The contributions from Greenland and Antarctica are given by

$$S_{\mathrm{GIS}} = S_{\mathrm{GIS}}\left(1 - V_{\mathrm{GIS}}(t)\right), \quad S_{\mathrm{AIS}} = S_{\mathrm{AIS}}\left(1 - V_{\mathrm{AIS}}(t)\right), \tag{69}$$



where $S_{\mathrm{GIS}}$ and $S_{\mathrm{AIS}}$ are the sea-level rise potential of Greenland's and Antarctic ice sheet and $V_{\mathrm{GIS}}$, $V_{\mathrm{AIS}}$ are the volume fractions of the ice sheets with respect to their preindustrial volume. The evolution of ice sheets volumes are described with the equation

$$\frac{\mathrm{d}V}{\mathrm{d}t} = \mu\left(V, \delta T_{\mathrm{U}}\right) \underbrace{\left(-V^3 + a_2 V^2 + a_1 V + c_1 \delta T_{\mathrm{U}} + c_0\right)}_{H}, \tag{70}$$

with

$$\mu\left(V, \delta T_{\mathrm{U}}\right) = \begin{cases} 1/\tau & \text{if } H > 0 \text{ or } (H < 0 \text{ and } V > 0), \\ 0 & \text{if } H < 0 \text{ and } V = 0, \end{cases} \tag{71}$$

and

$$\tau = \tau_- + \frac{\tau_+ - \tau_-}{2}\left(1 + \tanh\left(\frac{H}{k_\tau}\right)\right). \tag{72}$$

The time scales $\tau_+$ and $\tau_-$ are associated with the asymmetric processes of freezing and melting. The first case in equation 71
was separated in two cases in SURFER v2.0, depending on the sign of $H$. In SURFER v3.0, we introduce a smooth transition between $\tau_+$ and $\tau_-$ when $H$ changes sign. This formulation effectively prevents small fluctuations around the equilibrium having different timescales (when $H$ is close to zero). The parameter $k_\tau$ controls the smoothness of the transition, SURFER v2.0 being recovered with $k_\tau = \infty$. The constant parameters $(a_2, a_1, c_1, c_0)$ are given in terms of $(T_+, V+), (T_-, V_-))$, which are the bifurcation points of the steady state structure induced by equation 70:

$$a_2 = \frac{3\left(V_- + V_+\right)}{2}, \tag{73}$$

$$a_1 = -3V_- V_+, \tag{74}$$

$$c_1 = -\frac{\left(V_+ - V_-\right)^3}{2\left(T_+ - T_-\right)}, \tag{75}$$

$$c_0 = +\frac{T_+ V_-^2\left(V_- - 3V_+\right) - T_- V_+^2\left(V_+ - 3V_-\right)}{2\left(T_- - T_+\right)}. \tag{76}$$

These relationships allow SURFER to be easily calibrated on the steady-state structure of more complex ice-sheet models.

## 2.4  Calibration and initial conditions

We calibrate the parameters and initial conditions of the model using known physics, observations, model results, and the hypothesis that the carbon cycle was at equilibrium during preindustrial times. This follows standard practice, even though




processes involving longer time scales active at glacial-interglacial time scales are not necessarily in balance. We assume

$$0 = V - F_{A \to U}(t_{PI}) - F_{A \to L}(t_{PI}) + F_{CH_4,ox}(t_{PI}) - E_{nat}^{CH_4} - F_{weathering}(t_{PI}),\tag{77}$$

$$0 = E_{nat}^{CH_4} - F_{CH_4,ox}(t_{PI}),\tag{78}$$

$$0 = F_{A \to L}(t_{PI}),\tag{79}$$

$$0 = F_{A \to U}(t_{PI}) - F_{U \to I}(t_{PI}) + F_{river}(t_{PI}),\tag{80}$$

$$0 = F_{U \to I}(t_{PI}) - F_{I \to D}(t_{PI}),\tag{81}$$

$$0 = F_{I \to D}(t_{PI}) - F_{acc}(t_{PI}),\tag{82}$$

$$0 = -\tilde{F}_{U \to I}(t_{PI}) + \tilde{F}_{river}(t_{PI}),\tag{83}$$

$$0 = \tilde{F}_{U \to I}(t_{PI}) - \tilde{F}_{I \to D}(t_{PI}),\tag{84}$$

$$0 = \tilde{F}_{I \to D}(t_{PI}) - \tilde{F}_{acc}(t_{PI}),\tag{85}$$

$$0 = F_{acc}(t_{PI}) - F_{burial}(t_{PI}).\tag{86}$$

From equations 83-85 we get that $\tilde{F}_{acc}(t_{PI}) = \tilde{F}_{I \to D}(t_{PI}) = \tilde{F}_{U \to I}(t_{PI}) = \tilde{F}_{river}(t_{PI}) = 2F_{CaCO_3,0} + 2F_{CaSiO_3,0}$. The dissolution/precipitation of $CaCO_3$ produces/consumes moles of DIC and alklanity at a 1:2 ratio (see section 2.1.6), hence we have $F_{acc}(t_{PI}) = \frac{1}{2}\tilde{F}_{acc}(t_{PI}) = F_{CaCO_3,0} + F_{CaSiO_3,0}$. We then get from equations 81,82 and 86 that $F_{U \to I}(t_{PI}) = F_{I \to D}(t_{PI}) = F_{burial}(t_{PI}) = F_{acc}(t_{PI}) = F_{CaCO_3,0} + F_{CaSiO_3,0}$. Equation 80 gives us $F_{A \to L}(t_{PI}) = -(F_{CaCO_3,0} + F_{CaSiO_3,0})$, equation 78 tells us that $E_{nat}^{CH_4} = F_{CH_4,ox}(t_{PI})$, and finally, from equation 77 we find $V = F_{CaSiO_3,0}$. Equation 79 provides no extra information since $F_{A \to L}(t_{PI}) = 0$ by construction. Developing the expressions of the carbon fluxes, we get the following system of equations :

$$V = F_{CaSiO_3,0},\tag{87}$$

$$E_{nat}^{CH_4} = \frac{M_A^{CH_4}(t_{PI})}{\tau_{CH_4}},\tag{88}$$

$$0 = 0,\tag{89}$$

$$\bar{k}_{A \to U}K_0\left(M_A(t_{PI}) - \frac{m_A}{W_U K_0}M'_U(t_{PI})\right) = -(F_{CaCO_3,0} + F_{CaSiO_3,0}),\tag{90}$$

$$P^{CaCO_3} + P^{org} + k_{U \to I}M_U(t_{PI}) - k_{I \to U}M_I(t_{PI}) = F_{CaCO_3,0} + F_{CaSiO_3,0},\tag{91}$$

$$(1 - \phi_I^{CaCO_3})P^{CaCO_3} + (1 - \phi_I^{org})P^{org} + k_{I \to D}M_I(t_{PI}) - k_{D \to I}M_D(t_{PI}) = F_{CaCO_3,0} + F_{CaSiO_3,0},\tag{92}$$

$$2P^{CaCO_3} + \sigma_{Alk:DIC}P^{org} + \tilde{k}_{U \to I}\tilde{Q}_U(t_{PI}) - \tilde{k}_{I \to U}\tilde{Q}_I(t_{PI}) = 2(F_{CaCO_3,0} + F_{CaSiO_3,0}),\tag{93}$$

$$2(1 - \phi_I^{CaCO_3})P^{CaCO_3} + \sigma_{Alk:DIC}(1 - \phi_I^{org})P^{org} + \tilde{k}_{I \to D}\tilde{Q}_I(t_{PI}) - \tilde{k}_{D \to I}\tilde{Q}_D(t_{PI}) = 2(F_{CaCO_3,0} + F_{CaSiO_3,0}),\tag{94}$$

$$(1 - \phi_I^{CaCO_3} - \phi_D^{CaCO_3})P^{CaCO_3} - F_{diss,0} = F_{CaCO_3,0} + F_{CaSiO_3,0},\tag{95}$$

$$\alpha_{burial}M_S(t_{PI}) = F_{CaCO_3,0} + F_{CaSiO_3,0}.\tag{96}$$





The equilibrium hypothesis effectively provides us with constraints linking some parameters and initial conditions. In section 2.4.1 we discuss the choice and calibration of the model parameters. A sensitivity analysis for most parameters is presented in appendix D. In section 2.4.2, we provide a set of initial conditions for the model.

### 2.4.1 Parameters

**Carbon cycle submodel**

The parameters that control the $CO_2$ uptake by vegetation ($\beta_L$, $k_{A\rightarrow L}$) and the $CO_2$ uptake by the ocean on short time scales ($\bar{k}_{A\rightarrow U}$, $k_{U\rightarrow I}$ and $\tilde{k}_{U\rightarrow I}$ ) are tuned to reproduce historical observations of $CO_2$ concentrations (Figure 4), historical estimations of land and ocean sinks (Figure 6) as well as historical and SSP runs of CMIP6 (Figures 7 and 8). The parameters $k_{I\rightarrow D}$ and $\tilde{k}_{I\rightarrow D}$, which control the ocean carbon uptake on multi-centennial to multi-millennial time scales, are chosen to

produce a reasonable fit with the 1 Myr runs of cGENIE (see section 3.4). Overall, the calibration to other models is performed qualitatively, without optimising well-defined metrics.

The parameter $\bar{k}_{A\rightarrow U}$ is defined as $\bar{k}_{A\rightarrow U} = A_O \rho k / m_A$. With ocean area $A_O = 361 \times 10^{12}$ m$^2$, mean sea water density $\rho = 1026$ kg m$^{-3}$, gas transfer velocity $20$ cm h$^{-1} = 0.2 \cdot 365 \cdot 24$ m yr$^{-1}$ (Yang et al., 2022), and number of moles in the atmosphere $m_A = 1.727 \times 10^{20}$ mol, we obtain $\bar{k}_{A\rightarrow U} = 4.7$ kg(mol yr)$^{-1}$. The parameter $\beta_L$, which controls the amount of

$CO_2$ uptake by vegetation (see eq 59), is set to 1.7. This is the same value as for SURFER v2.0, which was calibrated on the outputs of the ZECMIP experiments. The parameter $k_{A\rightarrow L}$ is set to 0.044, which is an increase by a factor 1.75 compared to SURFER v2.0. This is done to have a better match with historical $CO_2$ concentrations.

The parameters that dictate the DIC and alkalinity exchanges between the ocean layers are physically determined by the oceanic circulation, hence we expect them to be similar ($k_{i\rightarrow j} \approx \tilde{k}_{i\rightarrow j}$ for $i,j \in \{U,I,D\}$), but we do not require them to be

equal because some processes such as the solubility pump can impact DIC and alkalinity independently. Yet, for simplicity, we set both $k_{U\rightarrow I}$ and $\tilde{k}_{U\rightarrow I}$ to 0.13 yr$^{-1}$, and both $k_{I\rightarrow D}$ and $\tilde{k}_{I\rightarrow D}$ to 0.009 yr$^{-1}$. Then, $k_{I\rightarrow U}$ and $\tilde{k}_{I\rightarrow U}$, $k_{D\rightarrow I}$ and $\tilde{k}_{D\rightarrow I}$ are computed from equations 91-94. We have

$$k_{I\rightarrow U} = \left( P^{CaCO_3} + P^{org} - (F_{CaCO_3,0} + F_{CaSiO_3,0}) + k_{U\rightarrow I} M_U(t_{PI}) \right) \frac{1}{M_I(t_{PI})}, \tag{97}$$

$$k_{D\rightarrow I} = \left( (1 - \phi_I^{CaCO_3}) P^{CaCO_3} + (1 - \phi_I^{org}) P^{org} - (F_{CaCO_3,0} + F_{CaSiO_3,0}) + k_{I\rightarrow D} M_I(t_{PI}) \right) \frac{1}{M_D(t_{PI})}, \tag{98}$$

$$\tilde{k}_{I\rightarrow U} = \left( 2 P^{CaCO_3} + \sigma_{Alk:DIC} P^{org} - 2(F_{CaCO_3,0} + F_{CaSiO_3,0}) + \tilde{k}_{U\rightarrow I} \tilde{Q}_U(t_{PI}) \right) \frac{1}{\tilde{Q}_I(t_{PI})}, \tag{99}$$

$$\tilde{k}_{D\rightarrow I} = \left( 2(1 - \phi_I^{CaCO_3}) P^{CaCO_3} + \sigma_{Alk:DIC}(1 - \phi_I^{org}) P^{org} - 2(F_{CaCO_3,0} + F_{CaSiO_3,0}) + \tilde{k}_{I\rightarrow D} \tilde{Q}_I(t_{PI}) \right) \frac{1}{\tilde{Q}_D(t_{PI})}. \tag{100}$$

With the choices for the other parameters described hereafter, we obtain $k_{I\rightarrow U} = 0.0383$ yr$^{-1}$, $\tilde{k}_{I\rightarrow U} = 0.0392$ yr$^{-1}$, $k_{D\rightarrow I} = 1.44 \times 10^{-3}$ yr$^{-1}$, and $\tilde{k}_{D\rightarrow I} = 1.43 \times 10^{-3}$ yr$^{-1}$. Overall, this corresponds to a time scale range of 7.7 - 26.1 yr ($1/k_{U\rightarrow I}$ - $1/k_{I\rightarrow U}$) for the oceanic carbon exchanges between surface and intermediate layers, and a time scale range of 111.1 - 693.8 yr

($1/k_{I\rightarrow D}$ - $1/k_{D\rightarrow I}$) for the oceanic carbon exchanges between intermediate and deep layers. This also gives preindustrial DIC fluxes of 175 PgC yr$^{-1}$ for subduction ($k_{U\rightarrow I} M_U(t_{PI})$) and 183 PgC yr$^{-1}$ for obduction ($k_{I\rightarrow U} M_I(t_{PI})$). Although these



carbon fluxes are an order of magnitude greater than the fluxes from the biological pumps, they are much less studied, and estimates in the literature are rare. IPCC AR5 gave estimates of $90\,\mathrm{PgC\,yr^{-1}}$ and $101\,\mathrm{PgC\,yr^{-1}}$ for subduction and obduction respectively (Stocker et al., 2013), while IPCC AR6 gave estimates of $264\,\mathrm{PgC\,yr^{-1}}$ and $275\,\mathrm{PgC\,yr^{-1}}$ (Levy et al., 2013; Canadell et al., 2021).

The $\mathrm{CaCO_3}$ export out of the ocean surface has been estimated between 0.6 and $1.8\,\mathrm{PgC\,yr^{-1}}$ (see supplementary material in Sulpis et al. (2021) and references therein). Sulpis et al. (2021) provide a tighter range of 0.77 to $1.06\,\mathrm{PgC\,yr^{-1}}$, of which 0.34 to $0.53\,\mathrm{PgC\,yr^{-1}}$ are dissolved in the water column before reaching the sediments. We set $P^{\mathrm{CaCO_3}}$ to $1\,\mathrm{PgC\,yr^{-1}}$, which is also the value given in Sarmiento and Gruber (2006) for the open-ocean export at a 100 m depth. We set $\phi_{\mathrm{I}}^{\mathrm{CaCO_3}} = 0.15$ and $\phi_{\mathrm{D}}^{\mathrm{CaCO_3}} = 0.39$. This gives us a total of $0.54\,\mathrm{PgC\,yr^{-1}}$ dissolved in the water column and $0.46\,\mathrm{PgC\,yr^{-1}}$ that rains on the sediments, which is close to estimates from Sulpis et al. (2021) and Sarmiento and Gruber (2006, see Figure 9.1.1).

Estimates of the export of organic carbon out of the euphotic zone range from 4 to $12\,\mathrm{PgC\,yr^{-1}}$ (DeVries and Weber (2017), and references therein). The euphotic zone is the uppermost layer of the ocean that receives sunlight and where photosynthesis can happen. Since the remineralisation of organic matter is quite fast in the water column, estimates of carbon export vary greatly depending on the specific definition of the euphotic zone and its depth. Based on a data-assimilated model, DeVries and Weber (2017) give an estimate of $9.1\pm0.2\,\mathrm{PgC\,yr^{-1}}$ for the organic carbon export out of the euphotic zone and an estimate of $6.7\,\mathrm{PgC\,yr^{-1}}$ for the organic carbon export at 100 m depth. In our model, we set the organic carbon export $P^{\mathrm{org}}$ at a 150 m depth to $7\,\mathrm{PgC\,yr^{-1}}$, which corresponds to the estimate given for the open ocean in Sarmiento and Gruber (2006), and we set $\phi_{\mathrm{I}}^{\mathrm{org}} = 0.72$. Equation R5 suggests that for a 106-mole decrease in DIC due to organic matter production, alkalinity will increase by 17 moles (the uptake of 18 moles of $\mathrm{H}^+$ increases alkalinity by 18 moles, while the uptake of $\mathrm{HPO_4^{2-}}$, which is one of the minor bases included in the full definition of alkalinity (eq 12), decreases alkalinity by one mole). Here, we follow Sarmiento and Gruber (2006) and set $\sigma_{\mathrm{Alk:DIC}}$ to $-16/117$, meaning that for 1 mole uptake of DIC in organic matter production, alkalinity is increased by $16/117 = 0.14$ moles. Setting $\sigma_{\mathrm{Alk:DIC}} = -16/117$ instead of $\sigma_{\mathrm{Alk:DIC}} = -17/106$ doesn't result in much change, as can be seen from the sensitivity analysis in appendix D.

For our parametrisation of $\mathrm{CaCO_3}$ dissolution, we need to calibrate 4 parameters. The value of $F_{\mathrm{diss,0}}$ is computed using equilibrium conditions. From equation 95, we obtain

$$F_{\mathrm{diss,0}} = (1 - \phi_{\mathrm{I}}^{\mathrm{CaCO_3}} - \phi_{\mathrm{D}}^{\mathrm{CaCO_3}})P^{\mathrm{CaCO_3}} - (F_{\mathrm{CaCO_3,0}} + F_{\mathrm{CaSiO_3,0}}) \tag{101}$$

With the choice of $P^{\mathrm{CaCO_3}}$, $\phi_{\mathrm{I}}^{\mathrm{CaCO_3}}$, $\phi_{\mathrm{D}}^{\mathrm{CaCO_3}}$ described above, and the choice for $F_{\mathrm{CaCO_3,0}}$, $F_{\mathrm{CaSiO_3,0}}$ described below, we get $F_{\mathrm{diss,0}} = 0.33\,\mathrm{PgC\,yr^{-1}}$. The parameters $\alpha_{\mathrm{diss}}$, $\beta_{\mathrm{diss}}$ and $\gamma_{\mathrm{diss}}$ are obtained based on a parametrisation provided in Archer et al. (1998) for accumulation (rain minus dissolution). The values we use are given in table 2.

We split the total weathering flux evenly between silicate and carbonate weathering ($F_{\mathrm{CaCO_3,0}} = F_{\mathrm{CaSiO_3,0}}$) and we set them such as to obtain a preindustrial burial flux of $0.13\,\mathrm{PgC\,yr^{-1}}$. This is the estimate given in Sarmiento and Gruber (2006). For comparison, the IPCC AR6 WG1 gives an estimate of $0.2\,\mathrm{PgC\,yr^{-1}}$. From equation 89, we have $F_{\mathrm{CaCO_3,0}} + F_{\mathrm{CaSiO_3,0}} = \alpha_{\mathrm{burial}}M_{\mathrm{S}}(t_{\mathrm{PI}}) = F_{\mathrm{burial}}(t_{\mathrm{PI}}) = 0.13\,\mathrm{PgC\,yr^{-1}}$ and this gives us $F_{\mathrm{CaCO_3,0}} = F_{\mathrm{CaSiO_3,0}} = 0.065\,\mathrm{PgC\,yr^{-1}}$ or $5.42\,\mathrm{Tmol\,yr^{-1}}$. This is close to the value of $\sim 5.6\,\mathrm{Tmol\,yr^{-1}}$ used in Colbourn et al. (2015) and Lord et al. (2016) for similar





parametrisations employed in cGENIE (see section 3.4). From these choices, we get $\alpha_{\text{burial}} = (F_{\text{CaCO}_3,0} + F_{\text{CaSiO}_3,0})/M_{\text{S}}(t_{\text{PI}}) = 0.13/M_{\text{S}}(t_{\text{PI}})$. Volcanic outgassing is set to $V = F_{\text{CaSiO}_3,0} = 0.065$ PgC yr$^{-1}$ as per equation 87. The IPCC estimate is 0.1 PgC yr$^{-1}$. Following Colbourn et al. (2013), we set $k_{\text{Ca}}$ to 0.049 K$^{-1}$ and $k_{\text{T}}$ to

$$k_{\text{T}} = \frac{1000 E_a}{R T_0^2}, \tag{102}$$

where $R$ is the gas constant (in J K$^{-1}$mol$^{-1}$), $T_0$ is the global mean preindustrial temperature (in K) and $E_a$ is the activation energy for dissolution (in kJ mol$^{-1}$). West et al. (2005) provide an estimate for the activation energy: $E_a = 74 \pm 29$ kJ mol$^{-1}$. With $T_0 = T_{\text{U}}(t_{\text{PI}}) = 288.38$ K (as set in section 2.4.2), this gives a range for $k_{\text{T}}$ between 0.065 K$^{-1}$ and 0.149 K$^{-1}$ . We set $k_{\text{T}} = 0.095$ K$^{-1}$ .

Natural methane emissions are set to $E_{\text{nat}}^{\text{CH}_4} = M_{\text{A}}^{\text{CH}_4}(t_{\text{PI}})/\tau_{\text{CH}_4}$. With $M_{\text{A}}^{\text{CH}_4}(t_{\text{PI}})$ chosen as in section 2.4.2 and $\tau_{\text{CH}_4} = 9.5$
yr, we get natural emissions of 0.157 PgC yr$^{-1}$ or 209 TgCH$_4$ yr$^{-1}$. This is in the range of the top-down estimate of the IPCC (176-243 TgCH$_4$ yr$^{-1}$) and a bit below the bottom-up estimate range (245-484 TgCH$_4$ yr$^{-1}$) (Canadell et al., 2021).

| Symbol | Comment | Value |
|--------|---------|-------|
| $m_{\text{A}}$ | number of moles in the atmosphere | $1.727 \times 10^{20}$ mol |
| $m_{\text{O}}$ | number of moles in the ocean | $7.8 \times 10^{22}$ mol |
| $\bar{m}_{\text{C}}$ | carbon molar mass | $12 \times 10^{-3}$ kg mol$^{-1}$ |
| $\bar{m}_{\text{w}}$ | water molar mass | $18 \times 10^{-3}$ kg mol$^{-1}$ |
| $c_{\text{vol}}$ | sea water volumetric heat capacity | $0.13$ W yr m$^{-3}$ |
| $R$ | gas contant | $8.314$ J mol$^{-1}$K$^{-1}$ |
| $h_{\text{U}}$ | upper layer depth | $150$ m |
| $h_{\text{I}}$ | intermediate layer depth | $500$ m |
| $h_{\text{D}}$ | deep layer depth | $3150$ m |
| $W_{\text{U}}$ | upper layer weight | $h_{\text{U}}\bar{m}_{\text{w}}m_{\text{O}}/(h_{\text{U}} + h_{\text{I}} + h_{\text{D}})$ |
| $W_{\text{I}}$ | intermediate layer weight | $h_{\text{I}}\bar{m}_{\text{w}}m_{\text{O}}/(h_{\text{U}} + h_{\text{I}} + h_{\text{D}})$ |
| $W_{\text{D}}$ | deep layer weight | $h_{\text{D}}\bar{m}_{\text{w}}m_{\text{O}}/(h_{\text{U}} + h_{\text{I}} + h_{\text{D}})$ |
| $z_{\text{U}}$ | upper layer depth mid-depth point | $75$ m |
| $z_{\text{I}}$ | intermediate layer mid-depth point | $400$ m |
| $z_{\text{D}}$ | deep layer mid-depth point | $2225$ m |

**Table 1.** Physical parameters and geometry.

**Climate submodel**

For the parametrisation of the heat exchange between the ocean layers, we first distinguished $\gamma_{\text{U}\to\text{I}}$ and $\gamma_{\text{I}\to\text{D}}$, and ended up setting $\gamma_{\text{U}\to\text{I}} = \gamma_{\text{I}\to\text{D}} = 0.8357$ W m$^{-2}$ °C$^{-1}$. This is the value chosen for the unique $\gamma$ in SURFER v2.0. This gives us a
transient climate response (TCR) of 1.9 °C , well in the likely range of 1.4°C - 2.2°C given by the IPCC (Forster et al., 2021). In section 4 we show that this choice gives a good fit to the estimated heat uptake by the deep ocean in the period 1971-2018.




| Symbol | Comment | Value |
|---|---|---|
| $F_{\mathrm{CaCO_3},0}$ | PI weathering of carbonate rocks | $0.065\,\mathrm{PgC\,yr^{-1}}$ |
| $F_{\mathrm{CaSiO_3},0}$ | PI weathering of silicate rocks | $0.065\,\mathrm{PgC\,yr^{-1}}$ |
| $k_{\mathrm{Ca}}$ | parametrisation of carbonate weathering flux | $0.049\,\mathrm{K^{-1}}$ |
| $k_{\mathrm{T}}$ | parametrisation of silicate weathering flux | $0.095\,\mathrm{K^{-1}}$ |
| $k_{\mathrm{A\to L}}$ | controls rate of carbon uptake by vegetation | $0.044\,\mathrm{yr^{-1}}$ |
| $\beta_{\mathrm{L}}$ | controls amount of carbon uptake by vegetation | $1.7$ |
| $\bar{k}_{\mathrm{A\to U}}$ | controls rate of air-sea $CO_2$ exchanges | $4.7\,\mathrm{kg\,(mol\,yr)^{-1}}$ |
| $k_{\mathrm{U\to I}}$ | controls rate of DIC transfer via ocean mixing from layer U to I | $0.13\,\mathrm{yr^{-1}}$ |
| $k_{\mathrm{I\to D}}$ | controls rate of DIC transfer via ocean mixing from layer I to D | $0.009\,\mathrm{yr^{-1}}$ |
| $\tilde{k}_{\mathrm{U\to I}}$ | controls rate of alkalinity transfer via ocean mixing from layer U to I | $0.13\,\mathrm{yr^{-1}}$ |
| $\tilde{k}_{\mathrm{I\to D}}$ | controls rate of alkalinity transfer via ocean mixing from layer I to D | $0.009\,\mathrm{yr^{-1}}$ |
| $P^{\mathrm{org}}$ | organic matter export at 150 m | $7\,\mathrm{PgC\,yr^{-1}}$ |
| $P^{\mathrm{CaCO_3}}$ | $CaCO_3$ export at 150 m | $1\,\mathrm{PgC\,yr^{-1}}$ |
| $\phi_{\mathrm{I}}^{\mathrm{org}}$ | fraction of organic matter rain that remineralises in layer I | $0.72$ |
| $\phi_{\mathrm{I}}^{\mathrm{CaCO_3}}$ | fraction of $CaCO_3$ rain that dissolves in layer I | $0.15$ |
| $\phi_{\mathrm{D}}^{\mathrm{CaCO_3}}$ | fraction of $CaCO_3$ rain that dissolves in layer D | $0.39$ |
| $\sigma_{\mathrm{Alk:DIC}}$ | alkalinity to DIC changes in organic matter production and remineralisation | $-16/117$ |
| $\alpha_{\mathrm{diss}}$ | parametrisation of $CaCO_3$ sediments dissolution | $-1.07\times10^{-2}\,\mathrm{PgC\,yr^{-1}(\mu mol\,kg^{-1})^{-1}}$ |
| $\beta_{\mathrm{diss}}$ | parametrisation of $CaCO_3$ sediments dissolution | $1.82\times10^{-5}\,\mathrm{yr^{-1}}$ |
| $\gamma_{\mathrm{diss}}$ | parametrisation of $CaCO_3$ sediments dissolution | $-4.53\times10^{-6}\,\mathrm{yr^{-1}\,(\mu mol\,kg^{-1})^{-1}}$ |
| $\tau_{\mathrm{CH_4}}$ | atmospheric lifetime of methane | $9.5\,\mathrm{yr}$ |
| $V$ | volcanic outgassing | $F_{\mathrm{CaSiO_3},0}$ |
| $k_{\mathrm{I\to U}}$ | controls rate of DIC transfer via ocean mixing from layer I to U | eq 97 |
| $k_{\mathrm{D\to I}}$ | controls rate of DIC transfer via ocean mixing from layer D to I | eq 98 |
| $\tilde{k}_{\mathrm{I\to U}}$ | controls rate of alkalinity transfer via ocean mixing from layer I to U | eq 99 |
| $\tilde{k}_{\mathrm{D\to I}}$ | controls rate of alkalinity transfer via ocean mixing from layer D to I | eq 100 |
| $F_{\mathrm{diss},0}$ | parametrisation of $CaCO_3$ sediments dissolution | eq 101 |
| $\alpha_{\mathrm{burial}}$ | parametrisation of $CaCO_3$ sediments burial | $(F_{\mathrm{CaCO_3},0}+F_{\mathrm{CaSiO_3},0})/M_{\mathrm{S}}(t_{\mathrm{PI}})$ |
| $E_{\mathrm{nat}}^{\mathrm{CH_4}}$ | natural methane emissions | $M_{\mathrm{CH_4}}(t_{\mathrm{PI}})/\tau_{\mathrm{CH_4}}$ |

**Table 2.** Parameters for the carbon cycle component. Second part of the table contains parameters that are computed from equilibrium conditions (equations 87-96).



| Symbol | Comment | Value |
|---|---|---|
| $F_{2\times}$ | extra radiative forcing due to a doubling of atmospheric $CO_2$ | $3.9\ \mathrm{W\,m^{-2}}$ |
| $\beta$ | climate feedback parameter | $1.1143\ \mathrm{W\,m^{-2}\,{}^\circ C^{-1}}$ |
| $\gamma_{\mathrm{U\to I}}$ | parametrisation of heat exchange between ocean layers | $0.8357\ \mathrm{W\,m^{-2}\,{}^\circ C^{-1}}$ |
| $\gamma_{\mathrm{I\to D}}$ | parametrisation of heat exchange between ocean layers | $0.8357\ \mathrm{W\,m^{-2}\,{}^\circ C^{-1}}$ |
| $\alpha_{\mathrm{CH_4}}$ | parametrisation of radiative forcing of methane | $0.791\ \mathrm{W\,m^{-2}\,PgC^{-1/2}}$ |
| $\alpha_{\mathrm{SO_2}}$ | parametrisation of radiative forcing of $SO_2$ | $65\ \mathrm{W\,m^{-2}}$ |
| $\beta_{\mathrm{SO_2}}$ | parametrisation of radiative forcing of $SO_2$ | $2246\ \mathrm{TgS\,yr^{-1}}$ |
| $\gamma_{\mathrm{SO_2}}$ | parametrisation of radiative forcing of $SO_2$ | $0.23$ |

**Table 3.** Parameters for the climate component.

For the contribution of methane to the radiative forcing $F_{\mathrm{CH_4}}$ (in $\mathrm{W\,m^{-2}}$), we use a common parametrisation (Myhre et al., 1998)

$$F_{\mathrm{CH_4}} = 0.036\sqrt{C_{\mathrm{CH_4}} - C_{\mathrm{CH_4}}(t_{\mathrm{PI}})}, \tag{103}$$

where $C_{\mathrm{CH_4}}$ is the methane atmospheric concentration in ppb. We have

$$C_{\mathrm{CH_4}} = \frac{M_{\mathrm{A}}^{\mathrm{CH_4}}}{\bar{m}_{\mathrm{C}} m_{\mathrm{A}}} 10^{21}, \tag{104}$$

where $M^{\mathrm{CH_4}}$ is expressed in $\mathrm{PgC}$, and thus

$$\alpha_{\mathrm{CH_4}} = 0.036\sqrt{\frac{10^{21}}{\bar{m}_{\mathrm{C}} m_{\mathrm{A}}}}. \tag{105}$$

**SLR submodel**

The thermal expansion coefficient (for the density) of a water parcel is defined as $\alpha = \frac{1}{\rho}\frac{\partial \rho}{\partial t}|_{P,S}$, where $\rho$, $P$, and $S$ are the density, the pressure and salinity of that water parcel. To obtain the averaged thermal expansion coefficients for each ocean layer $\alpha_{\mathrm{U}}$, $\alpha_{\mathrm{I}}$ and $\alpha_{\mathrm{D}}$, we proceed in three steps, as in Williams et al. (2012). First, we use the GLODAPv2.2016b mapped climatology (Lauvset et al., 2016) to compute the thermal expansion coefficient at each ocean point. To do this, we use the International Thermodynamic Equation Of Seawater - 2010 (TEOS-10) and the Python implementation of the GSW

Oceanographic toolbox of TEOS-10 (McDougall and Barker, 2011). Second, we average over each horizontal level of the GLODAPv2.2016b climatology to obtain a vertical profile of the thermal expansion coefficient. Third, we average over each of our defined ocean layers, using the areas of each horizontal level as weights for the horizontally averaged values of the thermal expansion coefficient. We obtain $\alpha_{\mathrm{U}} = 2.20 \times 10^{-4}\,\mathrm{K^{-1}}$, $\alpha_{\mathrm{I}} = 1.61 \times 10^{-4}\,\mathrm{K^{-1}}$ and $\alpha_{\mathrm{D}} = 1.39 \times 10^{-4}\,\mathrm{K^{-1}}$. This is close to the values used in SURFER v2.0 ($\alpha_{\mathrm{U}} = 2.3 \times 10^{-4}\,\mathrm{K^{-1}}$ and $\alpha_{\mathrm{D}} = 1.3 \times 10^{-4}\,\mathrm{K^{-1}}$). In section 4, we show that these

expansion coefficients give a good fit to the thermosteric sea level rise on multi-millennial time scales as simulated by the Earth





| Parameter | Comment | Value |
|-----------|---------|-------|
| $S_{\mathrm{gl\,pot}}$ | sea level rise potential from mountain glaciers | $0.5\,\mathrm{m}$ |
| $\xi$ | sensitivity coefficient for glacier parametrisation | $2°\mathrm{C}$ |
| $\tau_{\mathrm{gl}}$ | timescale for glacier melt | $200\,\mathrm{yr}$ |
| $\alpha_{\mathrm{U}}$ | thermal expansion coefficient for layer U | $2.20 \times 10^{-4}\,\mathrm{K}^{-1}$ |
| $\alpha_{\mathrm{I}}$ | thermal expansion coefficient for layer I | $1.61 \times 10^{-4}\,\mathrm{K}^{-1}$ |
| $\alpha_{\mathrm{D}}$ | thermal expansion coefficient for layer D | $1.39 \times 10^{-4}\,\mathrm{K}^{-1}$ |

**Table 4.** Parameter values for the sea level rise component.

| Parameter | Greenland's value | Antarctica's value |
|-----------|-------------------|--------------------|
| $T_+$ | $1.52\,°\mathrm{C}$ | $6.8\,°\mathrm{C}$ |
| $T_-$ | $0.3\,°\mathrm{C}$ | $4.0\,°\mathrm{C}$ |
| $V_+$ | 0.77 | 0.44 |
| $V_-$ | 0.3527 | -0.3200 |
| $\tau_+$ | $5500\,\mathrm{yr}$ | $5500\,\mathrm{yr}$ |
| $\tau_-$ | $470\,\mathrm{yr}$ | $3000\,\mathrm{yr}$ |
| $k_\tau$ | 0.05 | 0.05 |
| $S_{\mathrm{pot}}$ | $7.4\,\mathrm{m}$ | $55\,\mathrm{m}$ |

**Table 5.** Parameter values used for Greenland and Antarctic ice sheets.

System Model of Intermediate Complexity UVic 2.8. All other parameters of the sea level rise component are as in SURFER v2.0 and are recapped in tables 4 and 5.

### 2.4.2 Initial conditions

As in SURFER v2.0, the initial mass of carbon in atmospheric $CO_2$, $M_{\mathrm{A}}(t_{\mathrm{PI}})$, is set such as to have a preindustrial atmospheric
$CO_2$ concentration of 280 ppm. We have

$$M_{\mathrm{A}}(t_{\mathrm{PI}}) = 280 \times 10^{-18} m_{\mathrm{A}} \bar{m}_{\mathrm{C}} = 580.27\,\mathrm{PgC}. \tag{106}$$

The initial mass of carbon in atmospheric $CH_4$, $M_{\mathrm{A}}^{\mathrm{CH}_4}(t_{\mathrm{PI}})$, is set such as to have a preindustrial atmospheric $CH_4$ concentration of 720 ppb. We have

$$M_{\mathrm{A}}^{\mathrm{CH}_4}(t_{\mathrm{PI}}) = 720 \times 10^{-21} m_{\mathrm{A}} \bar{m}_{\mathrm{C}} = 1.49\,\mathrm{PgC}. \tag{107}$$





|  | Upper layer (0-150 m) | Intermediate layer (150-650 m) | Deep layer (650-3800 m) |
|---|---|---|---|
| Preindustrial DIC ($\mu$mol kg$^{-1}$) | 2017.65 | 2152.62 | 2266.57 |
| Total alkalinity ($\mu$mol kg$^{-1}$) | 2310.61 | 2310.60 | 2367.21 |
| Temperature (°C/ °K) | 16.34 / 289.49 | 8.95 / 282.10 | 2.65 / 275.80 |
| Salinity (psu) | 34.93 | 34.77 | 34.70 |

**Table 6.** GLODAPv2.2016b quantities averaged over ocean layers equivalent to those of SURFER.

The initial mass of carbon in land soils and vegetation $M_{\mathrm{L}}(t_{\mathrm{PI}})$ is set to 2200 PgC, as in SURFER v2.0. Hence, we have $M_{\mathrm{L}}^*(t_{\mathrm{PI}}) = M_{\mathrm{L}}(t_{\mathrm{PI}}) = 2200$ PgC. The initial mass of carbon in erodible CaCO$_3$ sediments, $M_{\mathrm{S}}(t_{\mathrm{PI}})$, is set to 1600 PgC, following Archer et al. (1998).

   For each ocean layer, we have 17 quantities ($T$, $S$, $K_0$, $K_1$, $K_2$, $K_{\mathrm{w}}$, $K_{\mathrm{b}}$, [H$^+$], [H$_2$CO$_3^*$], [HCO$_3^-$], [CO$_3^{2-}$], [OH$^-$], [H$_3$BO$_3$], [H(BO)$_4^-$], DIC, Alk, $p_{\mathrm{CO}_2}$) that are linked by a nonlinear system of 13 equations (equations 11, 13, 17-22, B1,
B2-B5 (+B7)). Hence, only $17-13 = 4$ of these quantities may be set independently. Equation B7 for the pressure dependence of the dissociation constants isn't counted because it can be combined with equations B2-B5. For each ocean layer, we will set initial temperature, salinity, alkalinity and DIC, except for the surface layer where we set [H$_2$CO$_3^*$] instead of DIC. This is because equilibrium conditions give a constraint on the H$_2$CO$_3^*$ mass (and thus [H$_2$CO$_3^*$]) in the upper layer. Equation 90 gives us

$$M_{\mathrm{U}}'(t_{\mathrm{PI}}) = \frac{W_{\mathrm{U}}K_0}{m_{\mathrm{A}}}M_{\mathrm{A}}(t_{\mathrm{PI}}) + \frac{W_{\mathrm{U}}}{\bar{k}_{\mathrm{A}\rightarrow\mathrm{U}}m_{\mathrm{A}}}(F_{\mathrm{CaSiO}_3,0} + F_{\mathrm{CaCO}_30}). \tag{108}$$

We obtain $M_{\mathrm{U}}'(t_{\mathrm{PI}}) = 6.94$ PgC and [H$_2$CO$_3^*$]$_{\mathrm{U}}(t_{\mathrm{PI}}) = M_{\mathrm{U}}'(t_{\mathrm{PI}})/(W_{\mathrm{U}}\bar{m}_{\mathrm{C}})\times 10^{18} = 10.43$ $\mu$mol kg$^{-1}$. To set the other quantities, we use the GLODAPv2.2016b mapped climatologies (Lauvset et al., 2016), which include climatologies for temperature, salinity, alkalinity, dissolved inorganic carbon, preindustrial dissolved inorganic carbon and pH, among other biogeochemical variables, which were computed based on data gathered between 1972 and 2013. The dissolved inorganic carbon data is nor-
malized to 2002, and the pH is computed based on temperature, alkalinity and the normalised DIC. We compute the global averages of these data fields over our defined ocean layers by the same averaging method as used for computing the thermal expansion coefficients (see section 2.4.1). The values obtained are in table 6.

   We set the initial (and constant) salinities $S_{\mathrm{U}}$, $S_{\mathrm{I}}$, $S_{\mathrm{D}}$ of our ocean layers to the computed averages from the GLODAP data. The temperatures of the ocean layers are defined as

$$T_i(t) = T_{i,0} + \delta T_i(t) \qquad \text{for } i \in \{\mathrm{U}, \mathrm{I}, \mathrm{D}\}. \tag{109}$$

By definition, the initial conditions for the temperature anomalies $\delta T_i$ are zero. We set $T_{i,0}$ such that $T_i(t = 2002)$, obtained from experimental runs, is approximately equal to the temperature average computed from the GLODAP data. We set $\tilde{Q}_{\mathrm{U}}(t_{\mathrm{PI}})$, $\tilde{Q}_{\mathrm{I}}(t_{\mathrm{PI}})$, $\tilde{Q}_{\mathrm{D}}(t_{\mathrm{PI}})$, $M_{\mathrm{I}}(t_{\mathrm{PI}})$, and $M_{\mathrm{D}}(t_{\mathrm{PI}})$ based on the computed averages for DIC and Alk, converted to carbon masses with equations 14 and 15. $M_{\mathrm{U}}(t_{\mathrm{PI}})$ is computed the from the fixed [H$_2$CO$_3^*$]$_{\mathrm{U}}(t_{\mathrm{PI}})$, $S_{\mathrm{U}}$, $T_{\mathrm{U},0}$, Alk$_{\mathrm{U}}(t_{\mathrm{PI}})$. Details for the computa-





| Variable | Comment | Initial (PI) Value |
|---|---|---|
| $M_A$ | mass of carbon in atmospheric $CO_2$ | 580.27 PgC |
| $M_A^{CH_4}$ | mass of carbon in atmospheric $CH_4$ | 1.49 PgC |
| $M_L$ | mass of carbon on land (soils+vegetation) | 2200 PgC |
| $M_L^*$ | additional land variable that integrates land-use emission | 2200 PgC |
| $M_U$ | dissolved inorganic carbon mass in ocean layer U | 1344.78 PgC |
| $M_I$ | dissolved inorganic carbon mass in ocean layer I | 4772.02 PgC |
| $M_D$ | dissolved inorganic carbon mass in ocean layer D | 31655.16 PgC |
| $Q_U$ | alkalinity mass in ocean layer U | 1536.67 PgC |
| $Q_I$ | alkalinity mass in ocean layer I | 5122.24 PgC |
| $Q_D$ | alkalinity mass in ocean layer D | 33060.77 PgC |
| $M_S$ | erodible $CaCO_3$ sediments mass | 1600 PgC |
| $\delta T_U$ | temperature anomaly in ocean layer U | 0 K |
| $\delta T_I$ | temperature anomaly in ocean layer I | 0 K |
| $\delta T_D$ | temperature anomaly in ocean layer D | 0 K |
| $S_{gl}$ | sea level rise contribution from mountain glaciers | 0 msle |
| $V_{GIS}$ | volume fraction of Greenland ice sheet with respect to preindustrial value | 1 |
| $V_{AIS}$ | volume fraction of Antarctic ice sheet with respect to preindustrial value | 1 |
| $S_U$ | salinity of ocean layer U (constant) | 34.93 psu |
| $S_I$ | salinity of ocean layer I (constant) | 34.77 psu |
| $S_D$ | salinity of ocean layer D (constant) | 34.70 psu |
| $T_{U,0}$ | preindustrial temperature of ocean layer U (constant) | 288.38 K |
| $T_{I,0}$ | preindustrial temperature of ocean layer I (constant) | 281.75 K |
| $T_{D,0}$ | preindustrial temperature of ocean layer D (constant) | 275.76 K |

**Table 7.** Initial conditions for SURFER v3.0. The upper part of the table correspond to the model 17 variables. The lower part of the table fixes salinity and preindustrial temperature, this is necessary to compute the dissociation constants and the solubility constant of $CO_2$.

tion are provided in appendix C. We obtain $M_U(t_{PI}) = 1344.78\,\mathrm{PgC}$, which corresponds to $\mathrm{DIC}_U(t_{PI}) = 2022.08\,\mathrm{\mu mol\,kg^{-1}}$. This is only 0.22% off compared to the averaged value for the upper layer obtained from GLODAP. The total dissolved inorganic carbon in the ocean is 37772 PgC, which is close to the 38000 PgC estimate from the IPCC (Canadell et al., 2021).

For the sea level rise components, as in SURFER v2.0, we set $S_{gl}(t_{PI}) = 0$, $V_{GIS}(t_{PI}) = 1$, and $V_{AIS}(t_{PI}) = 1$. All initial conditions a recapped in table 7.

In Figure 3, we compare the horizontally averaged vertical depth profiles of GLODAP to the vertical profiles of SURFER v3.0 for different model quantities. The vertical profiles of SURFER are computed by running the model from 1750 to 2002,



forced with historical CH$_4$ and CO$_2$ emissions, and starting from the initial conditions described above. We observe that the chosen initial conditions produce a model state in 2002 that matches the GLODAP data.

## 2.5 Numerics

The model is implemented in Python 3.0. using the library solve_ivp with the integration method LSODA. The LSODA method has an automatic stiffness detection and switches accordingly between an Adams and BDF method (Petzold, 1983). The local error estimates are kept below $atol + rtol \times abs(y)$ where $atol$ and $rtol$ are parameters that control the relative and absolute accuracy and where $y$ is a model variable. By default, we set $atol$ to $10^{-3}$ for the variables $M_A^{CH_4}$ $M_S$, $\delta T_U$, $\delta T_I$, $\delta T_D$, $S_{gl}$, $V_{GIS}$, $V_{AIS}$ , and we set $atol$ to $10^{-6}$ for the other variables. The reason for this difference is that the variables in the first

group can have small or near zero values, meaning that $atol$ will dominate the local error estimate. If it is too small, the solver takes too many steps and is slow. We set $rtol$ to $10^{-6}$ for all variables. The code is compiled with Numba and the model runs fast. When forced with CO$_2$ and CH$_4$ emissions of a given SSP scenario, runs of $10^3$ to $10^6$ years take typically around 60 milliseconds on a laptop with processor Intel® Core™ i5-10210U CPU @ 1.60GHz × 8. The run time is not a linear function of simulated time because the LSODA method uses an adaptive time step.

## 3 Numerical results and comparisons

In this following section, we test SURFER v3.0 and show that it is an adequate representation of the real climate system. We show that it reproduces well-known dynamics of the carbon cycle and we compare it with outputs of other models over a large range of time scales.

### 3.1 Historical period

We show here that SURFER v3.0 is able to reproduce the measured historical CO$_2$ and CH$_4$ concentrations, and the estimated land and ocean carbon sinks. We perform a historical run by starting SURFER in 1750 with the parameters and initial conditions described in section 2.4. We force the model with fossil and land-use emissions of CO$_2$ and CH$_4$. Emissions from other greenhouse gases such as nitrous oxide (N$_2$O), ozone (O$_3$), halogenated gases and aerosols are not taken into account. Figures 4 and 5 show the historical CO$_2$ and CH$_4$ concentrations as simulated by SURFER v3.0 compared to the measurements from

Köhler et al. (2017). SURFER v3.0 is in good agreement with the historical CO$_2$ observations, with a difference of at most ~6 ppm, which is better than SURFER v2.0. For methane concentrations, SURFERv3.0 is in relatively good agreement with the historical observations, although it doesn't capture well the apparent stabilisation in the 2000s. The cause for this stabilisation is not totally clear (Turner et al., 2019). Main hypothesises include a decline in fossil emissions (Chandra et al., 2024), and a shortening of the lifetime of atmospheric CH$_4$, due to increasing concentrations of the hydroxyl radical caused by changes in

emissions of other gases such as N$_2$O and carbon monoxide (CO) (Skeie et al., 2023). These processes are not modelled in SURFER v3.0, where the atmospheric lifetime of CH$_4$ is kept constant.




In Figure 6, we compare the partitioning of $CO_2$ emissions in the atmosphere, land and ocean reservoirs with the estimates from the Global Carbon Budget (GCB) (Friedlingstein et al., 2022). Fossil and land-use $CO_2$ emissions used in SURFER v3.0 up to the year 1990 are the estimates provided by the GCB, and and after that, we start using the emission values provided
for the SSP scenarios. These are slightly different than the estimates from the GCB (see appendix A), which explains the small mismatch visible in Figure 6. In SURFER, we compute the ocean sink as $S_{\text{ocean}} = F_{A \to U} - F_{A \to U}(t_{PI})$, following the definition of Hauck et al. (2020), the land sink as $S_{\text{land}} = F_{A \to L} - F_{A \to L}(t_{PI}) = F_{A \to L}$, and the atmospheric growth as $\frac{dM_A}{dt}$. These quantities simulated by SURFER v3.0 for the historical period are very close to the estimates from the GCB. For the years 2000 to 2010, the GCB gives a mean estimate of $2.3 \pm 0.4 \, \text{PgC yr}^{-1}$ for the ocean sink, $2.7 \pm 0.5 \, \text{PgC yr}^{-1}$ for the land
sink, and $4 \pm 0.02 \, \text{PgC yr}^{-1}$ of atmospheric growth. Values simulated in SURFER v3.0 are respectively $2.16 \, \text{PgC yr}^{-1}$, $3.05 \, \text{PgC yr}^{-1}$ and $3.96 \, \text{PgC yr}^{-1}$, with only the atmospheric growth being just below the GCB estimated range. The cumulative budgets are also very similar: of the total amount of emissions in the period 1850-2014, the GCB estimates that around $26\pm 5\%$ are absorbed by the ocean, $31\pm7\%$ are absorbed by the land and $40\pm1\%$ stay in the atmosphere, while for SURFER v3.0, those numbers are respectively 24%, 37% and 41%. In the GCB, there is a cumulative budget imbalance of 15 PgC for the
years 1850-2014, which arises from errors in independent estimates of emissions and sinks, as well as from missing terms in the budget computation. In SURFER v3.0, however, carbon is explicitly conserved and the budget imbalance ($B_{\text{im}}$) only results from the definition of the sinks, which don't capture processes such as methane oxidation or changes in carbonate and silicate weathering fluxes. Indeed, we have

$$B_{\text{im}} = E_{\text{fossil}}^{CO_2} + E_{\text{land-use}}^{CO_2} - \left( \frac{dM_A}{dt} + S_{\text{ocean}} + S_{\text{land}} \right) \tag{110}$$

$$= -V + F_{A \to U}(t_{PI}) + F_{\text{weathering}} - \left( F_{CH_4,\text{ox}} - E_{\text{natural}}^{CH_4} \right) \tag{111}$$

$$= (F_{\text{weathering}} - F_{\text{weathering}}(t_{PI})) - \left( F_{CH_4,\text{ox}} - E_{\text{natural}}^{CH_4} \right), \tag{112}$$

and the cumulative budget imbalance for the years 1850-2014 is -15 PgC, with a contribution of +1 PgC from increased weathering fluxes and -16 PgC from methane oxidation.

## 3.2   CMIP6 projections

We now compare SURFER v3.0 to the CMIP6 ensemble for the SSP1-2.6 and SSP3-7.0 scenarios. As for the historical runs in section 3.1, SURFER v3.0 is forced with $CO_2$ and $CH_4$ fossil and land-use emissions, but no other greenhouse gases nor aerosols. Runs are started in 1750 and results for atmospheric $CO_2$, temperature, surface ocean pH, ocean carbon uptake and land carbon uptake are plotted in Figure 7. Additionally, we compare with outputs from SURFER v2.0, forced with the total $CO_2$ emissions and run with the parameters and initial conditions described in Martínez Montero et al. (2022).

As already shown in Figure 4, SURFER v3.0 reproduces well the historical $CO_2$, and for the SPP scenario projections, falls within the lower range of the CMIP6 model ensemble. SURFER v3.0 can simulate a global mean temperature anomaly that generally remains within the CMIP6 range, considering only the effects of $CO_2$ and methane. This is because the contributions from the other major drivers of temperature changes such as nitrous oxide ($N_2O$), ozone ($O_3$), halogenated gases and aerosols





approximately cancel each other (Forster et al., 2021, see Figure 7.8). However, between 1960 and 2015, the cooling effect of
aerosols from anthropogenic and volcanic sources was likely more significant, and without accounting for this, SURFER v3.0
simulates temperatures slightly above the CMIP6 range.

Surface pH as simulated by SURFER v3.0 generally aligns with the CMIP6 range for both the historical period and SSP
projections, an improvement over SURFER v2.0 which showed too rapid ocean acidification. This improvement is primarily
due to the addition of a new intermediate layer in SURFER v3.0, which facilitates faster carbon transfer out of the upper ocean
layer, thereby slowing surface acidification. This enhanced carbon transfer to intermediate and deep ocean layers also allows
the ocean to absorb $CO_2$ more efficiently. As a result, the ocean carbon uptake in SURFER v3.0 now falls within the CMIP6
model range. The land carbon uptake in SURFER v3.0 and v2.0 are very similar, and both are in the range of CMIP6 models,
which is quite large and demonstrates a higher uncertainty.

In Figure 8, we compare the land and ocean sinks of SURFER to four CMIP6 models and one EMIC that have been run to
the year 2300 under the SSP1-2.6, SSP3-4.3, and SSP5-8.5 scenarios. We observe that SURFERv3.0 remains within the range
of CMIP6-class models even for these longer time scales. For all three scenarios, the land sink is expected to become negative
at some point, indicating that the land reservoir will release some of the carbon it had previously absorbed (Canadell et al.,
2021; Tokarska et al., 2016; Zickfeld et al., 2013). For the SSP1-2.6 and SSP-3.4 scenarios, this negative land sink in CMIP6
models is attributed to the land carbon-concentration feedback: as $CO_2$ concentrations decrease after strong negative emissions,
vegetation releases carbon. For the SSP5-8.5 scenario, the negative land sink is rather due to a stronger land carbon-climate
feedback, where warming leads to a release $CO_2$ from the land reservoir, for example through increased decomposition rates
(Tokarska et al., 2016). In SURFER, the parametrisation of the atmosphere-to-land flux, ($F_{A \to L}$) depends on the atmospheric
$CO_2$ concentration ($M_A$) but not on temperature, effectively including only a carbon-concentration feedback. This explains
why, for the SSP5-8.5 scenario, the land sink in SURFER only becomes slightly negative around 2250, when the atmospheric
$CO_2$ concentrations begin to decline. Despite this, the land sink from SURFER remains mostly in the range of the other models,
which is quite large and reflects the large uncertainty in processes related to the terrestrial biosphere.

### 3.3 LTMIP

In the previous sections, we have seen that SURFER v3.0 can reproduce the historical record and outputs from CMIP6-class
models for projections up to 2300. Here, we focus on longer time scales and compare SURFER v3.0 with results from the
LTMIP (Long Tail Model Intercomparison Project, Archer et al., 2009). In these experiments, several models were used to
assess the $CO_2$ draw-down from the atmosphere for 10000 years, after emissions pulses of 1000 PgC and 5000 PgC. For each
emission pulse, 5 experiments are performed with different physical processes progressively included to assess their impact on
atmospheric $CO_2$ uptake. These experiences are named with a combination of letters indicating the processes included: climate
feedbacks (C), sediments (S), weathering (W), and vegetation (V). We reproduce these experiments with SURFER v3.0 by
successively reducing the number of active processes. For, the CSWV experiment, we use the standard version of SURFER
v3.0. For the CSW experiment, we set $k_{A \to L} = 0$, so that vegetation is kept constant and has no influence on carbon uptake
($F_{A \to L} = 0$). For the CS experiment, we additionally keep the weathering fluxes $F_{CaCO_3}$ and $F_{CaSiO_3}$ constant and equal





to their preindustrial values, thus eliminating weathering feedbacks. For the C experiment, we further keep the accumulation and burial fluxes constant and equal to their preindustrial values, thus effectively eliminating interactions with the sediments. Finally for the baseline experiment, on top of all the modifications described above, we keep the solubility and dissociation constants constant. In this last case, SURFER v3.0 is very similar to SURFER v2.0.

Results for these five experiments are visible in Figure 9 for the 1000 PgC pulse, and in Figure 10 for the 5000 PgC pulse. Overall, SURFER v3.0 falls within the range of other models, except in the following cases: the 5000 PgC baseline experiment after 1000 years, the 5000 PgC C experiment between years 1000 and 5000, and the CSWV experiments after year 1000, where SURFER v3.0 simulates slightly lower atmospheric CO2 levels than the other models. For the baseline experiment, SURFER v2.0 doesn't absorb $CO_2$ from the atmosphere fast enough in the first thousand years after the emission pulse. As already mentioned, this is improved in SURFER v3.0 thanks to the addition of a third oceanic layer at intermediate depth.

We can define and quantify the climate, sediment, weathering, and vegetation feedbacks by taking the difference in simulated atmospheric $CO_2$ between consecutive experiments (C-baseline, CS-C, CSW-CS, and CSWV-CSW). Results are plotted in Figure 11 for the 1000 PgC pulse, and in Figure 12 for the 5000 PgC pulse. Not all experiments were performed for each model so feedbacks can't always be computed. All experiments are only available for CLIMBER and SURFER v3.0. Overall, the feedbacks in SURFER v3.0 fall within the range of the other models, demonstrating that the associated processes are reasonably well simulated by SURFER. For 5000 PgC pulse experiments, the climate feedback in SURFER v3.0 is very similar to the LOSCAR and GEOCYC models but quite different to the other models. This is probably explained by SURFER V3.0, as well as LOSCAR and GEOCYC, all missing a dynamic ocean circulation and hence feedbacks associated with temperature-induced circulation changes. The sediment feedback in SURFER v3.0 for the 1000 PgC is in the higher range (more negative) of the other models, which is consistent with the dissolution flux being generally larger (accumulation more negative) than in the other models (see Figure 13). For the 5000 PgC, the sediment feedback in SURFER v3.0 is in the mid-to-lower range of the other models, despite the dissolution flux still being in the higher range. In general, other than oceanic invasion, vegetation has the biggest impact on $CO_2$ uptake before the year 1000, while sediments have the biggest impact between the year 1000 and 10000.

### 3.4 cGENIE

Only a few models of intermediate complexity have been run for 100 kyr or more to investigate the carbon cycle's response to (anthropogenic) $CO_2$ emissions. Some examples include cGENIE (Colbourn et al., 2013, 2015; Lord et al., 2016), and CLIMBER-X (Kaufhold et al., 2024). Here, we compare SURFER v3.0 with the 1 Myr runs performed with the cGENIE model of intermediate complexity (Lord et al., 2016). This model comprises a 2-D Energy-Moisture Balance atmosphere, a 3-D frictional geostrophic ocean circulation model, and a representation of the global carbon cycle, with ocean cycling of DIC, alkalinity, and a nutrient ($PO_4$), $CaCO_3$ marine sediments, and terrestrial weathering (Edwards and Marsh, 2005; Ridgwell et al., 2007; Ridgwell and Hargreaves, 2007; Colbourn et al., 2013). For the runs presented here (Lord et al., 2016), cGENIE was used without the terrestrial biosphere module and its associated carbon fluxes. The model had 8 ocean levels, with the surface layer being 175 meters deep, comparable to the surface layer in SURFER v3.0.



We perform equivalent runs in SURFER v3.0 with $k_{A\rightarrow L} = 0$ to neglect the role of vegetation. Results are plotted in figures 14 and 15 for atmospheric $CO_2$, global mean temperature, ocean surface pH, ocean surface calcite saturation state and $CaCO_3$ content in sediments. Ocean surface calcite saturation state, $\Omega_U$, is defined as

$$\Omega_U = \frac{[CO_3^{2-}]_U [Ca^{2+}]_U}{K_{sp}^{CaCO_3}} . \tag{113}$$

The solubility product $K_{sp}^{CaCO_3} = [CO_3^{2-}]_{sat}[Ca^{2+}]_{sat}$ depends on salinity , temperature and pressure. For the computation of $\Omega_U$, we use parametrisations of $K_{sp}^{CaCO_3}$ described in appendix B, and we assume that $[Ca^{2+}]_U$ is constant and equal to $0.01028 \, \mathrm{mol \, kg^{-1}}$.

Overall, SURFER v3.0 reproduces well the behaviour of cGENIE. For all emission pulses, the relative difference in simulated atmospheric $CO_2$ with cGENIE doesn't exceed 18%, is lower than 8% after 1000 yrs and below 5% after 50 kyr. These are smaller differences than those between models for the LTMIP experiments (see Figures 9 and 10). This good agreement for millenial and longer time scales was expected, as SURFER v3.0 was qualitatively tuned to cGENIE's long-term atmospheric $CO_2$ output. The agreement in ocean surface pH is also very strong, with absolute differences below 0.06 pH units after 5 years, below 0.04 pH units after 1000 years, and below 0.02 pH units after 50 kyr, corresponding to relative difference below 1% after 5 years for all emissions pulses. Because the carbon exchanges between the atmosphere and the surface ocean reach equilibrium relatively fast, the good agreement for ocean surface pH directly results from the good agreement for atmospheric $CO_2$ concentrations. Regarding temperatures, peak warming occurs later in SURFER v3.0 than in cGENIE, primarily because SURFER models only ocean temperatures, leading to slower global warming compared to cGENIE, which also accounts for the thermal balance of the continents.

After 1 Myr, the state is almost back to equilibrium in SURFER v3.0, with atmospheric $CO_2$ concentrations ranging from $280.68$ ppm for the $1000$ PgC emission pulse, to $292.08$ ppm for the $20000$ PgC pulse. Carbon is removed from the atmosphere through a range of processes. First, atmospheric $CO_2$ dissolves in the upper ocean layer following the reaction

$$CO_2 + H_2O + CO_3^{2-} \rightleftharpoons 2HCO_3^- , \tag{R9}$$

which is equivalent to reactions R1-R3. This causes a decrease in ocean surface pH (acidification) and consumes $CO_3^{2-}$, which results in a decrease of the surface calcite saturation state. Both these effects are observable in SURFER v3.0 and cGENIE. On centennial to millennial timescales, the $CO_3^{2-}$ anomaly mixes into the ocean interior and deep waters become less saturated, causing an increase in the dissolution of deep-sea $CaCO_3$ sediments and a release of carbonate ions. Some of these carbonate ions can then react with $CO_2$ (reaction R9), leading to further oceanic $CO_2$ uptake. This process is called sea-floor neutralisation, as it is the dissolution of previously deposited deep-sea sediments that allows the neutralisation of atmospheric $CO_2$ (Archer et al., 1997, 1998). Moreover, the increased dissolution of $CaCO_3$ sediments compared to the preindustrial state creates an imbalance between the alkalinity input to the ocean by weathering and the alkalinity output by accumulation. This replenishes the ocean $CO_3^{2-}$ concentration and the erodible $CaCO_3$ sediments stock while leading to a further uptake of atmospheric $CO_2$. This second process is called terrestrial neutralisation, as it is the (imbalanced) dissolution of carbonate and silicate rocks on land that neutralise the atmospheric $CO_2$ (Archer et al., 1997, 1998; Ridgwell and Hargreaves, 2007).





We observe an overshoot in surface calcite saturation state and deep sea ocean sediment content compared to the preindustrial situation in both SURFER v3.0 and cGENIE because of increased weathering rates due to warming. The extra $CaCO_3$ in the sediments will eventually be buried, leading to a permanent transfer of carbon to the geological reservoir.

## 4 Sea level rise and importance of long time scale processes

So far, we have focused primarily on the carbon cycle, as the additions to SURFER v3.0 are related to it. In this section,
we examine sea-level rise (SLR), which is computed as the sum of four contributions: thermosteric (thermal expansion), glaciers, Greenland and Antarctica. The parametrisation for thermosteric sea level rise in SURFER v3.0 is essentially the same as in SURFERv2.0, with the key differences being the use of three 3 ocean layers instead of two, and the use of new thermal expansion coefficients. In subsection 4.1, we verify that these changes still provide a reasonable approximation of thermosteric SLR. The parameterisations for glaciers, Greenland and Antarctica remain unchanged between SURFER v2.0
and SURFER v3.0, so any difference in SLR contributions under a given forcing scenario results only from differences in simulated temperatures. We investigate the SLR contribution from the ice sheets in subsection 4.2.

### 4.1 Thermosteric sea level rise and ocean heat content

Heat transfer in SURFER v3.0's ocean is controlled by the parameters $\gamma_{U \rightarrow I}$ and $\gamma_{I \rightarrow D}$. The heat accumulated in ocean layers, along with the corresponding temperature increases, determine thermosteric sea level rise. To ensure that our chosen values for
$\gamma_{U \rightarrow I}$, $\gamma_{I \rightarrow D}$, and for the thermal expansion coefficients are reasonable, we compare the ocean heat content and thermosteric sea level rise in SURFER v3.0 with IPCC estimates for 1971-2018 (Figure 16).

SURFER v3.0 simulates significantly higher ocean heat content above 700 m (layers U+I) than the IPCC estimates, resulting in a larger thermosteric sea level rise. This discrepancy arises for two reasons. First, in SURFER, all energy imbalance is absorbed by the ocean, with no energy allocated to warming the land and atmosphere or for melting glaciers and ice sheets.
Consequently, the ocean warms more than it should. Second, and more importantly, SURFER v3.0 doesn't account for faster and larger land temperature increases, which causes the global mean temperature rise to be higher when averaged over land and oceans. In other words, SURFER assumes that the global mean temperature is equivalent to the ocean's mean surface temperature. As a result, SURFER needs more energy to reproduce observed global mean temperatures, and overestimates surface ocean temperatures. For the ocean below 700 m (layer D in SURFER), the ocean heat content and the thermosteric sea
level rise simulated by SURFER v3.0 matches the IPCC estimates quite well. This indicates that the oceanic heat transport at depth is a little too slow, which compensates for the ocean receiving more energy than it should.

We also check that our parametrisation for thermosteric sea level rise is valid on longer time scales. In Figure 17, we compare outputs from the intermediate complexity model UVic, versions 2.8 and 2.9 (Eby et al., 2009; Clark et al., 2016), to outputs from SURFER v2.0 and SURFER v3.0. The UVic 2.8 and UVic 2.9 models both include a sediment module and have
equilibrium climate sensitivities around 3.5°C, the same as in SURFER. Emission scenarios used to force the models follow



historical estimates of $CO_2$ emissions up to the year 2000. Following this, cumulative emissions of either 1280 or 3840 PgC are added between the years 2000 and 2300. For more details on the experimental setup, see Clark et al. (2016).

SURFER v3.0 has a faster and larger atmospheric $CO_2$ uptake than both UVic 2.8 and UVic 2.9. This is consistent with the LTMIP experiments (figures 9 and 10) where UVic 2.8 was already the model with the smallest $CO_2$ uptake after 10000 years
for the CSWV experiments. UVic 2.9 has an even slower $CO_2$ uptake than UVic 2.8 due to the difference in the sediments representation. For SURFER v2.0, the atmospheric $CO_2$ concentration reaches equilibrium after ∼4000 years since it only takes into account the process of ocean $CO_2$ invasion. These differences in atmospheric $CO_2$ concentrations lead to relatively large differences in global mean surface temperatures. Nevertheless, the thermosteric SLR is comparable in all models, except for UVic 2.9 under the 3840 PgC scenario, where it is larger than for the other models. The thermosteric SLR from SURFER
v3.0 is close to the one from UVic 2.8 for both scenarios, even though the simulated temperature is lower in SURFER v3.0. This is again a consequence of SURFER not simulating land temperatures. The global mean ocean temperature increase in UVic 2.8 is smaller than its global mean temperature increase, and probably comparable to the mean ocean temperature observed in SURFER v3.0. SURFER v3.0 and SURFER v2.0 also have comparable thermosteric SLR despite SURFER v2.0 simulating a larger temperature increase. This is because the ocean in SURFER v3.0 is deeper than in SURFER v2.0 (3800 m deep vs 3150
m) and has thus more potential for expansion.

## 4.2 Ice sheets

In SURFER v2.0, atmospheric $CO_2$ concentrations, and hence temperatures, stabilise a few thousand years after the end of emissions. In SURFER v3.0, thanks to new processes added in the carbon cycle, $CO_2$ drawdown from the atmosphere continues until return to preindustrial conditions, leading to lower temperatures than in SURFER v2.0 on millennial and longer
time scales. Since ice sheets respond on the millennial time scale, we expect these differences to have a significant impact on their melting. To test this, we force both SURFER v3.0 and SURFER v2.0 with the $CO_2$ emissions from five SSP scenarios (SSP1-2.6, SSP2-4.5, SSP4-6.0, SSP3-7.0, SSP5-8.5) which cover a range of possibles futures. Simulations last for 500 kyr and results are plotted in Figure 18.

In SURFER, the ice sheets are designed as tipping elements (see section 2.3). For both SURFER v2.0 and SURFER v3.0,
and under all scenarios, the simulated temperature increase overshoots the critical warming threshold of the Greenland ice sheet. However, the Greenland ice sheet doesn't always collapse. Indeed, in general, tipping can be avoided if the overshoot duration is short relative to the effective time scale of the tipping element (Ritchie et al., 2019, 2021). For SURFER v2.0 this happens for the SSP1-2.6 scenario. For all other scenarios, the temperature stabilises past the critical threshold and thus Greenland eventually transitions to a completely melted state. In SURFER v3.0, thanks to a greater decrease in temperature
after reaching a maximum, Greenland overshoots safely its critical threshold also under SSP2-4.5 and SSP4-6.0 scenarios. This happens even though peak warming reaches 2.62 °C and 3.18 °C respectively, well above the Greenland ice sheet's critical threshold of 1.52 °C (as set in SURFER). For these scenarios, This leads to a ∼6 m reduction in the long-term sea level rise contribution simulated by SURFER v3.0 compared to SURFER v2.0.





Meanwhile, The Antarctic Ice Sheet doesn't tip in our simulations, regardless of using SURFER v2.0 or SURFER v3.0.

This is because the critical warming threshold for the Antarctic ice sheet, set to 6.8 °C in SURFER, is much higher than that for Greenland and seldom reached, even for the SSP5-8.5 scenario. However, despite no changes in the tipping behaviour, differences in simulated temperatures still give rise to large differences in SLR contribution, particularly for high-emission scenarios. The long-term sea level rise contributions from Antarctica for the SSP3-7.0 and SSP5-8.5 scenarios are respectively reduced by $\sim$8 m and $\sim$18 m in SURFER v3.0 compared to SURFERv2.0.

**5 Discussion**

Our main goal with SURFER v3.0 was to include processes to model the carbon cycle dynamics on multi-millennial time scales. To this end, we have added to SURFER v2.0 a dynamic and more precise representation of alkalinity, an explicit representation of the carbonate and soft-tissue pumps, a sediments reservoir with associated accumulation and burial fluxes, and weathering as well as volcanic out-gassing fluxes. We have shown that these additions allow for an accurate simulation of carbon cycle

dynamics on multi-millennial time scales by comparing SURFERv3.0 to the outputs from the LTMIP experiments, and to the outputs of cGENIE for 1 Myr runs. Furthermore, we showed that the stabilisation of atmospheric $CO_2$ and temperature at lower levels in SURFER v3.0 than in SURFER v2.0 leads to a significant reduction in simulated sea level rise. Particularly noteworthy is the substantial reduction in contributions from Greenland for the intermediate scenarios (SSP2-4.5 and SSP4-6.0), which align with the pathways we are currently following.

A secondary goal with SURFER v3.0 was to improve on SURFER v2.0 for the decadal to centennial time scales. This was done by adding an ocean layer of intermediate depth, a temperature and pressure dependence of the solubility and dissociation constants, a representation of atmospheric methane, and by carefully setting the initial conditions of the oceanic variables based on the GLODAP dataset. We have shown that SURFER v3.0 successfully reproduces the historical $CO_2$ and $CH_4$ concentrations, the estimated historical land and ocean sinks, the CMIP6 ensemble mean for the evolution of different quantities

under the SSP1-2.6 and SSP3-7.0 scenarios, and that in all these tasks it performs equally well or better than SURFER v2.0.

In summary, SURFER v3.0 outperforms SURFER v2.0 on all time scales, and despite having doubled the number of differential equations, it stays fast, transparent and easy to modify. This manuscript contains all the equations for the model, all the parameter values, and the initial conditions for all the variables. SURFER v3.0 is coded in Python, which is one of the most widely used open-source programming languages. The code of the model, as well as the code for all the figures, is available

online in a Jupyter notebook, providing already a wide range of example use cases.

Of course, many coupled atmosphere-ocean box models of the carbon cycle already exist, some of them including carbonate sediments and weathering processes (Keir, 1988; Munhoven and François, 1996; Lenton and Britton, 2006; Zeebe, 2012; Köhler and Munhoven, 2020). These models, often primarily focused on the ocean, offer a more complex representation of the carbon cycle than SURFER v3.0. For instance, BICYCLE-SE (Köhler and Munhoven, 2020) includes 1 atmospheric box, 10

ocean boxes (5 surface boxes), ocean cycling of DIC, alkalinity, dissolved oxygen, and phosphate, sediment columns in each




ocean basin and at various depths, and 7 terrestrial biosphere boxes. Still, despite being simpler, SURFER v3.0 effectively captures the essential dynamics of the carbon cycle, as demonstrated in the precedent sections.

Moreover, SURFER includes a dynamic representation of temperature and sea level rise, which is often lacking in other models where those quantities are constant or prescribed. Among existing models, the one from Lenton and Britton (2006)
is the closest to SURFER v3.0, with 4 ocean boxes, 10 sediment boxes, 2 terrestrial biosphere boxes, and a weathering flux parameterisation. As in SURFER v3.0, its carbon cycle is coupled with an energy balance representation of global mean temperature and both models can be classified as "simple Earth system models". While it is possible to integrate SURFER's temperature equations and sea level rise parameterisation into other models, we believe the unique combination of included processes, simplicity, transparency, and speed makes SURFERv3.0 a valuable addition to the literature, on top of being a
worthwhile update to SURFER v2.0.

Although we are quite satisfied with SURFER's capabilities, we are certainly not claiming it is *the one model to rule them all*. Indeed, being that simple comes with several limitations that we discuss here.

First, SURFER v3.0 lacks horizontal resolution. In particular, SURFER v3.0 has no high latitude ocean surface box, which is a feature of most ocean carbon cycle box models and could help with simulating more realistically the atmosphere-to-ocean
$CO_2$ flux. The lack of spatial resolution also implies that SURFER v3.0 doesn't represent land temperatures, hence equating global mean temperatures with ocean mean surface temperatures. As shown in section 4.1, this leads to an overestimation of ocean surface temperature and thermosteric SLR for historical and SSP forced runs.

Additionally, several oceanic and land carbon-climate feedbacks are missing in SURFER v3.0. For example, there is no dynamic ocean circulation and as such, changes in atmosphere-to-ocean $CO_2$ fluxes resulting from climate-induced changes in
the oceanic circulation are not represented. This was suggested in section 3.3 by comparing the "climate feedback" of SURFER v3.0 with other models for the 5000 PgC pulse LTMIP experiment. A second example is the organic matter and $CaCO_3$ productions in the upper ocean layer that are kept constant, neglecting eventual changes in marine ecosystem production and associated carbon uptake. Last but not least, is our parametrisation of atmosphere-to-land carbon flux, which depends on the atmospheric $CO_2$ concentration, but not on temperature. As explained in section 3.2, this is probably the reason why
SURFER's land sink stays positive longer than other models for high-emission scenarios. Furthermore, this parametrisation doesn't account for hypothesised tipping elements such as the Amazon and boreal forests or the permafrost, which may release important amounts of greenhouse gases past critical warming thresholds (Armstrong McKay et al., 2021).

For the sea level rise module, the contribution of the Antarctic ice sheet should probably be split into several components. Indeed, it has been shown that the West Antarctic ice sheet, as well as some East Antarctic subglacial basins, could collapse for
lower warming thresholds than the East Antarctic ice sheet and contribute to several meters of sea level rise already by 2300 for high emission scenarios (Garbe et al., 2020; Alley et al., 2015; Coulon et al., 2024).

Finally, with the weathering of silicate rocks, SURFFER v3.0 includes only one process that impacts the carbon cycle on the 100 kyr time scale, but there exist others. For example, organic carbon burial, which is presently neglected in the model, could also act as a long-term carbon sink. Another example is the changes in the astronomical forcing, which is one of the
main driver of glacial-interglacial cycles.





## 6 Conclusions

We have presented SURFER v3.0, a simple Earth system model that includes a dynamic carbon cycle and simulates various important quantities such as atmospheric $CO_2$ and $CH_4$ concentrations, temperature anomalies, ocean surface pH, and sea-level rise in response to anthropogenic greenhouse gases emissions. SURFER v3.0 extends SURFER v2.0 by incorporating

dynamic alkalinity cycling, $CaCO_3$ sediments, and weathering processes. These additions enable SURFER v3.0 to accurately simulate the dynamics of the coupled carbon-climate system over timescales ranging from decades to millions of years. We have validated this by comparing SURFER v3.0 to historical data and outputs from GCMs, EMICs, and other box models.

We have also demonstrated that SURFER v3.0 can simulate thermosteric sea-level rise reasonably well on millennial timescales, though it tends to overestimate it for shorter timescales. Furthermore, we have shown that incorporating long-

term carbon cycle processes in SURFER v3.0 leads to a significant reduction in simulated contributions of Greenland and Antarctica to sea-level rise, compared to SURFER v2.0. These results highlight the critical importance of considering these processes to better predict committed sea-level changes.

SURFER v3.0 is fast, transparent, easy to modify and use, and hence an ideal tool for policy assessments that wish to take into account centennial to multi-millennial time scales. In the future, we plan to add to SURFER a representation of

glacial cycle dynamics and include several tipping elements to investigate the stability of the Earth system and the impact of anthropogenic emissions on its future long-term trajectories.

*Code availability.*   The exact version of SURFER used to produce the results showed in this paper is archived on Zenodo (https://zenodo.org/records/12774163, Couplet et al., 2024), as is the input data to run the model and most of the data to produce the plots. The code of SURFER is licensed under MIT license.

*Data availability.*   Data from other references used in this paper for emission scenarios are as follows :

- Historical $CO_2$ emissions are from Friedlingstein et al. (2022) and are available at https://doi.org/10.18160/GCP-2022.

- Historical $CH_4$ emissions are from Jones et al. (2023) and are available at https://zenodo.org/records/10839859.

- $CO_2$ and $CH_4$ emissions for the SSP scenarios are available in the SSP database hosted by the IIASA Energy Program at https://tntcat.iiasa.ac.at/SspDb.

Data from other references used in this paper for comparison with SURFER's output are as follows :

- Figure 3 : GLODAPv2.2016b mapped climatologies (Lauvset et al., 2016) are available at https://www.ncei.noaa.gov/access/metadata/landing-page/bin/iso?id=gov.noaa.nodc:0286118.

- Figures 4 and 5 : data from Köhler et al. (2017) available at https://doi.org/10.1594/PANGAEA.871273.

- Figure 6 : data from Friedlingstein et al. (2022) available at https://doi.org/10.18160/GCP-2022.



– Figure 7 : CMIP6 data can be accessed through the Earth System Grid Federation (ESGF) nodes (e.g. https://esgf-index1.ceda.ac.uk/projects/cmip6-ceda/).

– Figure 8 : data is from Figure 5.30 of IPCC AR6 WG1 (Canadell et al., 2021). CMIP6 data can be accessed through the Earth System Grid Federation (ESGF) nodes (e.g. https://esgf-index1.ceda.ac.uk/projects/cmip6-ceda/). The code to process the data necessary for this specific IPCC figure is available online in a jupyter notebook at https://github.com/IPCC-WG1/Chapter-5_Fig30/blob/main/
longterm_carboncycle_withssp126.ipynb.

– Figures 9-13 : LTMIP data is available at https://terra.seos.uvic.ca/LTMIP/. The LOSCAR data is from Zeebe (2012) and available through personal correspondence with author.

– Figures 14 an 15 : cGENIE data is from Lord et al. (2016) and available through personal correspondence with author.

– Figure 16 : data is from Figure 1 in IPCC AR6 WG1 Cross-Chapter Box 9.1 (IPCC, 2021). The code to produce the IPCC figure
and the associated analysis is freely available online in a jupyter notebook : https://github.com/BrodiePearson/IPCC_AR6_Chapter9_Figures/blob/main/Plotting_code_and_data/Cross_Chapter_Box9_1/Plot_Figure/plot_AR6_CCBox9.1_FGD.ipynb.

– Figure 17 : data from Clark et al. (2016) available at https://www.nature.com/articles/nclimate2923.





**Figure 3.** Horizontally averaged quantities from the GLODAPv2.2016b mapped climatologies (Lauvset et al., 2016) compared to initial and simulated values by SURFER v3.0. The carbonate species are not provided in the GLODAP climatologies. We computed their values at each ocean point based on the climatologies of DIC, Alk, temperature and salinity, and then averaged them horizontally. Details for the computation of the carbonates species are in appendix C.





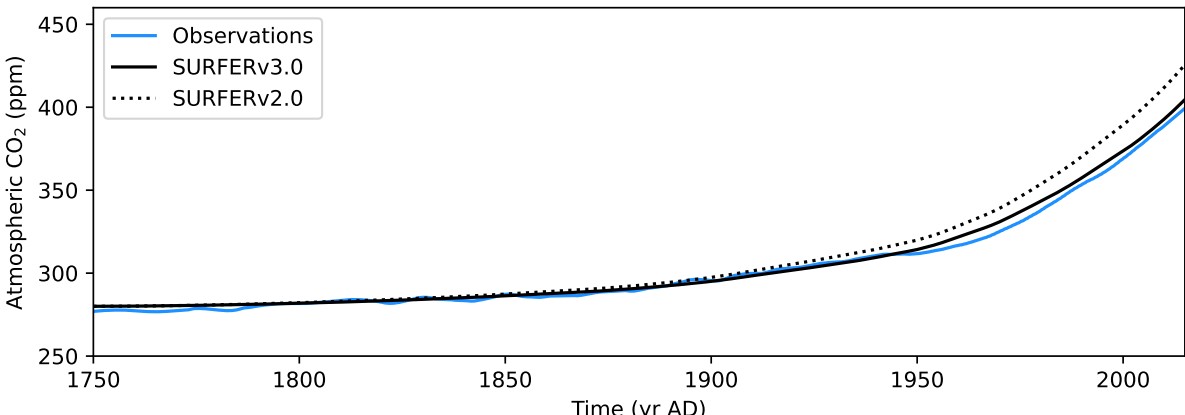

**Figure 4.** Historical atmospheric $CO_2$ concentrations. Comparaison between (smoothed) observations (Köhler et al., 2017) and outputs from SURFER v2.0 and SURFER v3.0 when forced with historical emissions.

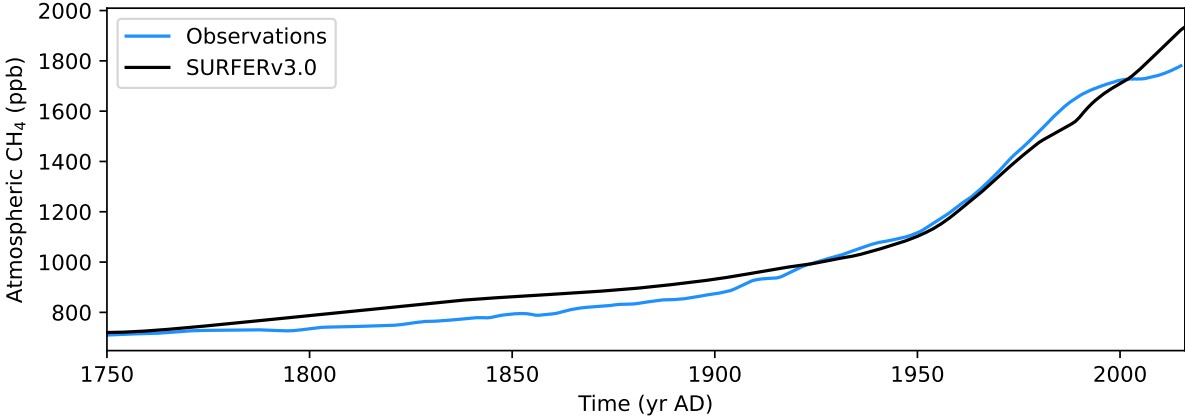

**Figure 5.** Historical atmospheric $CH_4$ concentrations. Comparaison between (smoothed) observations (Köhler et al., 2017) and outputs from SURFER v3.0 when forced with historical emissions.





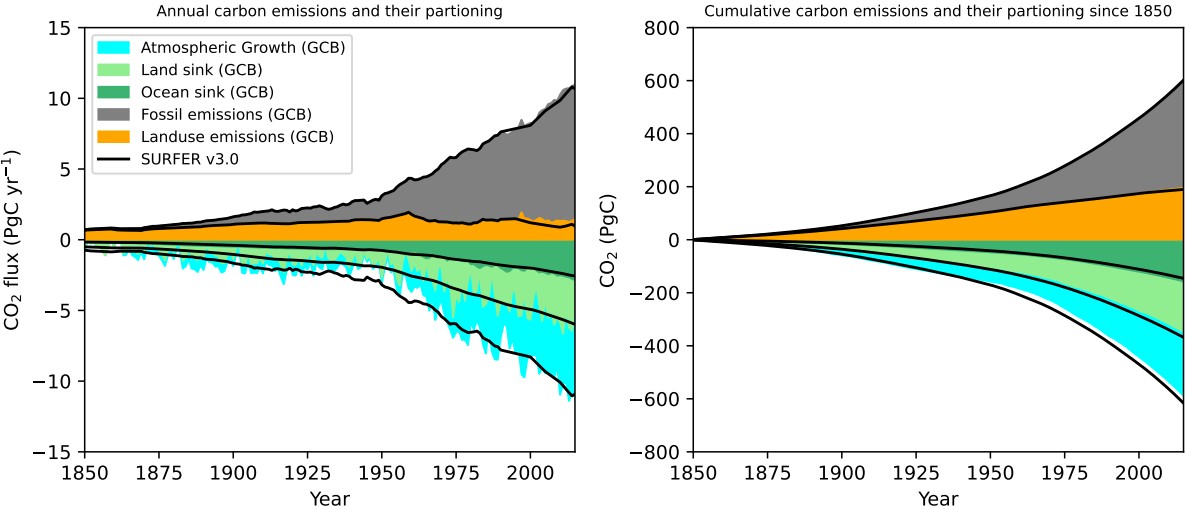

**Figure 6.** Partitionning of fossil and land-use $CO_2$ emissions in the atmosphere, ocean and land in SURFER v3.0 compared to estimates from the Global Carbon Budget (GCB) (Friedlingstein et al., 2022).





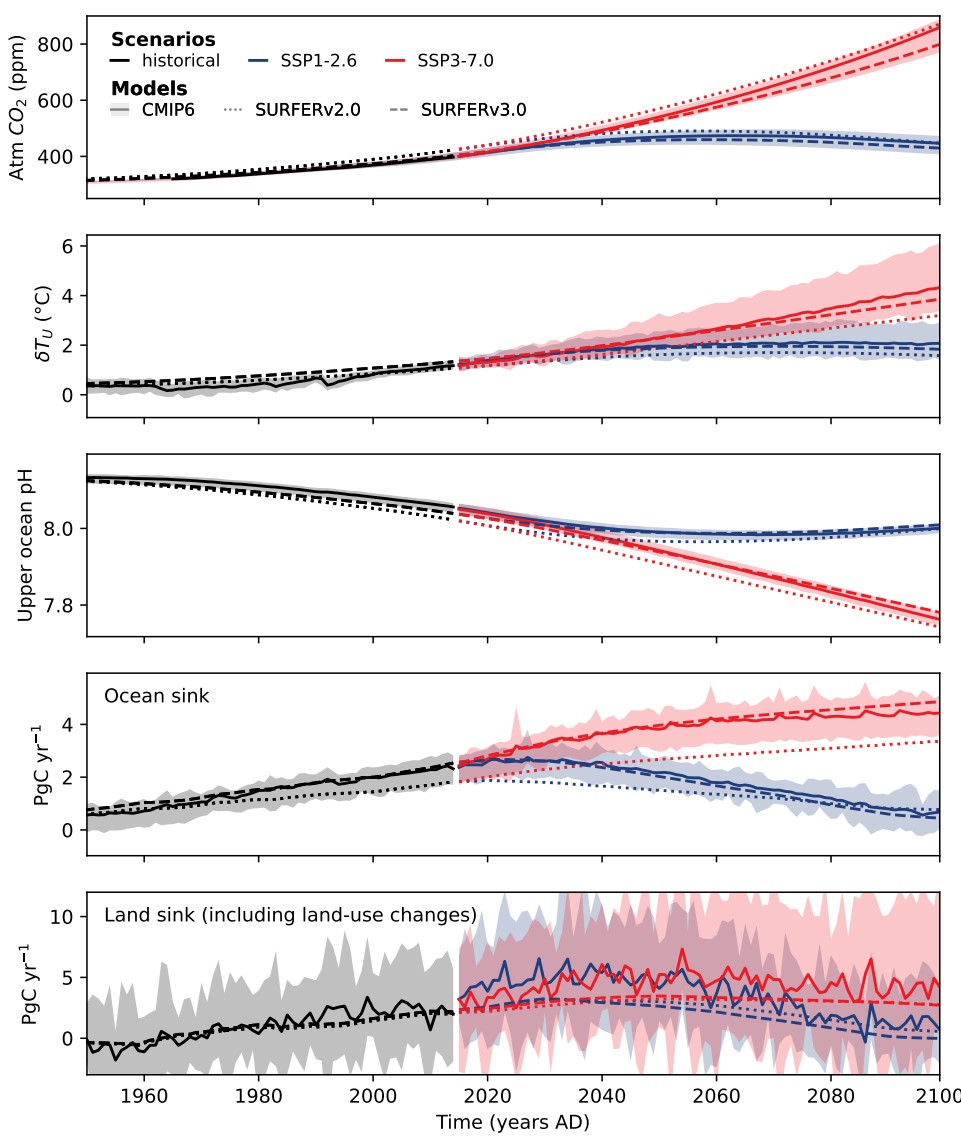

**Figure 7.** Comparison between SURFER v3.0, SURFER v2.0 and the CMIP6 model ensemble mean for the historical period (1750-2014) and the near future (2015-2100) under SSP1-2.6 and SSP3-7.0 scenarios. The CMIP6 data is from concentration driven runs, except the atmospheric $CO_2$ which comes from emission driven runs. The ocean sink is computed in SURFER as $F_{A \to U} - F_{A \to U}(t_{PI})$. The land sink is taken here as the Net Biome Produtivity (NBP) which includes land-use fluxes. In SURFER, it is computed as $F_{A \to U} - E_{\text{land-use}}^{CO_2} - E_{\text{land-use}}^{CH_4}$.



**Figure 8.** Comparison of atmosphere-to-ocean and atmosphere-to-land carbon fluxes as simulated by SURFER v3.0 and CMIP6 models for the historical period (1750-2014) and the future (2015-2300) under SSP1-2.6, SSP5-3.4 over, and SSP5-8.5 scenarios. The ocean sink is computed in SURFER as $F_{A \to U} - F_{A \to U}(t_{PI})$. The land sink is taken here as the Net Biome Produtivity (NBP) which includes land-use fluxes. In SURFER, it is computed as $F_{A \to U} - E_{\text{land-use}}^{CO_2} - E_{\text{land-use}}^{CH_4}$.




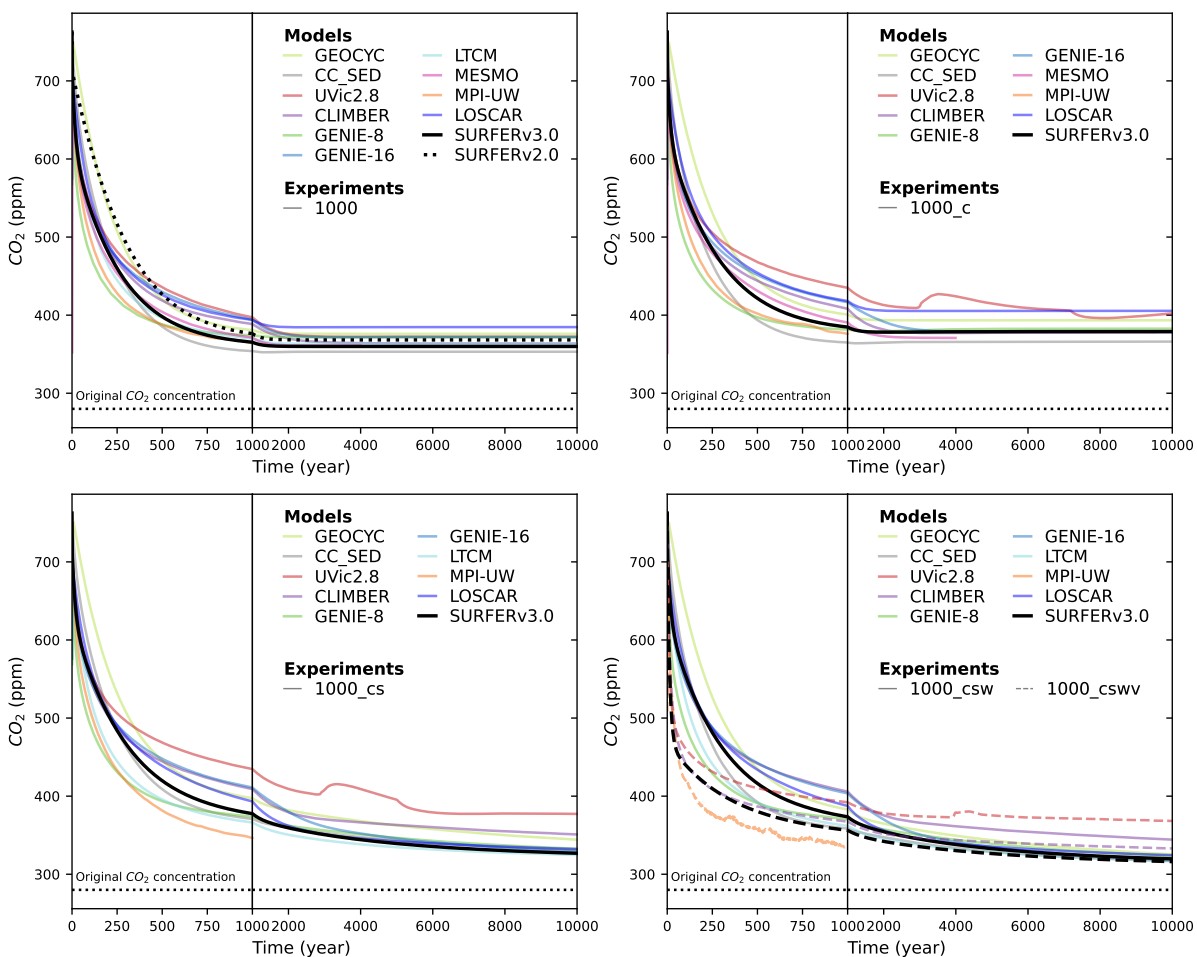

**Figure 9.** Atmospheric $CO_2$ simulated by different models and SURFER v3.0 after a 1000 $PgC$ emission pulse for the LTMIP experiments. The five experiments are the baseline (ocean only) experiment, the climate (C), the climate plus sediments (CS), the climate plus sediments plus weathering (CSW) and the climate plus sediments plus weathering plus vegetation (CSWV) experiments. We have added the results of the LOSCAR model (Zeebe, 2012) which wasn't part of the original LTMIP publication (Archer et al., 2009).





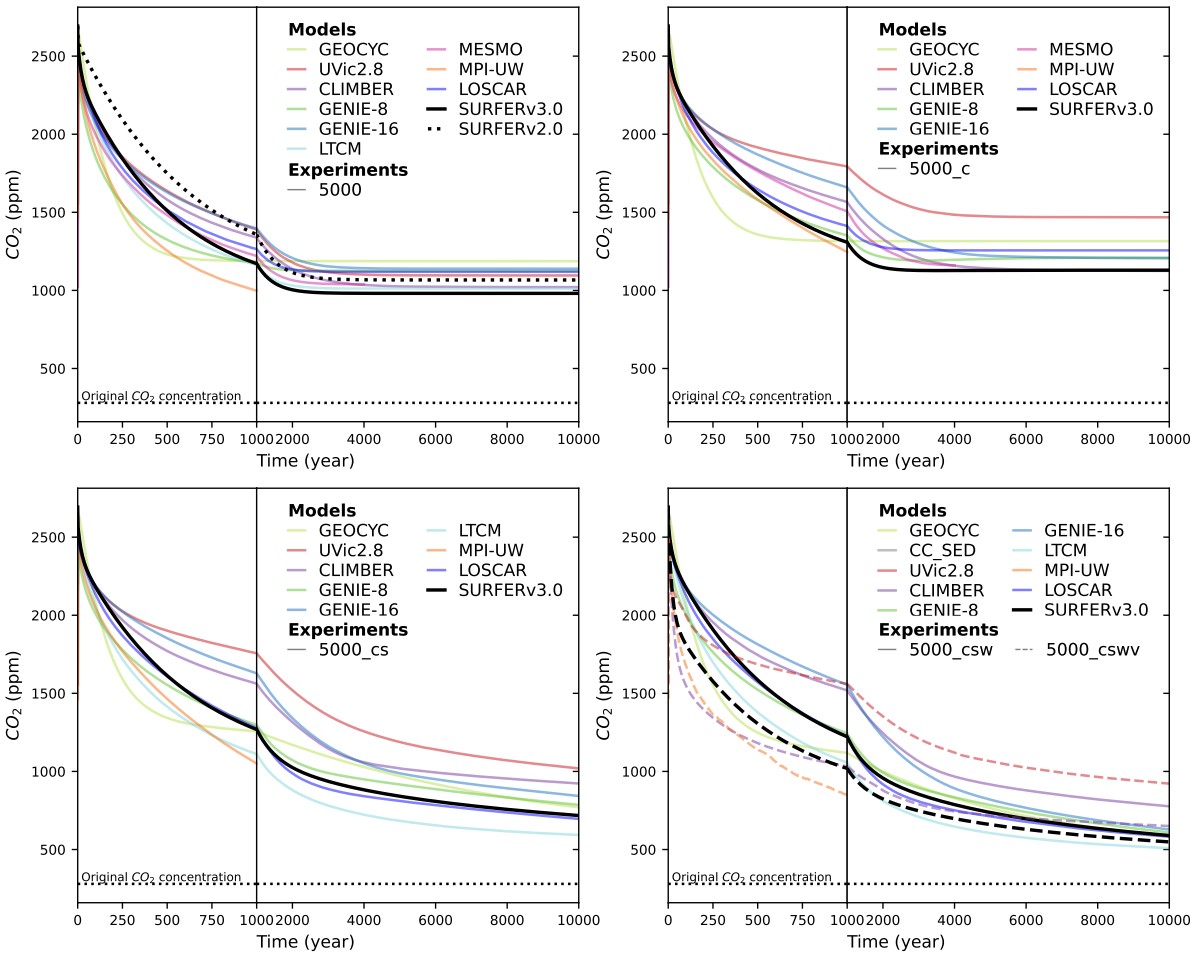

**Figure 10.** Atmospheric $CO_2$ simulated by different models and SURFER v3.0 after a 5000 $PgC$ emission pulse for the LTMIP experiments. The five experiments are the baseline (ocean only) experiment, the climate (C), the climate plus sediments (CS), the climate plus sediments plus weathering (CSW) and the climate plus sediments plus weathering plus vegetation (CSWV) experiments. We have added the results of the LOSCAR model (Zeebe, 2012) which wasn't part of the original LTMIP publication (Archer et al., 2009).



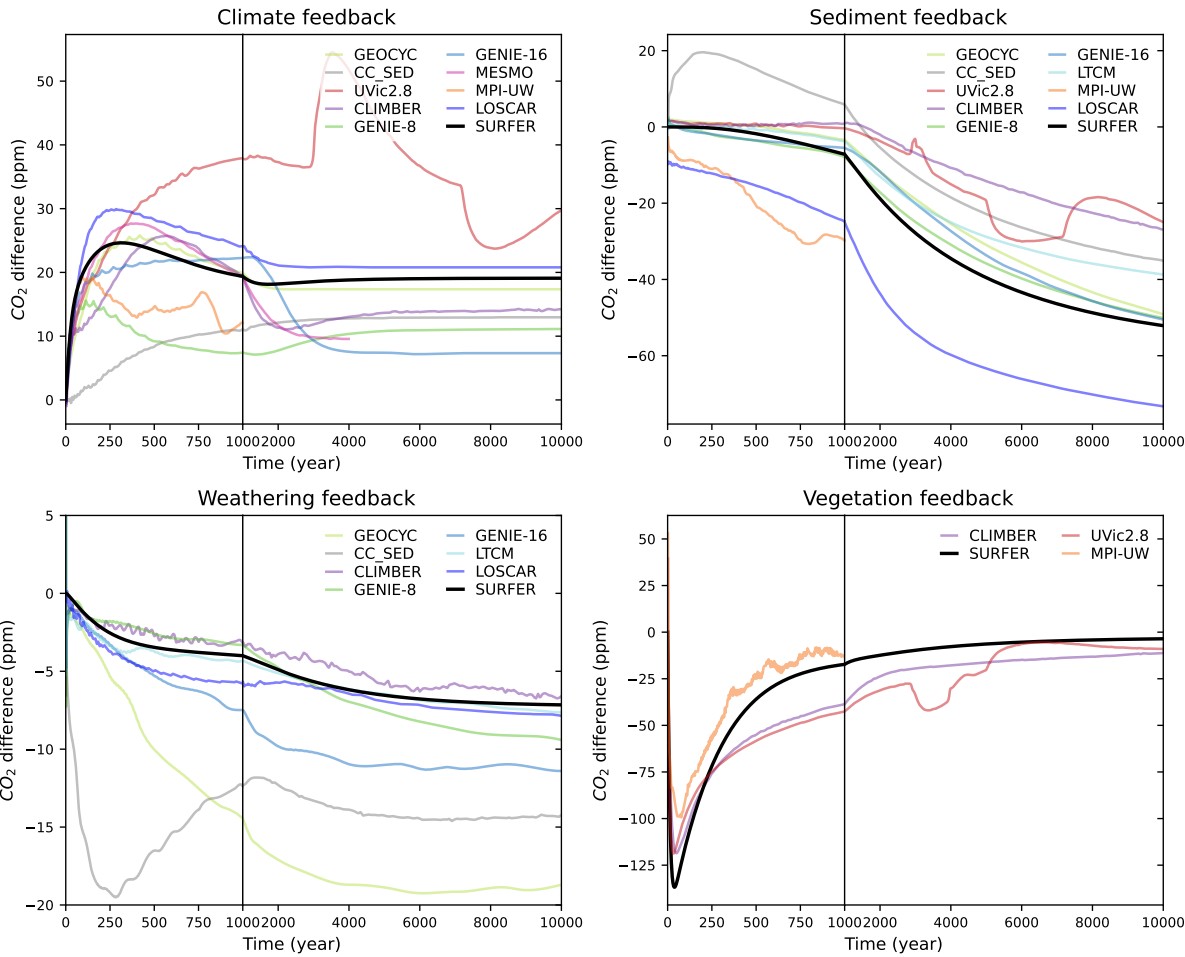

**Figure 11.** Impacts of the climate, sediments, weathering, and vegetation feedbacks on the atmospheric $CO_2$ concentration, after a 1000 PgC emission pulse. Here, a feedback is defined as the difference in $CO_2$ concentration resulting from the addition of the associated process.





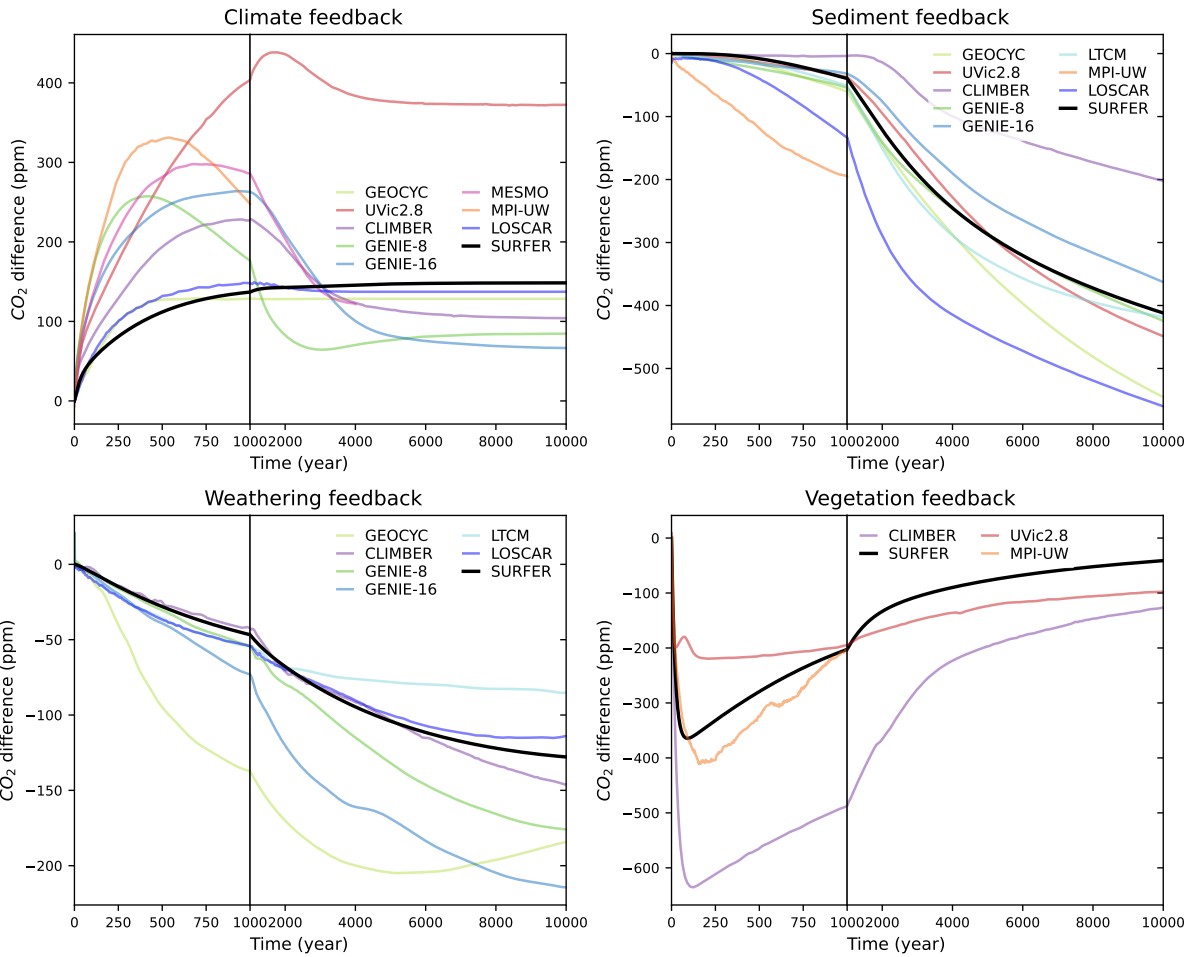

**Figure 12.** Impacts of the climate, sediments, weathering, and vegetation feedbacks on the atmospheric $CO_2$ concentration, after a 5000 $PgC$ emission pulse. Here, a feedback is defined as the difference in $CO_2$ concentration resulting from the addition of the associated process.



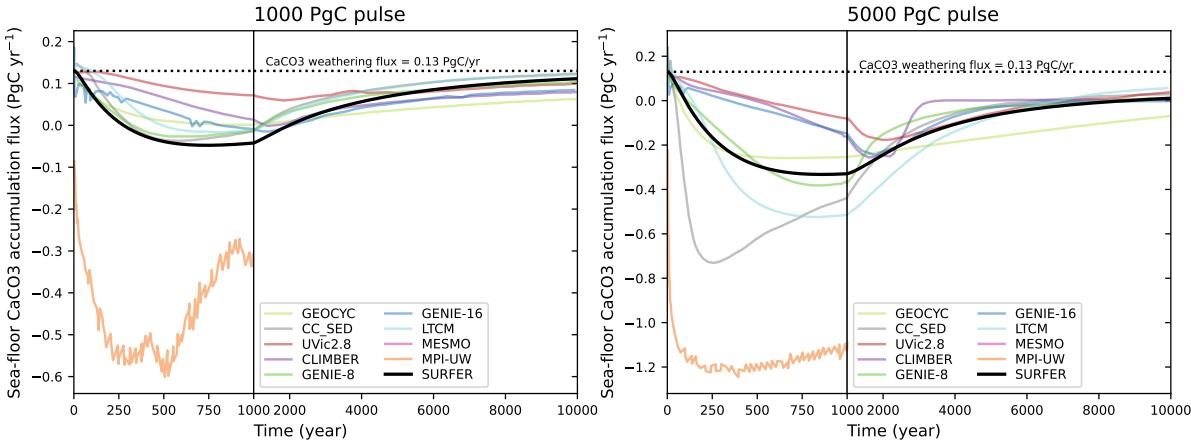

**Figure 13.** CaCO$_3$ accumulation fluxes in sediments for the different models. Negative values indicate net dissolution of CaCO$_3$.

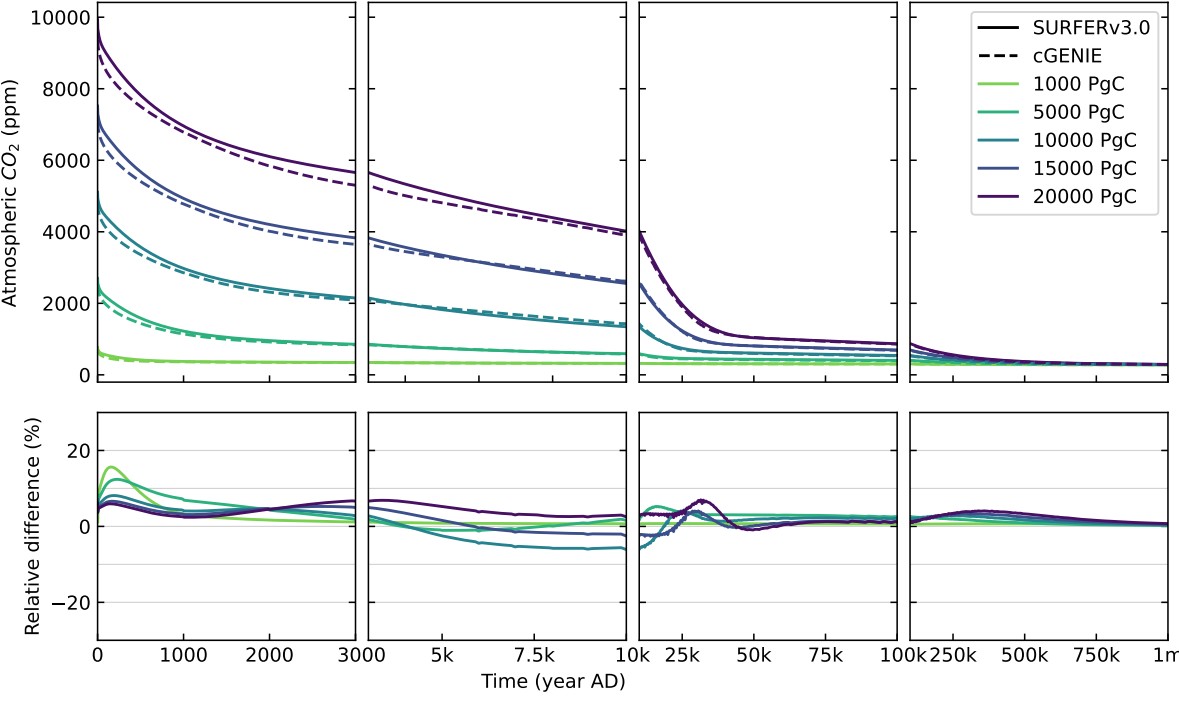

**Figure 14.** Atmospheric CO$_2$ concentrations simulated by cGENIE and SURFER v3.0 after emissions pulses of different sizes. The bottom plot shows the relative difference between SURFER v3.0 and cGENIE.



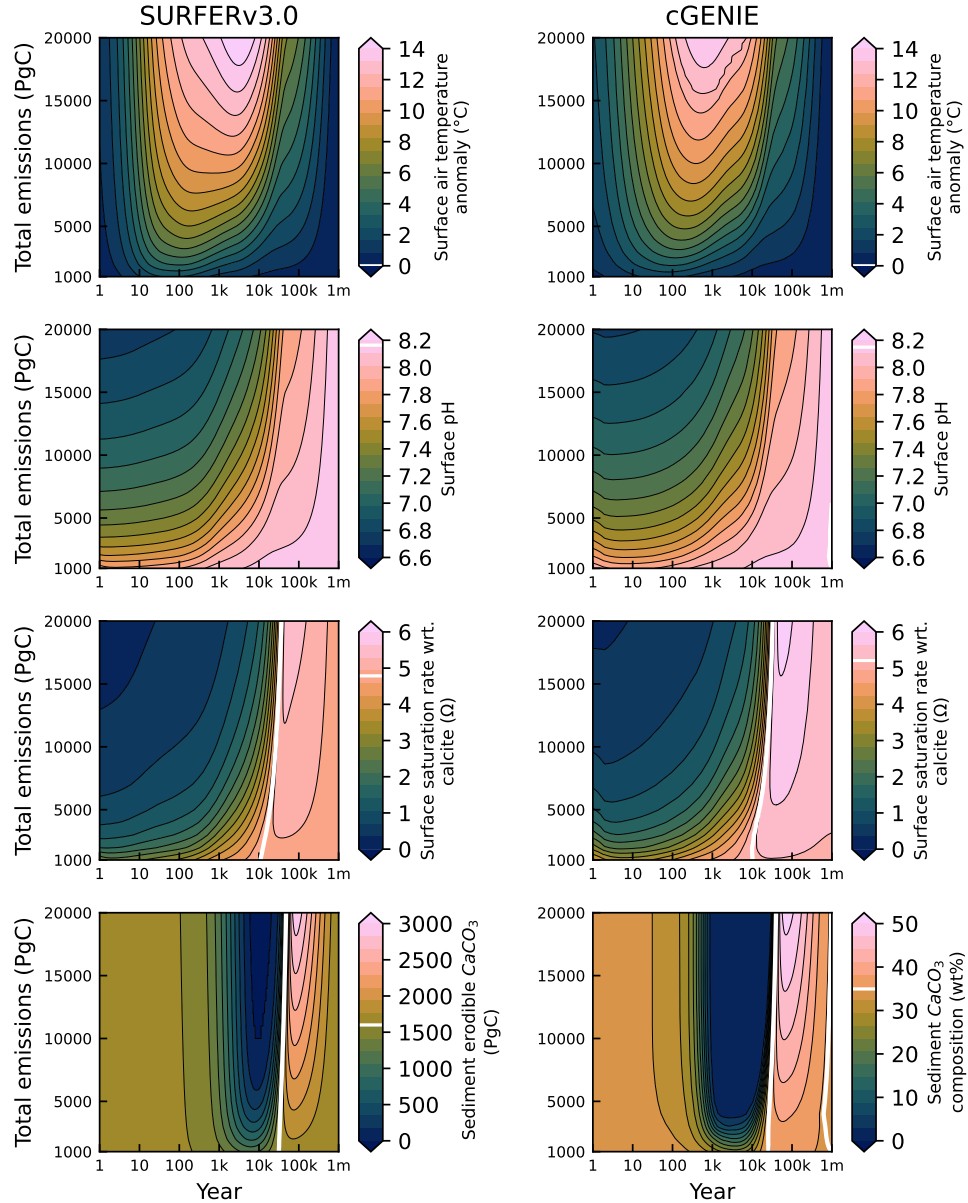

**Figure 15.** Global mean temperature anomaly, surface ocean pH, surface ocean saturation state with respect to calcite, and CaCO₃ sediments content in SURFER v3.0 and cGENIE after emissions pulses ranging from 1000 PgC to 20000 PgC. White lines indicate preindustrial values used in each model. Note : SURFER v3.0 and cGENIE use different units for the CaCO₃ sediments content. SURFER uses the total erodible CaCO₃ mass, while cGENIE uses the mean dry weight fraction (mass of CaCO₃ divided by the mass of CaCO₃ and non erodible material in sediments). Although these two quantities are strongly correlated, they do not necessarily depend linearly on one another, complicating direct comparisons.




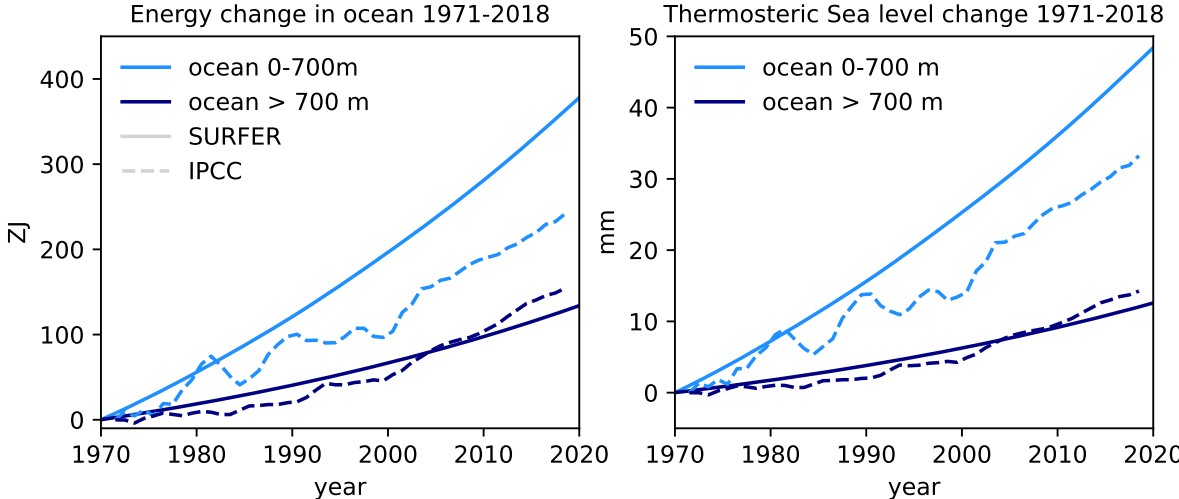

**Figure 16.** Ocean heat content and thermosteric sea level changes between the years 1971 and 2018 as estimated by the IPCC (Fox-Kemper et al., 2021), and simulated by SURFER v3.0.





**Figure 17.** Atmospheric $CO_2$, surface temperature, and thermosteric sea level rise simulated by SURFER v3.0, SURFER v2.0, and two versions of the UVic model of intermediate complexity for two emission scenrarios : 1280 PgC (blue) and 3840 PgC (red).





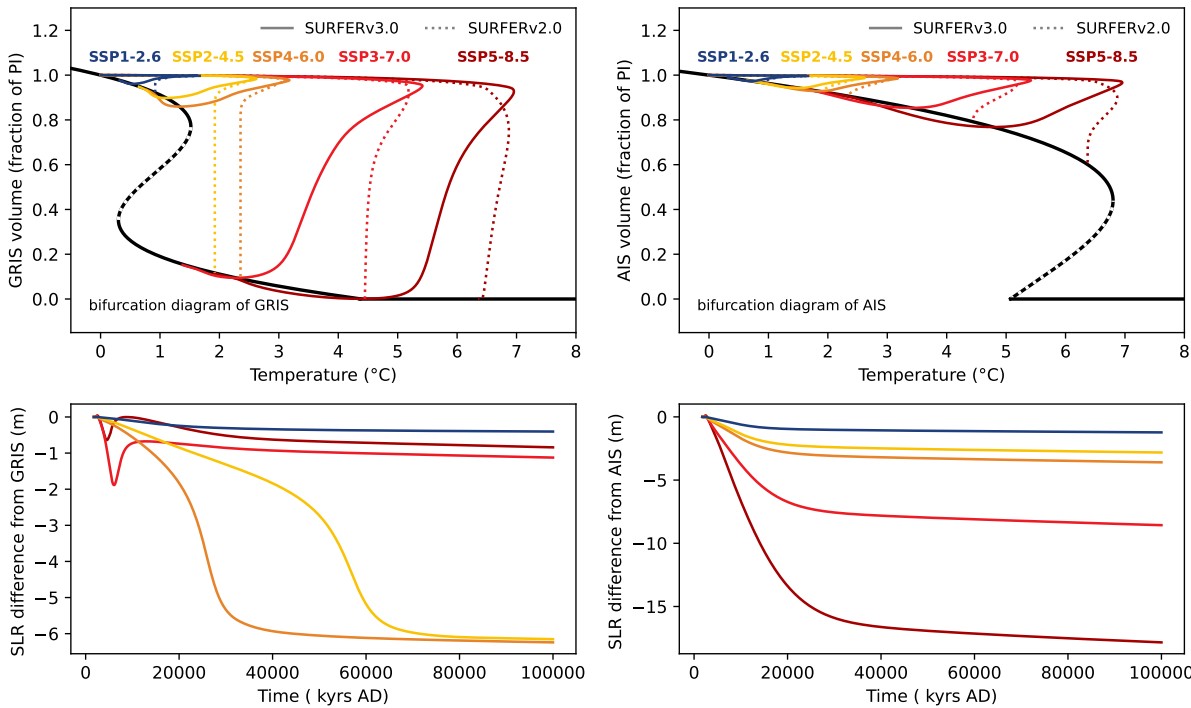

**Figure 18.** Comparison of Greenland and Antarctic sea level rise contributions in SURFER v3.0 and SURFER v2.0 for SSP emission scenarios. Top plots: Volume fractions of Greenland and Antarctica ice sheets as a function of temperature increase. Equilibrium values are shown in black, with stable equilibria represented by solid lines and unstable equilibria by dashed lines. Bottom plots: difference in sea level rise contributions from Greenland and Antarctica in SURFER v3.0 and SURFER v2.0. Negatives values indicate a reduced sea level rise contribution in SURFER v3.0 compared to SURFER v2.0.



**Appendix A: Emission scenarios**

**Historical and SSP runs** (Figures 3-8,16, 18, D1-D11)

For $CO_2$ emissions up to 1989 (included), we use the data from the Global Carbon Budget (Friedlingstein et al., 2022). Fossil emissions start in 1750 and we include the cement carbonation sink in them. Land-use emissions are only provided from the year 1850 and so we assume that they have grown linearly from zero since 1750. This adds around 33 PgC of emissions compared to a scenario where land-use emissions are considered zero before 1850.

For $CH_4$ emissions up to 1989 (included), we use the data from Jones et al. (2023). Both fossil and land-use emissions start

in 1830 so, similarly to the $CO_2$ land-use emissions, we assume that they increased linearly from 0 since 1750. This adds a total of ~1147 MtCH$_4$ of emissions compared to a scenario where land-use and fossil $CH_4$ emissions are considered zero before 1830.

For all emissions ($CO_2$ and $CH_4$, fossil and land-use) from 1990 to 2100, we use the values provided in the SSP database (https://tntcat.iiasa.ac.at/SspDb/dsd, Riahi et al. (2017); Gidden et al. (2019)). All SSP scenarios have the same emissions

between 1990 and 2015, so we perform our historical runs (1750-2014) with the values from any SSP scenarios. The $CO_2$ and $CH_4$ emissions provided in 1990 by Friedlingstein et al. (2022) and Jones et al. (2023) are not equal to the values provided in 1990 in the SSP database, which causes small jumps in our emissions. For emissions between 2100 and 2300, we use the SSP extensions as described in Meinshausen et al. (2020). Before the year 1750, after 2300, all anthropogenic emissions are set to zero.

**Pulse experiments** (Figures 9-15)

For these experiments, the model is started with an additional amount od carbon $x$ in the atmospheric reservoir, depending on the emission pulse : $M_A(t_{PI}) = 580.3 + x$ PgC. No other emissions are used.

**Others** (Figure 17)

For the experiments in section 4.1 where we compare SURFER to UVic, we use the $CO_2$ emissions provided in the supple-

mentary material of Clark et al. (2016), while $CH_4$ emissions are set to zero.

**Appendix B: Temperature and pressure dependance of solubility and dissociation constants**

For the temperature (and salinity) dependence of $K_0$, $K_1$, $K_2$, $K_b$, $K_w$, and $K_{sp}^{CaCO_3}$, we use the equations from Sarmiento and Gruber (2006), and originally from Weiss (1974); Mehrbach et al. (1973); Dickson and Millero (1987); Millero (1995);





Dickson (1990); Mucci (1983):

$$\ln K_0 = -60.2409 + 93.4517\left(\frac{100}{T}\right) + 23.3585 \cdot \ln\left(\frac{T}{100}\right)$$

$$+ S\left(0.023517 - 0.023656\left(\frac{T}{100}\right) + 0.0047036\left(\frac{T}{100}\right)^2\right), \tag{B1}$$

$$-\log K_1^0 = -62.008 + \frac{3670.7}{T} + 9.7944 \cdot \ln T$$

$$-0.0118 \cdot S + 0.000116 \cdot S^2, \tag{B2}$$

$$-\log K_2^0 = +4.777 + \frac{1394.7}{T} - 0.0184 \cdot S + 0.000118 \cdot S^2, \tag{B3}$$

$$\ln K_b^0 = \frac{1}{T}\left(-8966.9 - 2890.53 \cdot S^{1/2} - 77.942 \cdot S + 1.728 \cdot S^{3/2} - 0.0996 \cdot S^2\right)$$

$$+ 148.0248 + 137.1942 \cdot S^{1/2} + 1.62142 \cdot S + 0.053105 \cdot S^{1/2}T$$

$$+ \ln T(-24.4344 - 25.085 \cdot S^{1/2} - 0.2474 \cdot S), \tag{B4}$$

$$\ln K_w^0 = 148.96502 + \frac{-13847.26}{T} - 23.6521 \cdot \ln T$$

$$S^{1/2}\left(-5.977 + \frac{118.67}{T} + 1.0495 \cdot \ln T\right) - 0.01615 \cdot S, \tag{B5}$$

$$\ln K_{sp}^{CaCO_3,0} = -395.8293 + \frac{6537.773}{T} + 71.595 \cdot \ln T - 0.17959 \cdot T$$

$$+ \left(-1.78938 + \frac{410.64}{T} + 0.0065453 \cdot T\right)S^{1/2}$$

$$-0.17755 \cdot S + 0.0094979 \cdot S^{2/3}. \tag{B6}$$

Here, $K_0$, $K_1^0$, $K_2^0$, $K_w^0$, $K_b^0$, and $K_{sp}^{CaCO_3,0}$ are the values of the constants at atmospheric pressure, given in $\mathrm{mol\,(kg\,atm)^{-1}}$ for $K_0$, in $\mathrm{mol\,kg^{-1}}$ for $K_1^0$, $K_2^0$, $K_b^0$, and in $(\mathrm{mol\,kg^{-1}})^2$ for $K_w^0$ and $K_{sp}^{CaCO_3,0}$, $T$ is the temperature in Kelvin and $S$ is the salinity on the practical salinity scale. In SURFER, only temperature anomalies are computed, meaning that to compute the absolute temperature and then the dissociation constants, we need to provide an initial temperature for each layer. This is done in section 2.4.2.

The pressure (depth) dependence for $K_1$, $K_2$, $K_w$, $K_b$, and $K_{sp}^{CaCO_3}$ is from Millero (1995),

$$\ln\left(K_i^p/K_i^0\right) = -\frac{\Delta V_i}{RT}P + \frac{0.5\Delta K_i}{RT}P^2, \tag{B7}$$

where $K_i^p$ is the value of the dissociation constant at pressure $P$ (in bar), $K_i^0$ is the value of the dissociation constant at atmospheric pressure (1.01325 bar or 101325 Pa), $T$ is the temperature in Kelvin and $R$ is the gas constant in $\mathrm{bar\,cm^3\,mol^{-1}\,K^{-1}}$ (10 times the value in $\mathrm{J\,mol^{-1}\,K^{-1}}$). The quantities $\Delta V_i$ and $\Delta K_i$ are changes in molal volume and compressibility and are estimated following :

$$\Delta V_i = a_{0,i} + a_{1,i}(T - 273.15) + a_{2,i}(T - 273.15)^2, \tag{B8}$$

$$1000\Delta K_i = b_{0,i} + b_{1,i}(T - 273.15). \tag{B9}$$





| | $a_0$ | $a_1$ | $a_2$ | $b_0$ | $b_1$ |
|---|---|---|---|---|---|
| $K_1$ | -25.50 | 0.1271 | 0 | -3.08 | 0.0877 |
| $K_2$ | -15.82 | -0.0219 | 0 | 1.13 | -0.1475 |
| $K_\mathrm{b}$ | -29.48 | 0.1622 | -0.002608 | -2.84 | 0 |
| $K_\mathrm{w}$ | -25.60 | 0.2324 | -0.0036246 | -5.13 | 0.0794 |
| $K_\mathrm{sp}^{\mathrm{CaCO3}}$ | -48.76 | -0.5304 | 0 | -11.76 | 0.3692 |

**Table B1.** Parametrisation for the effect of pressure on dissociation constants ; coefficients for equations B8 and B9. The values are taken from Millero (1995). We have flipped the signs of $a_1$ and $a_2$ for $K_\mathrm{b}$, and $b_1$ for $K_\mathrm{sp}^{\mathrm{CaCO3}}$ to match the values given in Millero (1979).

Values for the coefficients are in table B1.

Hydrostatic balance provides the pressure at a given depth, thus, pressure (in $\mathrm{bar}$) is given by $P = \rho g z / 100000$ where $\rho = 1026 \ kg/m^3$ is sea water mean density, $g = 9.81 \ \mathrm{kg\,m\,s^{-2}}$ is the gravitational acceleration and $z$ is the depth (in m). For the computation of mean quantities in the different ocean levels U, I and D, we use the depths $z_\mathrm{U} = 75$ m, $z_\mathrm{I} = 400$ m and $z_\mathrm{D} = 2225$ m which corresponds to the mid-depth points of the layers.

**Appendix C: Alkalinity and solving the carbonate system**

In SURFER v2.0, we approximated alkalinity by carbonate alkalinity

$$\mathrm{Alk} \approx \mathrm{Alk_C} = \left[\mathrm{HCO_3^-}\right] + 2\left[\mathrm{CO_3^{2-}}\right] . \tag{C1}$$

To compute $[\mathrm{H^+}]$ and the other carbonate species, we had to rewrite eq C1 as a function of the four known quantities DIC, Alk, $T$, and $S$, and the unknown quantity $[\mathrm{H^+}]$. Using the definition of DIC (eq 11) and of the dissociation constants $K_1$ and $K_2$ (eq 17 and eq 18) we can express DIC as a function of $[\mathrm{HCO_3^-}]$ and $[\mathrm{H^+}]$:

$$\mathrm{DIC} = \left(1 + \frac{K_2}{[\mathrm{H^+}]} + \frac{[\mathrm{H^+}]}{K_1}\right)[\mathrm{HCO_3^-}], \tag{C2}$$

or equivalently, we can express $[\mathrm{HCO_3^-}]$ as a function of DIC and $[\mathrm{H^+}]$

$$[\mathrm{HCO_3^-}] = \frac{\mathrm{DIC} \cdot K_1[\mathrm{H^+}]}{K_1 K_2 + K_1[\mathrm{H^+}] + [\mathrm{H^+}]^2} . \tag{C3}$$

Using equation 18, we can then express $[\mathrm{CO_3^{2-}}]$ as a function of DIC and $[\mathrm{H^+}]$

$$\begin{aligned}
[\mathrm{CO_3^{2-}}] &= \frac{[\mathrm{HCO_3^-}]K_2}{[\mathrm{H^+}]} \\
&= \frac{\mathrm{DIC} \cdot K_1 K_2}{K_1 K_2 + K_1[\mathrm{H^+}] + [\mathrm{H^+}]^2} .
\end{aligned} \tag{C4}$$

Inserting eqs C3 and C4 in C1, we get

$$\mathrm{Alk_C} = \mathrm{DIC} \cdot \frac{K_1[\mathrm{H^+}] + K_1 K_2}{K_1 K_2 + K_1[\mathrm{H^+}] + [\mathrm{H^+}]^2} , \tag{C5}$$



which we can solve for $[H^+]$, given Alk, DIC and the dissociations constants $K_1$ and $K_2$ (which depend on $T$ and $S$). To do so, we write eq C5 as a degree 2 polynomial equation

$$P_{\mathrm{C}}([H^+]) \equiv [H^+]^2 + a_1[H^+] + a_0, \tag{C6}$$

with

$$a_1 = K_1 \left(1 - \frac{\mathrm{DIC}}{\mathrm{Alk}}\right), \tag{C7}$$

$$a_0 = K_1 K_2 \left(1 - \frac{2 \cdot \mathrm{DIC}}{\mathrm{Alk}}\right). \tag{C8}$$

The positive root is given by

$$[H^+] = \frac{K_1}{2 \cdot \mathrm{Alk}} \left(\sqrt{(\mathrm{DIC} - \mathrm{Alk})^2 - 4\frac{K_2}{K_1}\mathrm{Alk} \cdot (\mathrm{Alk} - 2 \cdot \mathrm{DIC})} + (\mathrm{DIC} - \mathrm{Alk})\right). \tag{C9}$$

Then, $[\mathrm{HCO_3^-}]$ can be computed from equation C3, $[\mathrm{CO_3^{2-}}]$ can be computed from equation C4, and finally, $[\mathrm{H_2CO_3^*}]$ can be computed from equation 17.

In SURFER v3.0, we approximate alkalinity by the carbonate, borate and water self-ionisation alkalinity

$$\mathrm{Alk} \approx \mathrm{Alk_{CBW}} = \left[\mathrm{HCO_3^-}\right] + 2\left[\mathrm{CO_3^{2-}}\right] + \left[\mathrm{OH^-}\right] - \left[\mathrm{H^+}\right] + \left[\mathrm{B(OH)_4^-}\right] \tag{C10}$$

As before, to compute $[H^+]$ and the other carbonate species, we have to rewrite eq C10 as a function of the 4 know quantities DIC, Alk, $T$, and $S$ and the unknown quantity $[H^+]$. We already know how express $[\mathrm{HCO_3^-}]$ and $[\mathrm{CO_3^{2-}}]$ as a function of DIC and $[H^+]$ (eqs C3 and C4). Equation 19 further gives us

$$\left[\mathrm{OH^-}\right] = \frac{K_{\mathrm{w}}}{[H^+]}, \tag{C11}$$

and we can use equations 20 and 21 to obtain

$$\left[\mathrm{B(OH)_4^-}\right] = \frac{c_{\mathrm{b}} \cdot S \cdot K_{\mathrm{b}}}{[H^+] + K_{\mathrm{b}}}. \tag{C12}$$

Inserting these results in equation C10, we get

$$\mathrm{Alk} \approx \mathrm{Alk}_{CBW} = \mathrm{DIC} \cdot \frac{K_1[H^+] + K_1 K_2}{K_1 K_2 + K_1[H^+] + [H^+]^2} + \frac{K_{\mathrm{w}}}{[H^+]} - \left[H^+\right] + \frac{c_{\mathrm{b}} \cdot S \cdot K_{\mathrm{b}}}{[H^+] + K_{\mathrm{b}}} \tag{C13}$$

which we can solve for $[H^+]$, given Alk, DIC and the dissociations constants $K_1$, $K_2$, $K_{\mathrm{w}}$ and $K_{\mathrm{b}}$ (which depend on $T$ and $S$). To do this, we follow Munhoven (2013) and we write eq C13 as a polynomial equation that is now of degree 5

$$P_{\mathrm{CBW}}([H^+]) \equiv [H^+]^5 + q_4[H^+]^4 + q_3[H^+]^3 + q_2[H^+]^2 + q_1[H^+] + q_0, \tag{C14}$$





with

$$q_4 = \mathrm{Alk} + K_1 + K_\mathrm{b}\,, \tag{C15}$$

$$q_3 = (\mathrm{Alk} - \mathrm{DIC} + K_\mathrm{b})K_1 + (\mathrm{Alk} - c_\mathrm{b} \cdot S)K_\mathrm{b} + K_1 K_2 - K_\mathrm{w}\,, \tag{C16}$$

$$q_2 = (\mathrm{Alk} - 2 \cdot \mathrm{DIC} + K_\mathrm{b})K_1 K_2 + (\mathrm{Alk} - \mathrm{DIC} - c_\mathrm{b} \cdot S)K_1 K_\mathrm{b} - K_1 K_\mathrm{w} - K_\mathrm{b} K_\mathrm{w}\,, \tag{C17}$$

$$q_1 = (\mathrm{Alk} - 2 \cdot \mathrm{DIC} - c_\mathrm{b} \cdot S)K_1 K_2 K_\mathrm{b} - K_1 K_2 K_\mathrm{w} - K_1 K_\mathrm{b} K_\mathrm{w}\,, \tag{C18}$$

$$q_0 = -K_1 K_2 K_\mathrm{b} K_\mathrm{w}\,. \tag{C19}$$

We solve this equation using the Newton-Raphson method. To ensure quick convergence, we need a good initial guess that is not too far from the real solution. We adopt the following procedure from Munhoven (2013) and Humphreys et al. (2022). We define the following coefficients

$$c_2 = K_\mathrm{b}\left(1 - \frac{c_\mathrm{b} \cdot S}{\mathrm{Alk}}\right) + K_1\left(1 - \frac{\mathrm{DIC}}{\mathrm{Alk}}\right)\,, \tag{C20}$$

$$c_1 = K_1 K_\mathrm{b}\left(1 - \frac{c_\mathrm{b} \cdot S}{\mathrm{Alk}} - \frac{\mathrm{DIC}}{\mathrm{Alk}}\right) + K_1 K_2\left(1 - 2\frac{\mathrm{DIC}}{\mathrm{Alk}}\right)\,, \tag{C21}$$

$$c_0 = K_1 K_2 K_\mathrm{b}\left(1 - \frac{2 \cdot \mathrm{DIC} + c_\mathrm{b} \cdot S}{\mathrm{Alk}}\right)\,, \tag{C22}$$

from which we construct the quantities

$$H_\mathrm{min} = \frac{-c_2 + \sqrt{c_2^2 - 3c_1}}{3}\,, \tag{C23}$$

and

$$H_0 = H_\mathrm{min} + \sqrt{-\frac{H_\mathrm{min}^3 + c_2 H_\mathrm{min}^2 + c_1 H_\mathrm{min} + c_0}{\sqrt{c_2^2 - 3c_1}}}\,. \tag{C24}$$

The initial guess for the Newton-Raphson method is taken as

$$[\mathrm{H^+}]_0 = \begin{cases} 10^{-3} & \text{if } \mathrm{Alk} \leq 0\,, \\ H_0 & \text{if } 0 < \mathrm{Alk} < 2 \cdot \mathrm{DIC} + c_\mathrm{b} \cdot S \text{ and } c_2^2 - 3c_1 > 0\,, \\ 10^{-7} & \text{if } 0 < \mathrm{Alk} < 2 \cdot \mathrm{DIC} + c_\mathrm{b} \cdot S \text{ and } c_2^2 - 3c_1 < 0\,, \\ 10^{-10} & \text{if } \mathrm{Alk} \geq 2 \cdot \mathrm{DIC} + c_\mathrm{b} \cdot S\,. \end{cases} \tag{C25}$$

The rationale behind this choice is given in Munhoven (2013) and Humphreys et al. (2022). With it, we only need 5 Newton-Raphson iterations to obtain an accurate value for $[\mathrm{H^+}]$ and pH, which is defined as

$$\mathrm{pH} = -\log_{10}([\mathrm{H^+}])\,. \tag{C26}$$

Once $[\mathrm{H^+}]$ is computed, $[\mathrm{HCO_3^-}]$ and $[\mathrm{CO_3^{2-}}]$ can be computed from equations C3 and C4, $[\mathrm{H_2CO_3}^*]$ can be computed from equation 17, $[\mathrm{OH^-}]$ can be computed from equation C11, and $[\mathrm{B(OH)_4^-}]$ can be computed from equation C12. Equation




C10 is solved following this procedure at every time step of the numerical integration of the model. This is because we need $[\mathrm{H}^+]_\mathrm{U}$ for the computation of $B_\mathrm{U}$ in the atmosphere to upper ocean carbon flux $F_{\mathrm{A}\to\mathrm{U}}$ (see equations 27 and 28), and we need $[\mathrm{CO}_3^{2-}]_\mathrm{D}$ for the computation of the dissolution flux $F_\mathrm{diss}$ (see eq 48). This is also the procedure used to obtain climatologies (and vertical profiles) of the carbonate species from the GLODAP climatologies of DIC, Alk, temperature and salinity (see Figure 3). In this case, we computed the carbonate species at each ocean point where DIC, Alk, temperature and salinity were all given.

To determine the initial condition for the dissolved inorganic carbon mass in the upper layer $M_\mathrm{U}(t_\mathrm{PI})$, we need to compute $\mathrm{DIC}_\mathrm{U}(t_\mathrm{PI})$ as a function of $\mathrm{Alk}_\mathrm{U}(t_\mathrm{PI})$, $T_\mathrm{U}(t_\mathrm{PI})$, $S_\mathrm{U}$, and $[\mathrm{H}_2\mathrm{CO}_3^*]_\mathrm{U}(t_\mathrm{PI})$, which is fixed by equilibrium conditions (eq 108). Now DIC is an unknown and we can't use the procedure described above. Instead, we write Alk as a function of $[\mathrm{H}_2\mathrm{CO}_3^*]$ and $[\mathrm{H}^+]$ (and the dissociation ). Using equations 17 and 18 we get

$$[\mathrm{HCO}_3^-] = \frac{K_1[\mathrm{H}_2\mathrm{CO}_3^*]}{[\mathrm{H}^+]}, \tag{C27}$$

$$[\mathrm{CO}_3^{2-}] = \frac{K_1 K_2[\mathrm{H}_2\mathrm{CO}_3^*]}{[\mathrm{H}^+]^2}, \tag{C28}$$

and thus

$$Alk \approx Alk_\mathrm{CBW} = \frac{K_1[\mathrm{H}_2\mathrm{CO}_3^*]}{[\mathrm{H}^+]} + 2\frac{K_1 K_2[\mathrm{H}_2\mathrm{CO}_3^*]}{[\mathrm{H}^+]^2} + \frac{K_\mathrm{w}}{[\mathrm{H}^+]} - [\mathrm{H}^+] + \frac{c_\mathrm{b}\cdot S \cdot K_\mathrm{b}}{[\mathrm{H}^+]+K_\mathrm{b}}. \tag{C29}$$

This equation can be solved numerically for $[\mathrm{H}^+]$ using any algorithm. The speed of the algorithm is not of great importance here since we only perform the computation for the setting of the initial conditions, and not at each time step of the numerical integration. Once $[\mathrm{H}^+]$ is obtained, $[\mathrm{HCO}_3^-]$ and $[\mathrm{CO}_3^{2-}]$ are recovered with equations 17 and 18 and DIC can then be computed as the sum of $[\mathrm{HCO}_3^-]$, $[\mathrm{CO}_3^{2-}]$ and $[\mathrm{H}_2\mathrm{CO}_3^*]$.

## Appendix D: Sensitivity analysis

We present here a sensitivity analysis of the model in response to changes in specific parameters. While the primary goal is to assess how these sometimes arbitrary parameter choices affect model behaviour, an added benefit is the ability to investigate the timescales on which the associated processes are significant.

We first test how atmospheric $\mathrm{CO}_2$ levels simulated for the SSP3-7.0 scenario from 1750 AD to 1 000 000 AD respond to changes in 20 parameters associated with the carbon cycle component of SURFER v3.0 (all parameters from the first part of table 2, except $\tau_{\mathrm{CH}_4}$). Most parameters are adjusted one at a time, with interdependent parameters and $M_\mathrm{U}(t_\mathrm{PI})$ being modified accordingly to obey the preindustrial equilibrium condition (see second part of table 2). Each parameter is varied within a range of 0.25 to 1.75 times its default value. Note that these ranges do not necessarily reflect physically plausible values.

Changing $k_{\mathrm{A}\to\mathrm{U}}$ in the range described above has a negligible impact on the atmospheric $\mathrm{CO}_2$ draw-down (figure D1). This is because regardless of the values set for $k_{\mathrm{A}\to\mathrm{U}}$, the transfer rate of $\mathrm{CO}_2$ between the atmosphere and the ocean surface layer is very fast compared to the subsequent transport of dissolved carbon at depth, which therefore is the real limiting factor of for oceanic $\mathrm{CO}_2$ uptake. Indeed, we observe that increasing $k_{\mathrm{U}\to\mathrm{I}}$ or $k_{\mathrm{I}\to\mathrm{D}}$ leads to an increased $\mathrm{CO}_2$ uptake for up to 100



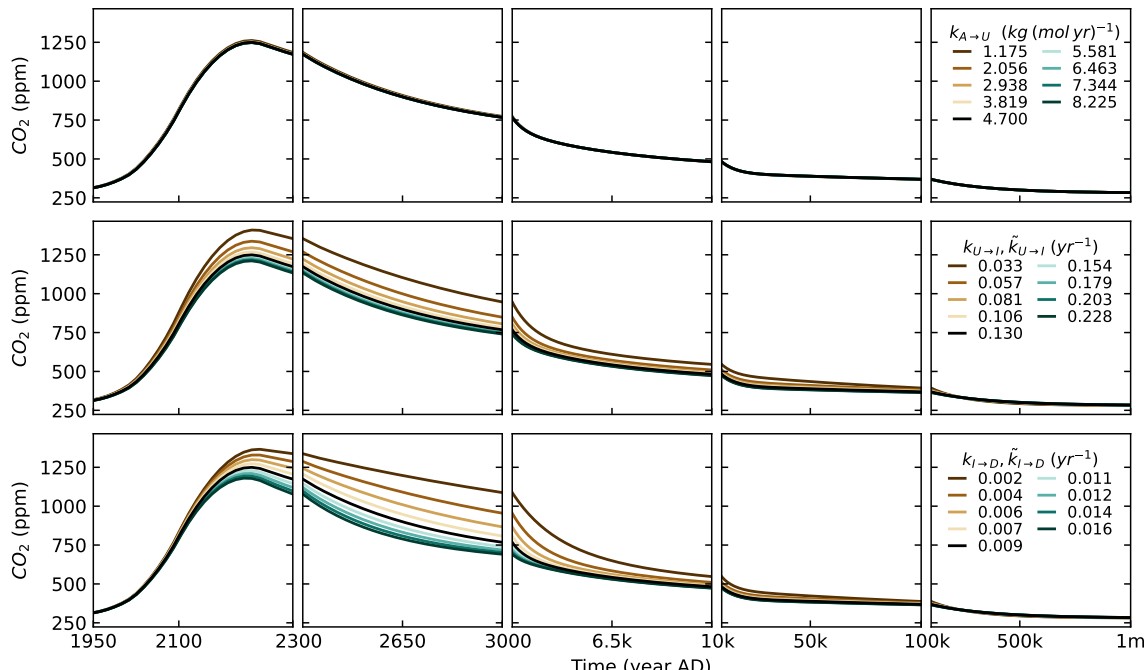

**Figure D1.** Atmospheric $CO_2$ concentration simulated by SURFER v3.0 under the SSP3-7.0 scenario for different values of $k_{A \rightarrow U}$, $k_{U \rightarrow I}$, and $k_{I \rightarrow D}$ (and $\tilde{k}_{U \rightarrow I}$, and $\tilde{k}_{I \rightarrow D}$). Black lines indicate the atmospheric $CO_2$ concentration simulated for the default parameter values, as described in section 2.4.1.

kyr. After that, the ocean, atmosphere and sediments reservoir are in equilibrium, and atmospheric $CO_2$ is regulated by the imbalance between volcanism and weathering. Changing $k_{I \rightarrow D}$ only has a very little effect before 2100 AD, reflecting the longer time scales associated with the deeper ocean. Note that we have varied $\tilde{k}_{U \rightarrow I}$ and $\tilde{k}_{I \rightarrow D}$ simultaneously as $k_{U \rightarrow I}$ and 1110 $k_{I \rightarrow D}$ because these parameters are all linked to the implicit ocean circulation.

Figure D3 shows the sensitivity of atmospheric $CO_2$ to changes in parameters associated with the carbonate and soft-tissue pumps. Overall, varying these parameters has a small or negligible impact. This is due to the preindustrial equilibrium condition that we impose, which creates a compensating effect. For example, if we have stronger pumps initially, we need stronger upwelling fluxes at equilibrium and so higher $k_{I \rightarrow U}$ and $k_{D \rightarrow I}$ (and $\tilde{k}_{I \rightarrow U}$ and $\tilde{k}_{D \rightarrow I}$). This leads to weaker $F_{U \rightarrow I}^{\text{res}}$ 1115 and $F_{I \rightarrow D}^{\text{res}}$ (and $\tilde{F}_{U \rightarrow I}^{\text{res}}$ and $\tilde{F}_{U \rightarrow I}^{\text{res}}$) fluxes which mostly compensates the effects of stronger pumps in transient runs (see equations 43-46). This compensation effect is not exact and we can observe a small impact for changes in $P^{\text{org}}$ and $\phi_I^{\text{org}}$. In comparison, the impact for changes in $P^{\text{CaCO}_3}, \phi_I^{\text{CaCO}_3}$, and $\sigma_{Alk:\text{DIC}}$ are almost non-existent because these parameters are associated to smaller DIC and alkalinity fluxes. Changing $\phi_D^{\text{CaCO}_3}$ has no impact on model results because we keep $P^{\text{CaCO}_3}$





constant. In this case the terms in $F_{acc}$ involving $\phi_{\mathrm{D}}^{\mathrm{CaCO_3}}$ cancel out (see equations 35, 48, and 101). Still, we have kept the parameter to facilitate further model updates. Increasing $P^{\mathrm{org}}$ implies that we need stronger upwelling fluxes to respect the preindustrial condition, and so overall the ocean is better ventilated and it is harder to store carbon in the deep ocean, slowing down atmospheric $CO_2$ uptake. Increasing $\phi_{\mathrm{I}}^{\mathrm{org}}$ has the opposite effect: it leads to a weaker soft-tissue pump, which implies weaker upwelling fluxes and thus a faster $CO_2$ uptake.

Figures D4 and D7 show the sensitivity of atmospheric $CO_2$ to changes in parameters associated with the dissolution of $CaCO_3$ sediments and with weathering fluxes. Atmospheric $CO_2$ is sensitive to changes in $\alpha_{\mathrm{diss}}$ and $\gamma_{\mathrm{diss}}$, but barely to changes in $\beta_{\mathrm{diss}}$, meaning that the deep $CO_3^{2-}$ conentration rather than the mass of erodible $CaCO_3$ sediments, $M_{\mathrm{S}}$, is the dominant driver of dissolution in our parametrisation. With the biggest changes in atmospheric $CO_2$ concentration observed between 3000 yrs AD and 50 kyr AD, these experiments showcase nicely the timescales associated with sediment processes. Changes in $F_{\mathrm{CaCO_3},0}$ and $k_{\mathrm{Ca}}$, associated to carbonate weathering have an negligible impact on $CO_2$ levels compared to changes in $F_{\mathrm{CaSiO_3},0}$ and $k_{\mathrm{T}}$, associated to silicate weathering. This results from silicate weathering rather than carbonate weathering being the process that is ultimately responsible for the draw-down of excess atmospheric $CO_2$, at least in our model. We observe that a larger initial silicate weathering flux ($F_{\mathrm{CaSiO_3},0}$), or a larger response of silicate weathering to warming ($k_{\mathrm{T}}$) leads to a faster uptake of $CO_2$, mostly noticeable after 100 kyr .

Focusing on vegetation in Figure D8, we observe that the impact of varying $k_{\mathrm{A}\to\mathrm{L}}$ is small and limited to years before 2300 AD. This is because the exchanges of carbon between the atmosphere and the land are relatively fast and so the limiting factor to land uptake is rather the amount of carbon that land can store, which is controlled by $\beta_{\mathrm{L}}$. A larger $\beta_{\mathrm{L}}$ means that more carbon can be stored in the land reservoir for a given atmospheric concentration, thus increasing the land sink. Together with $k_{\mathrm{U}\to\mathrm{I}}$ and $k_{\mathrm{I}\to\mathrm{D}}$ (and $\tilde{k}_{\mathrm{U}\to\mathrm{I}}$ and $\tilde{k}_{\mathrm{I}\to\mathrm{D}}$), $\beta_{\mathrm{L}}$ is the parameter that has the biggest influence on $CO_2$ uptake up to the year 6500AD, showing that the fertilisation effect and oceanic invasion of $CO_2$ are the main processes driving atmospheric $CO_2$ draw-down on these timescales in SURFER v3.0.





**Figure D2.** Atmospheric $CO_2$ concentration simulated by SURFER v3.0 under the SSP3-7.0 scenario for different values of $P^{\mathrm{CaCO_3}}$, $\phi_{\mathrm{I}}^{\mathrm{CaCO_3}}$, $\phi_{\mathrm{D}}^{\mathrm{CaCO_3}}$, $P^{\mathrm{org}}$, $\phi_{\mathrm{I}}^{\mathrm{org}}$, and $\sigma_{\mathrm{Alk:DIC}}$. Black lines indicate the atmospheric $CO_2$ concentration simulated for the default parameter values, as described in section 2.4.1.




**Figure D3.** Atmospheric $CO_2$ concentration simulated by SURFER v3.0 under the SSP3-7.0 scenario for different values of $P^{\mathrm{CaCO_3}}$, $\phi_{\mathrm{I}}^{\mathrm{CaCO_3}}$, $\phi_{\mathrm{D}}^{\mathrm{CaCO_3}}$, $P^{\mathrm{org}}$, $\phi_{\mathrm{I}}^{\mathrm{org}}$, and $\sigma_{\mathrm{Alk:DIC}}$. Black lines indicate the atmospheric $CO_2$ concentration simulated for the default parameter values, as described in section 2.4.1.



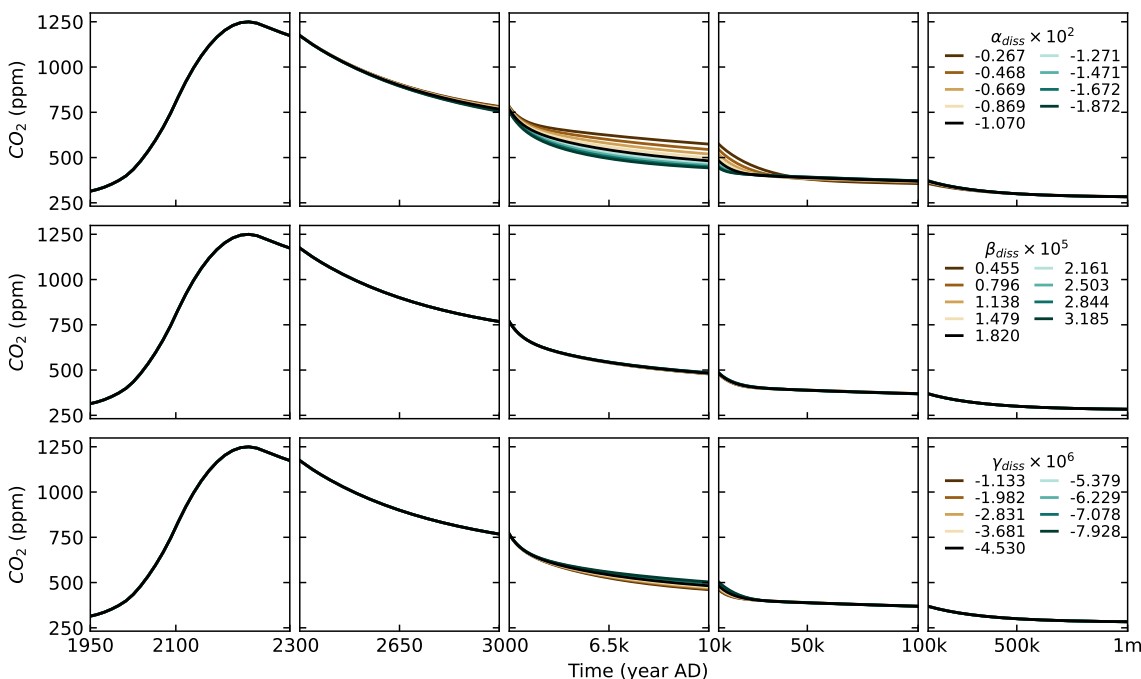

**Figure D4.** Atmospheric $CO_2$ concentration simulated by SURFER v3.0 under the SSP3-7.0 scenario for different values of $\alpha_{\mathrm{diss}}$, $\beta_{\mathrm{diss}}$, and $\gamma_{\mathrm{diss}}$. Black lines indicate the atmospheric $CO_2$ concentration simulated for the default parameter values, as described in section 2.4.1.



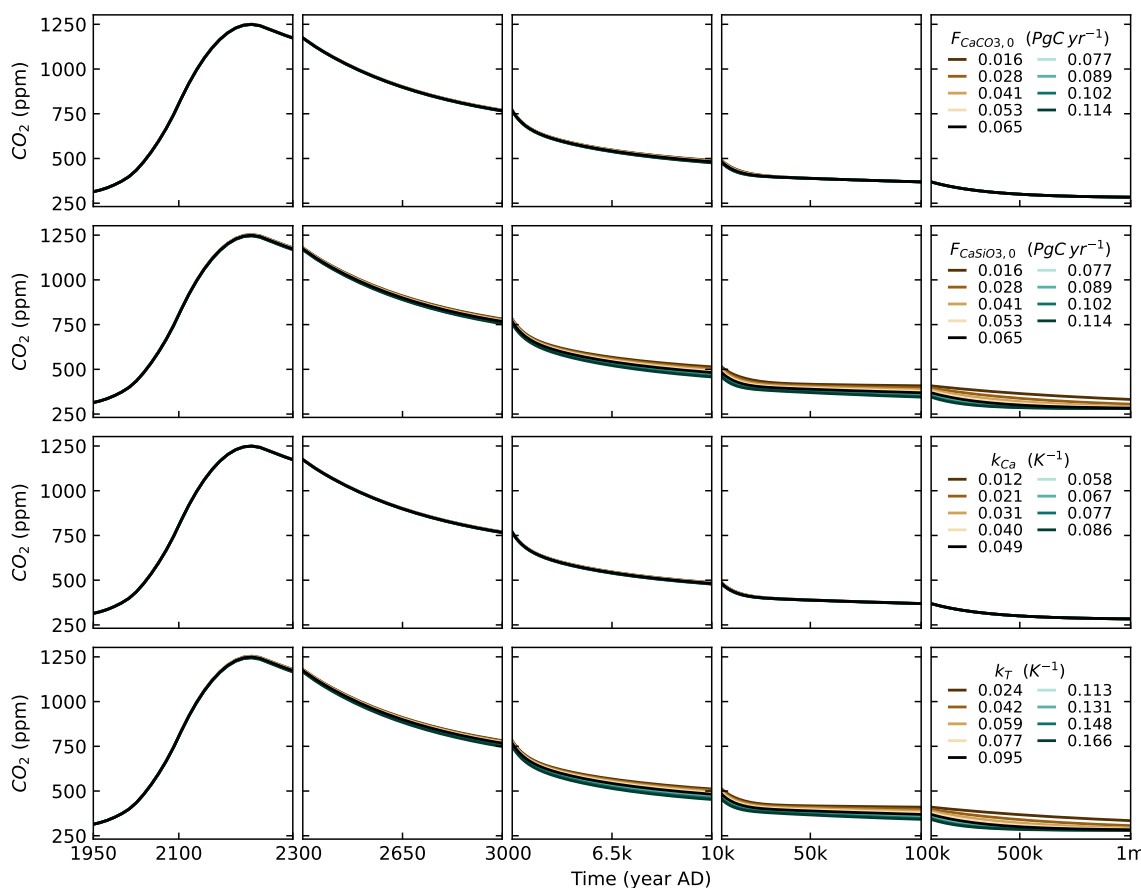

**Figure D5.** Atmospheric $CO_2$ concentration simulated by SURFER v3.0 under the SSP3-7.0 scenario for different values of $F_{CaCO_3,0}$, $F_{CaSiO_3,0}$, $k_{Ca}$, and $k_T$. Black lines indicate the atmospheric $CO_2$ concentration simulated for the default parameter values, as described in section 2.4.1.

We also test how simulated atmospheric $CO_2$ levels, temperatures and sea level rise respond to changes in $\gamma_{U\rightarrow I}$, $\gamma_{I\rightarrow D}$, and $\beta$, which are parameters linked to the climate module of SURFER v3.0. The experimental setup is the same as the one described above.

Figures D9 and D10 show the sensitivity analysis for $\gamma_{U\rightarrow I}$ and $\gamma_{I\rightarrow D}$. Increasing $\gamma_{U\rightarrow I}$ decreases the heat accumulation and thus temperatures in the surface layer, but increases temperature in the intermediate and deep layers. On the other hand, increasing $\gamma_{I\rightarrow D}$ decreases the heat accumulation and temperature in the surface and intermediate layers, but increases the



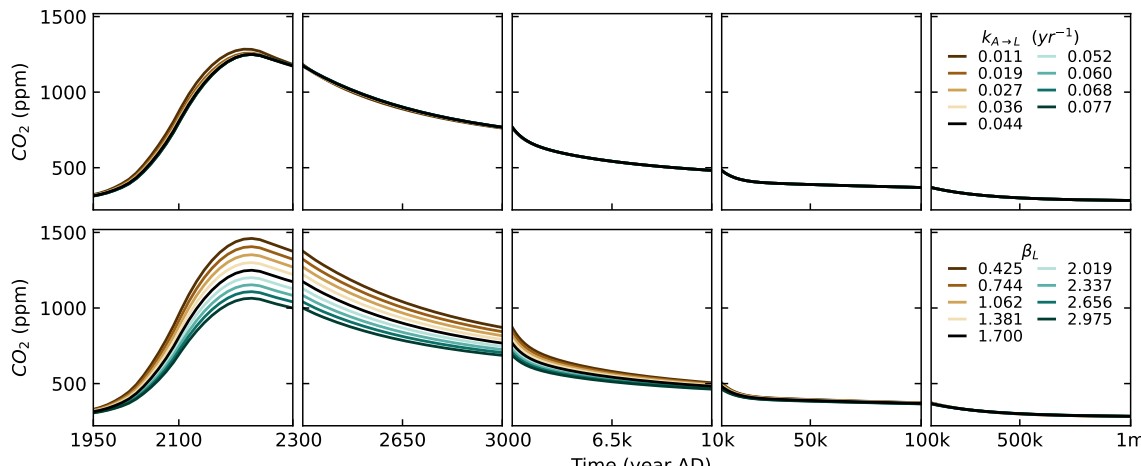

**Figure D6.** Atmospheric $CO_2$ concentration simulated by SURFER v3.0 under the SSP3-7.0 scenario for different values of $k_{A \to L}$, and $\beta_L$. Black lines indicate the atmospheric $CO_2$ concentration simulated for the default parameter values, as described in section 2.4.1.

temperature in the deep layer. For both parameters, an increase leads to an overall increase in thermosteric sea level rise because the deep layer dominates the thermal expansion. Although both parameters impact surface temperature, they have almost no impact on atmospheric $CO_2$ because SURFER v3.0 has a very weak carbon-climate feedback (see discussion, section 5). We also observe little to no impact on temperatures after 10 000 kyr , suggesting that by that time, the ocean has reached thermal equilibrium.





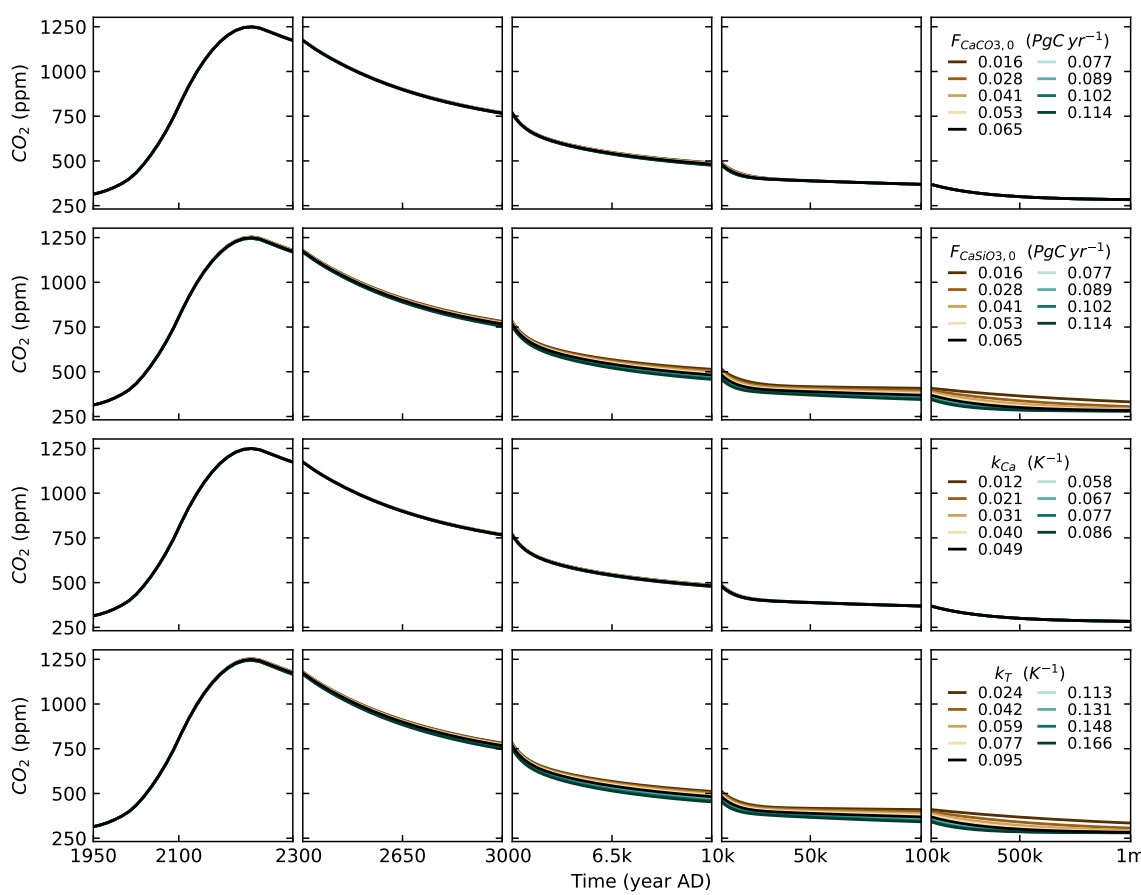

**Figure D7.** Atmospheric $CO_2$ concentration simulated by SURFER v3.0 under the SSP3-7.0 scenario for different values of $F_{\mathrm{CaCO_3},0}$, $F_{\mathrm{CaSiO_3},0}$, $k_{\mathrm{Ca}}$, and $k_{\mathrm{T}}$. Black lines indicate the atmospheric $CO_2$ concentration simulated for the default parameter values, as described in section 2.4.1.



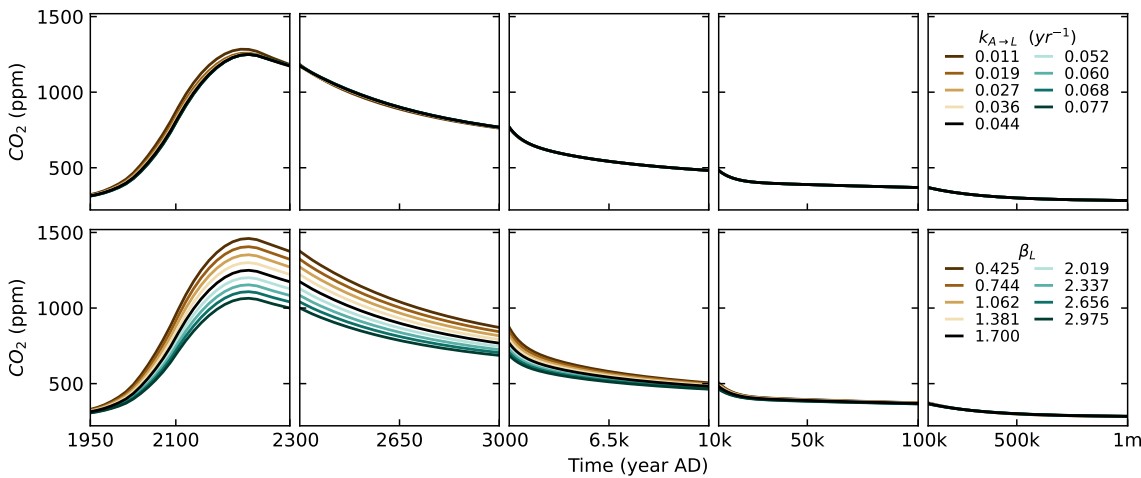

**Figure D8.** Atmospheric $CO_2$ concentration simulated by SURFER v3.0 under the SSP3-7.0 scenario for different values of $k_{A \to L}$, and $\beta_L$.

Black lines indicate the atmospheric $CO_2$ concentration simulated for the default parameter values, as described in section 2.4.1.





**Figure D9.** Atmospheric $CO_2$ concentration, temperatures, and thermosteric sea level rise simulated by SURFER v3.0 under the SSP3-7.0 scenario for different values of $\gamma_{U \rightarrow I}$. Black lines indicate outputs simulated for the default parameter values, as described in section 2.4.1.





**Figure D10.** Atmospheric $CO_2$ concentration, temperatures, and thermosteric sea level rise simulated by SURFER v3.0 under the SSP3-7.0 scenario for different values of $\gamma_{I \to D}$. Black lines indicate outputs simulated for the default parameter values, as described in section 2.4.1.



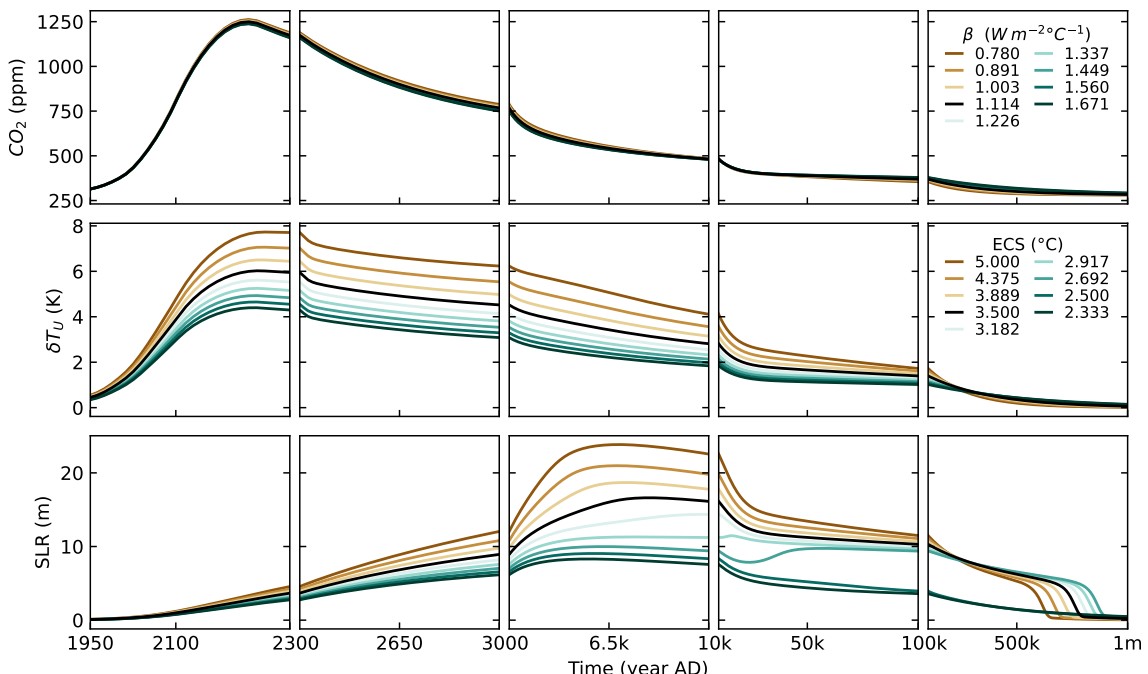

**Figure D11.** Atmospheric $CO_2$ concentration, surface temperature, and total sea level rise simulated by SURFER v3.0 under the SSP3-7.0 scenario for different values of $\beta$. The equilibrium climate sensitivity of SURFER v3.0 is related to $\beta$ through ECS $= F_{2\times}/\beta$. Black lines indicate outputs simulated for the default parameter values, as described in section 2.4.1.

In Figure D11, we investigate SURFER v3.0 's response to changes in $\beta$, the climate feedback parameter. Decreasing $\beta$ while keeping $F_{2\times}$ constant increases the equilibrium climate sensitivity. In this case, we vary $\beta$ in the range 0.780-1.671 W m$^{-2}$ °C$^{-1}$ giving an ECS range of 2.33-5 °C which is within the estimated very likely 2-5°C range given by the IPCC (Smith et al., 2021). Lower values for $\beta$, and thus higher values of climate sensitivity naturally lead to more warming and consequently a higher sea level rise. Big differences in projected sea level rise after 10 kyr AD, even for close $\beta$ values, are the consequence of tipping the Greenland ice sheet. As for $\gamma_{U\rightarrow I}$, and $\gamma_{I\rightarrow D}$, changes in $\beta$ do not impact $CO_2$ levels that much since the total carbon-climate feedback in SURFER v3.0 is weak. For higher values of climate sensitivity, the positive climate-carbon feedback resulting from the temperature-dependent solubility and dissociation constants leads to higher $CO_2$ levels up to around 10 kyr AD. Conversely, the negative carbon-climate feedback from silicate weathering, acting on longer timescales, results in lower $CO_2$ values after 10 kyr AD, and especially after 100 000 kyr.





*Author contributions.* VC, MMM, and MC conceptualised the project. VC, MMM, and MC contributed to the methodology by developing the model. MC managed and supervised the project. VC wrote the model code and carried out model validation. VC did the visualisations. VC wrote the original draft. MC reviewed and edited the paper.

*Competing interests.* The authors declare having no competing interests.

*Acknowledgements.* The authors thank ... . . The authors also thank Michael Eby and Natalie Lord for sharing their data. Victor Couplet is funded as Research Fellow by the Belgian National Fund of Scientific Research (F.S.R-FNRS). Marina Martínez Montero has received funding from the European Union's Horizon 2020 research and innovation programme under grant agreement no. 820970 (TiPES project). Michel Crucifix was funded as Research Director by the Belgian National Fund of Scientific Research (F.S.R-FNRS). The scientific colormap
batlouw (Crameri, 2023) is used in Figure 15.



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
