# Peer review of "SURFER v3.0: a fast model with ice sheet tipping points and carbon cycle feedbacks for short and long-term climate scenarios"

_EGUsphere, 2024_

## Author Response (AR1)

**Author response to Referee comments, and changes to the manuscript**

January 2025

We sincerely thank both reviewers for taking the time to thoroughly review our long paper and for their positive comments and insightful suggestions. We are especially pleased that the reviewers found the paper easy to read and that the model was clear, understandable, and user-friendly. Achieving this was a key objective of our work, which also partly explains the paper's length.

The line numbers used by the Referees refer to the first manuscript. The line numbers we refer to in this response correspond to the marked-up manuscript version, not the revised one.

**Comments from Referee 1**

Review for: SURFER v3.0: a fast model with ice sheet tipping points and carbon cycle feedbacks for short and long-term climate scenarios This paper presents the next iteration of the simple climate model SURFER. The previous version, v2.0, lacked some processes critical for long-term ($> 1 - 10$ kyr) simulations such as volcanic degassing, rock weathering, and ocean alkalinity cycling. V3.0 captures these processes, adds atmospheric methane, and improves or builds on the parameterizations for ice sheet tipping and land use emissions.

The paper is well-written, and I found the model decisions to be reasonable. I appreciated that the authors laid out the specific goals for updating SURFER in the introduction — it offered a clear frame of reference for my review. There's room to make certain formulations more mechanistic, but these limitations don't get in the way of SURFER's ability to capture complex system responses in a simple model. The authors also do a nice job of showing that SURFER v3.0 out-performs v2.0 and captures the basic response of more sophisticated models across a wide range of timescales. On top of that, the code worked out of the box, and it was easy to figure out how to run my own experiments separate from the examples provided. I really commend the authors on how well they've communicated their changes and made the model easy to understand and use.

I think the paper can be accepted after some minor revisions.

**Two points of general feedback:**

- Overall, more citations would be helpful. I mention a few places in my line-by-line comments where I expected a citation and didn't get one, but that issue persists throughout much of the text (especially when laying the background). I know a lot of these decisions come down to personal / editorial preferences, so my feedback isn't very specific. I just encourage the authors to double-check places where citations are absent to decide if it's worth adding any.

We added some citations in the text, particularly in sections 1 and 2 : see L16, L18, L26, L27, L104, L126, L131, L156, L176, L208, L222, L237, L273, L275, L282, L311, L343, L357, L374, L468, L512, L783, L785, L848. A few of these citations are Sarmiento et al. 2006, which was one of our main reference for the carbon cycle dynamics in the ocean.

- The calibration / tuning steps are a bit scattered in the text. I know that they're just qualitative (and I have no issue with that), but for others that might want to test different parameter values it would be very helpful to have a clearer map of what knobs were turned, how, and why. The best way to do this might be to add columns to table 2 (and subsequent, similar tables) for the relevant reference(s) used to tune that parameter and how the reference was used — e.g., was the value tuned to match an independent constraint on the same parameter, or to fit some climate response, etc.

To simplify the model's presentation, we intentionally separated the description of the equations (Sections 2.1, 2.2, and 2.3) from the calibration of the parameters in Section 2.4. Admittedly, this section is quite extensive and includes many numerical details.

We added a column in Table 2 to summarize the calibration targets and references used for the tuning of the carbon cycle. For the climate and sea level rise components of SURFER v3.0, the parameters used are the same as in SURFER v2.0 (Martínez Montero et al. 2022) (except for the thermal expansion coefficients). Therefore, we decided to mention that in the captions of Tables 3 and 4 instead of adding supplementary columns.

**Thoughts on C-cycle component**

I just wanted to acknowledge that the new carbon cycle component is very simplistic. For example, inorganic and organic C export from the upper ocean is fixed — it doesn't respond to climate / pCO2 / riverine inputs, etc. Weathering fluxes only scale with temperature and there's no runoff dependence (this could be highlighted more, especially as the runoff response to GHGs is less certain than the T response). There's also no ocean organic carbon burial, which is a substantial component of global geologic C sequestration.

I think this highly simplified treatment is okay because it's consistent with the rest of the model's complexity, more mechanistic / comprehensive formulations are often still elusive, and a primary goal is to capture the response of more sophisticated models, not the internal dynamics. The simplified treatment limits the questions SURFER v3.0 can be used to answer, but I think the authors reasonably communicate that in the text.

We acknowledge that the carbon cycle in our model is indeed quite simplistic and omits certain processes. As a preliminary exploration, we tested some arbitrary parameterizations of carbon export from the upper ocean as a function of temperature. Alongside a slightly modified parameterization for ice sheets, this allowed for the simulation of glacial cycles. This illustrates nicely that : (1) there are significant processes that are not yet

implemented in the model, and (2) using past climates as benchmarks can provide valuable constraints for integrating these processes. However, this exploration is beyond the scope of the current study and will be addressed in a future publication.

Regarding weathering fluxes, we chose to scale them only with temperature for simplicity and to align with the approach used in Lord et al. 2016 with the model cGENIE, whose simulations we used as a target for long-term atmospheric CO2 levels. Besides SURFER v3.0 lacks a representation of the hydrological cycle. We have added 2 sentences at L348 and L353 in Section 2.7 to emphasize that. Furthermore, as noted by Referee 2 (Jeremy Caves Rugenstein), the relatively high activation energy we use for silicate weathering compared to more recent research may compensate for the absence of a simulated hydrological cycle and runoff effects, at least for silicate weathering. We explain this in new lines L575-580.

Additionally, we have expanded the Discussion (Section 5, L958-974) to elaborate on the role of organic carbon burial in long-term carbon sequestration, as discussed in our response to Referee 2.

**Line by line:**

L15-17: Citations needed for warmest year on record and frequency/intensity of extremes.

Added.

L112-114: Citation needed for the two SURFER v2.0 problems noted here.

We included a citation of the SURFER v2.0 publication to address the issue related to acidification (Martínez Montero et al. 2022). The issue concerning ocean carbon intake, however, was not discussed in the original publication and was identified through subsequent experiments. Both of these issues are evident in Figure 7. To improve clarity, we referenced this figure earlier in the section (L115).

L132: Consider adding (CBW) after "..and water self-ionisation alkalinity" to clarify the subscript in equation 13.

Done. See L135.

L143: Should "dissolved organic carbon" be changed to "dissolved inorganic carbon"?

Done. See L146.

L330: Some references for drivers of $F_{CaCO3}$ and $F_{CaSiO3}$ would be helpful here.

We added the review from Kump et al. 2000. See L343.

L891: No action needed here. Just wanted to acknowledge the colorful tone of this mini paragraph. It doesn't work everywhere, but I'm cool with it in a lengthy model description paper like this

:)

L913-914: Another place where references would help.

We added some.

L976: change "od" to "of".

Done.

Acknowledgments: These are left incomplete (but probably intentionally so). Flagging just in case.

Completed.

**The SURFER_v3.ipynb file**

- First text block under "Setup": replace [url_link] with a url (I recognize this might be intentionally incomplete, just flagging it).

Thanks for checking even the code ! We added the url now that the paper is available online as a preprint.

- At least for me, markdown doesn't like the equation formatting in the text under "Emission scenarios". One option is to edit "$CO_2$" to "$CO_2$" (and repeat for $CH_4$).

Fixed.

**Comments from Jeremy Caves Rugenstein (Referee 2)**

Couplet and co-authors develop an update to SURFER v.2, which includes longer-term carbon cycle processes and additional land surface feedbacks that modify CO2 on long timescales as well as atmospheric CH4. They find that these long-term feedbacks generally act to lower long-term atmospheric CO2 and global temperature, reducing ice sheet melting and resulting in lower sea level rise for a given emission scenario.

I found the paper to be well-written, the equations well-described, and the arguments easy to follow. The figures are well-made and support the contentions in the paper. The references are appropriate. I think these simple models that capture overall Earth system dynamics are incredibly important for the field, so I applaud the authors' efforts to develop this model and make it easy to use. And I hope the field uses these models more often as tools to understand the basics of complex systems. I also think that these model description papers can be rather tedious, but I found this one to be easy to read and I learned quite a bit from it. Overall, I think this paper is ready for publication pending some minor changes, which I detail below.

First, having read the first reviewer's comments, I agree with their thoughts. I also found it somewhat surprising that the alkalinity flux from carbonate and silicate weathering should be split 50/50, when I think there is good evidence that it is more like a 66/34 split (Gaillardet et al., 1999; Moon et al., 2014). Also, the activation energy used in the paper for silicate weathering ($74 \pm 29$ kJ/mol) is on the high side. Later work by West (West, 2012) found a lower number, and more recent work (Brantley et al., 2023) finds an even lower number (ie, 22 kJ/mol).

I don't expect incorporating these values into the paper is likely to make a large difference in the results, and I don't think re-running the simulations to account for this is worthwhile prior to publication. Further, it could be that using a high number in a sense compensates for the lack of a simulated hydrological cycle in the model (a reasonable trade-off to reduce complexity!), which is key to representing the silicate and carbonate weathering fluxes

(Kukla et al., 2023; Maher and Chamberlain, 2014). Nevertheless, these exact numbers are going to change the CO2 emissions scenario that will cause more or less sea-level rise via melting of Greenland or Antarctica and some acknowledgment of these limitations in the discussion would be helpful.

We based our carbonate and silicate weathering fluxes on the work of Lord et al. 2016, whose simulations we used as a target for long-term atmospheric CO2. Following their approach (and that of Colbourn et al. 2013), we adopted a 50/50 split. However, as you pointed out, there is strong evidence suggesting that a 66/34 split is more accurate. Thank you for highlighting this.

We added a few sentences (L563-567) to explain that our approach likely underestimates carbonate weathering fluxes, but that we get similar values as Lord et al. 2016, whose 1 Myr cGENIE runs we use as a calibration target for SURFER. Besides, our sensitivity analysis indicates that doubling $F_{CaCO3,0}$ produces negligible changes in our results.

We also appreciate your pointing out more recent references suggesting lower activation energy values for silicate weathering than the one we used.

We incorporated these references into the manuscript and mentioned that our chosen value may compensate for the lack of a simulated hydrological cycle (as also discussed in our response to Referee 1). See L575-580.

You are correct that variations in these parameters—and others—can affect the long-term contributions of Greenland and Antarctica to sea-level rise. The inclusion of sediment and weathering processes in SURFER v3.0 compared to SURFER v2.0 consistently reduces the long-term contributions of ice sheets, but whether Greenland crosses a tipping point for a specific emissions scenario does indeed depend on the precise parameter choices. To better convey this, we added 2 sentences in the first paragraph of the Discussion. See L891-894.

Second, I'm curious why CaCO3 accumulation on shelves is ignored (line 279). Is the dissolution flux on shelves likely to be minimal, even with high and rapid emissions? A reference here on why this is a reasonable assumption would be helpful or at least an explanation about how incorporating it would be an unreasonable trade-off in terms of model complexity.

Estimates suggest that carbonate accumulation in shallow waters could be of the same order of magnitude as accumulation in open-ocean deep sediments (Milliman 1993; Milliman and Droxler 1996; Vecsei 2004). Moreover, these processes are likely influenced by human activities and anthropogenic emissions (Andersson et al. 2004). Including such processes in SURFER would not necessarily be that difficult—in principle, we could add two sediment boxes associated with the upper and intermediate ocean layers and parameterize the corresponding rain and dissolution fluxes. However, the challenge lies in identifying suitable parameterizations for these accumulation and dissolution fluxes. Shallow waters are characterized by high sedimentation rates, necessitating distinct models or parameterizations (Ridgwell et al. 2007). Additionally, processes such as carbonate fixation by corals and algae would need to be considered. Given the significant uncertainty surrounding these processes and the lack of reliable data to quantify them, we opted to exclude them from our model. This approach is equivalent to assuming that $CaCO_3$ accumulation

on shelves—and the weathering flux required to balance it—remains constant throughout the simulations (Archer et al. 1998; Ridgwell et al. 2007). To clarify this decision, we will added a paragraph in section 2.1.6. See L285-293.

Lastly, while the authors are clear that their goal is to explore sea-level/ice sheet tipping points, their discussion of missing processes that might impact this is short. For example, land surface-vegetation-albedo feedbacks are known to be important in just the recent past (ie, Green Sahara) and might impact temperatures and certainly high-latitude hydroclimate (Feng et al., 2022; Swann et al., 2014).

Since SURFER does not simulate the hydroclimate and lacks spatial extent, it indeed omits important feedbacks such as land surface-vegetation-albedo feedback. While it would be impossible to describe all the processes absent from SURFER, we agree that the discussion would benefit from further elaboration. To address this, we added a new paragraph in the Discussion (L939-948) focusing on vegetation feedbacks.

It is worth noting that the impact of vegetation on global temperatures—whether through albedo changes or indirectly via moisture recycling and cloud cover—is at least partially captured by the climate feedback parameter $\lambda$ (as defined in the IPCC framework (IPCC 2021)). However, tipping points in regional systems, such as the Amazon rainforest, boreal forests, or vegetation cover in the Sahara, could lead to significant $CO_2$ emissions or uptake that are not represented in our simplified parameterization of land-atmosphere carbon fluxes.

Additionally, we would like to clarify that our goal is to explore long-term Earth system trajectories in general, including sea-level and ice-sheet tipping points. We modified the text at L5 and L883-884 to better reflect this nuance.

Another potentially critical feedback involves the role of marine anoxia in changing atmospheric CO2. While the authors point out that changes in marine ecosystem production are neglected, organic carbon burial is also neglected and the geologic record at least has multiple examples of tipping points involving organic carbon burial (ie, ocean anoxic events) that could modify atmospheric CO2. Indeed, the geologic record holds many examples of tipping points on these timescales that have been difficult to simulate except in the simplest of models and yet these tipping points would impact the temperature and therefore sea-level over $10^4$ timescales. Clearly incorporating these processes would be another paper (and a lot more work), but discussing these missing tipping points more in-depth would be worthwhile, along with how they might be incorporated in to future versions of SURFER. This might help the wider community see the value of these simpler models.

We agree that our discussion of organic carbon burial is fairly brief. Implementing organic carbon burial in SURFER could be achieved by adding a sediment reservoir for organic carbon, along with accumulation and burial fluxes similar to those for CaCO3. This modification would be relatively straightforward to implement. However, parameterizing these fluxes would require dependencies on oxygen, which is not currently included in the model, as well as a limiting nutrient for primary production and respiration. As you pointed out, this would involve significantly more work.

Additionally, beyond its impact on atmospheric CO2, organic carbon also plays a critical role in carbonate sedimentation. Specifically, in oxic pore waters, the respiration of organic matter creates acidic micro-environments that enhance the dissolution of $CaCO_3$ sediments (Sarmiento et al. 2006). To address these points, we expanded our discussion of organic carbon burial with two paragraphs. See L958-974.

**Others**

We changed the value of parameter $k_\tau$ (Eq. 72) from 0.05 to 0.001. We did this for consistency with new papers we are currently writing using SURFER, and where we use this new value. This change only modifies slightly Figure 18, which we updated accordingly, but doesn't impact our general results.

We also deleted some figures from the sensitivity analysis (Appendix D) which for some reason appeared twice in the first manuscript.

**References**

Andersson, A. J. and F. T. Mackenzie (2004). "Shallow-water oceans: a source or sink of atmospheric CO2?" en. In: *Frontiers in Ecology and the Environment* 2.7. _eprint: https://onlinelibrary.wiley.com/doi/pdf/10.1890/1540-9295%282004%29002%5B0348%3ASOASOS% pp. 348–353. DOI: `10.1890/1540-9295(2004)002[0348:SOASOS]2.0.CO;2`.

Archer, D., H. Kheshgi, and E. Maier-Reimer (1998). "Dynamics of fossil fuel $CO_2$ neutralization by marine $CaCO_3$". en. In: *Global Biogeochemical Cycles* 12.2, pp. 259–276. DOI: `10.1029/98GB00744`.

Colbourn, G., A. Ridgwell, and T. M. Lenton (2013). "The Rock Geochemical Model (RokGeM) v0.9". en. In: *Geoscientific Model Development* 6.5, pp. 1543–1573. DOI: `10.5194/gmd-6-1543-2013`.

IPCC (2021). *Climate Change 2021: The Physical Science Basis. Contribution of Working Group I to the Sixth Assessment Report of the Intergovernmental Panel on Climate Change.* Vol. In Press. Type: Book. Cambridge, United Kingdom and New York, NY, USA: Cambridge University Press. DOI: `10.1017/9781009157896`.

Kump, L. R., S. L. Brantley, and M. A. Arthur (2000). "Chemical Weathering, Atmospheric CO2, and Climate". en. In: *Annual Review of Earth and Planetary Sciences* 28.Volume 28, 2000. Publisher: Annual Reviews, pp. 611–667. DOI: `10.1146/annurev.earth.28.1.611`.

Lord, N. S. et al. (2016). "An impulse response function for the "long tail" of excess atmospheric $CO_2$ in an Earth system model". en. In: *Global Biogeochemical Cycles* 30.1, pp. 2–17. DOI: `10.1002/2014GB005074`.

Martínez Montero, M. et al. (2022). "SURFER v2.0: a flexible and simple model linking anthropogenic $CO_2$ emissions and solar radiation modification to ocean acidification and sea level rise". en. In: *Geoscientific Model Development* 15.21, pp. 8059–8084. DOI: `10.5194/gmd-15-8059-2022`.

Milliman, J. D. and A. W. Droxler (1996). "Neritic and pelagic carbonate sedimentation in the marine environment: ignorance is not bliss". en. In: *Geologische Rundschau* 85.3, pp. 496–504. DOI: `10.1007/BF02369004`.

Milliman, J. D. (1993). "Production and accumulation of calcium carbonate in the ocean: Budget of a nonsteady state". In: *Global Biogeochemical Cycles* 7.4. Publisher: John Wiley & Sons, Ltd, pp. 927–957. DOI: 10.1029/93GB02524.

Ridgwell, A. and J. C. Hargreaves (2007). "Regulation of atmospheric $CO_2$ by deep-sea sediments in an Earth system model: REGULATION OF $CO_2$ BY DEEP-SEA SEDIMENTS". en. In: *Global Biogeochemical Cycles* 21.2, n/a–n/a. DOI: 10.1029/2006GB002764.

Sarmiento, J. L. and N. Gruber (2006). *Ocean biogeochemical dynamics*. en. OCLC: ocm60651167. Princeton: Princeton University Press.

Vecsei, A. (2004). "A new estimate of global reefal carbonate production including the fore-reefs". In: *Global and Planetary Change* 43.1, pp. 1–18. DOI: 10.1016/j.gloplacha.2003.12.002.

---

## Author Response (AR2)

**Author response to Topic Editor comments, and changes to the manuscript**

**March 2025**

The line numbers we refer to in this response correspond to the latest revised version of the manuscript.

**Comments from Sarah Arndt (Topic editor)**

You estimate alkalinity based on the carbonate, borate and water self-ionisation alkalinity (CBW). This is fine for oxygenated oceans, but inaccurate under anoxic/euxinic conditions because this approach neglects the contribution of TH2S. Since the model is also suitable for paleoclimate simulations, I suggest that you highlight this limitation in the appropriate section so that future users are aware.

We have added two sentences, new lines 139-143 : It is also important to note that, despite these improvements, our treatment still omits certain chemical species that may become relevant under specific conditions. For instance, the contributions of $H_2S$ and $HS^-$ to alkalinity are non-negligible in anoxic or euxinic environments (Xu et al. 2017).

**References**

Xu, Y.-Y., D. Pierrot, and W.-J. Cai (2017). "Ocean carbonate system computation for anoxic waters using an updated CO2SYS program". In: *Marine Chemistry*. SI: Honoring Frank Millero 195, pp. 90–93. DOI: `10.1016/j.marchem.2017.07.002`.